# A Fast Federated Method for Minimax Problems with Sequential Convergence Guarantees

## Abstract

Federated learning (FL) has recently been actively studied to collaboratively train machine learning models across clients without directly sharing data and to address data-hungry issues. Many FL works have been focusing on minimizing a loss function but many important machine learning tasks such as adversarial training, GANs, fairness learning, and AUROC maximization are formulated as minimax problems. In this paper, we propose a new federated learning method for minimax problems. Our method allows client drift and addresses the data heterogeneity issue. In theoretical analysis, we prove that our method can improve sample complexity from $O(\epsilon^{-3})$ to $O(\epsilon^{-2})$. We also give convergence guarantees for the updates of the model parameters, i.e., the sequences generated by the method. Given the Kurdyka-Łojasiewicz (KL) exponent of a novel potential function related to the objective function, we demonstrate that the sequences generated by our method converge finitely, linearly, or sublinearly. Our assumptions on the KL property are weaker than previous work on the sequential convergence of centralized minimax methods. Additionally, we further weaken the KL assumption by deducing the KL exponent of the maximizer-dependent potential function from that of the maximizer-free function. We validate our federated learning method on AUC maximization tasks. The experimental results demonstrate that our method outperforms state-of-the-art federated learning methods when the distributions of local training data are non-IID.

## 1 Introduction

In recent years, federated learning (FL) has garnered significant attention within the machine learning community, owing to its wide real-world applications in finance, healthcare, edge computing, AIoT, and more. Federated learning allows multiple clients to collaboratively train the same model locally on their own devices. Once trained, the local models are sent to a central server, where they are aggregated, and the updated global model is returned to the clients for further local training. This decentralized approach enables the training of machine learning models using datasets from different clients without the need for data sharing. Additionally, it avoids the transfer of large datasets to a central server, thereby reducing bandwidth requirements and associated costs.

The classical federated learning problem focuses on minimizing a loss function using local training datasets. However, many emerging scenarios, such as adversarial training (Tramèr et al., 2018; Bai et al., 2021), distributionally robust optimization (Levy et al., 2020; Gao & Kleywegt, 2023; Madras et al., 2018), generative adversarial networks (GANs) (Goodfellow et al., 2014), and AUROC (Area Under the ROC Curve) maximization (Lei & Ying, 2021), often formulate their objectives as minimax optimization problems. While centralized methods for solving minimax problems are well-explored, federated learning methods for minimax optimization are still in their early stage. These problems face similar challenges as traditional federated learning, particularly regarding data sharing and communication overhead. Hence, it is necessary develop federated methods for these minimax problems.

Table 1: Local(L) SDGA (Sharma et al., 2022), Momentum Local (ML) SGDA (Sharma et al., 2023), FedSGDA (Wu et al., 2023), FEDNEST (Tarzanagh et al., 2022). BH=Bounded Heterogeneity Assumption, F/P=Partial/Full attendance, $\alpha$ is the KL exponent, $\rho_1 \in (0, 1)$, $b$ and $c$ are constants.

| | F/P | Free of BHA | Sample Complexity $(\mathbb{E}\text{dist}(0, \nabla \sum_{i=1}^{n} \frac{1}{n} f_i(z^t) + \partial g(z^t)))$ | Model Parameter Convergence $(\|z^t - z^*\|)$ |
|---|---|---|---|---|
| LSDGA | F | ✗ | $O(\kappa^4 n^{-1} \epsilon^{-4})$ | ✗ |
| MLSGDA | P | ✗ | $O(\kappa^4 n^{-1} \epsilon^{-4})$ | ✗ |
| FedSGDA | F | ✗ | $O(\kappa^3 n^{-1} \epsilon^{-3})$ | ✗ |
| FEDNEST | P | ✗ | $O(\kappa^3 \epsilon^{-4})$ | ✗ |
| FedSGDA | F | ✗ | $O(\kappa^3 n^{-1} \epsilon^{-3})$ | ✗ |
| Ours | P | ✓ | $O(\kappa^2 \log(\kappa) n^{-1} \epsilon^{-2})$ | finite step convergence when $\alpha = 0$ $O(\rho_1^t)$ (linear convergence) when $\alpha \in (0, \frac{1}{2}]$ $O(t^{-\frac{1}{4\alpha-2}})$ (sublinear convergence) when $\alpha \in (\frac{1}{2}, 1)$ |

In this work, we focus on developing federated methods specifically for minimax optimization problems. We consider the following general formulation:

$$\min_{x \in \mathbb{R}^l} \max_{y \in \mathbb{R}^d} \frac{1}{n} \sum_{i=1}^{n} f_i(x, y) + g(x), \tag{1}$$

where each $f_i(x, y) = \sum_{j \in \mathcal{D}_i} f(x, y; \xi_j)$, with $\mathcal{D}_i$ being the dataset of the $i$th client and $\xi_j$ representing individual data points within it. Here, $f$ is a smooth function that is nonconvex in $x$ and strongly concave in $y$, and $g$ represents a proper closed function. Examples of strongly concave $f$ include fairness classification problems (Nouiehed et al., 2019), adversarial training (Sinha et al., 2017), and GAN training (Vlatakis-Gkaragkounis et al., 2021). Common choices for $g$ include convex regularizers or indicator functions corresponding to convex constraints. In this work, we assume that the proximal operator for $g$ is easy to compute.

A key challenge in federated minimax optimization lies in handling the max problem nested within the min problem, particularly when training must occur locally. In centralized settings, the Gradient Descent Ascent (GDA) method is a classical approach to minimax problems. To extend this to federated learning, one could adapt GDA to the FedAvg method, resulting in LocalSGDA (Deng et al., 2020). Other variations, such as Momentum Local SGDA (Sharma et al., 2022), accelerate convergence by adding momentum to local updates, while FedSGDA+ (Wu et al., 2023) further reduces complexity. However, these methods require all clients to participate in every training round, which introduces the risk of client drift due to unstable network connections. To address this, we propose methods that allow only a subset of clients to participate in each training round.

In addition to client drift, data heterogeneity—where local data distributions vary significantly—poses another challenge in federated learning. This heterogeneity can slow down training and reduce the model's performance. Previous works (Sharma et al., 2023; 2022; Wu et al., 2023) have proposed methods to address heterogeneity, assuming bounds on the degree of heterogeneity and studying its impact on convergence complexity. However, in real-world scenarios, these bounds can be large, leading to loose convergence guarantees. Our work introduces methods that offer convergence guarantees without relying on these heterogeneity bounds.

Moreover, while much of the existing research focuses on the complexity of federated learning methods—such as the convergence of $\frac{1}{T} \sum_{t=1}^{T} \mathbb{E} \text{dist}(0, \nabla \sum_{i=1}^{n} \frac{1}{n} f_i(z^t) + \partial g(z^t))$, ($z^t$ representing model parameters), little attention has been given to the convergence of the model parameters themselves. Even for minimization problems, such as those tackled by the classical LocalSGD method (Stich, 2019), the primary focus has been on complexity rather than parameter convergence. Understanding the convergence of model parameters is crucial for evaluating the method's ability to reach a solution. To the best of our knowledge, parameter convergence has only been studied for strongly convex minimization problems in federated learning (Pathak & Wainwright, 2020). In this work, we provides the first analysis of parameter convergence for nonconvex minimax problems.

## 1.1 CONTRIBUTIONS

In this work, we develop a novel federated learning method specifically designed for minimax optimization problems, addressing the unique challenge of solving nested minimax problems in a federated setting. Our approach allows for partial client participation during training rounds,

mitigating client drift caused by unstable network conditions. Additionally, it effectively handles data heterogeneity without relying on strict bounds for data distribution discrepancies, ensuring robust convergence in real-world applications. By introducing a new termination criterion for local training, we enhance the sample complexity of existing federated minimax methods, reducing the complexity from $O(\epsilon^{-3})$ to $O(\epsilon^{-2})$ while maintaining a fixed number of local iterations.

In addition, we provide convergence guarantees for the sequence of model parameters generated by the method, which we refer to as *sequential convergence*. We demonstrate that when all clients participate in training and the local solvers are deterministic, the accumulation points of the sequence generated by our method converge to a stationary point. Furthermore, we establish the convergence rate of the sequence in nonsmooth and nonconvex settings. To achieve this, we leverage the Kurdyka-Łojasiewicz (KL) framework, which specializes in analyzing sequence convergence in nonsmooth, nonconvex cases (Attouch et al., 2010; Li & Pong, 2018; Attouch et al., 2013; Bolte et al., 2017). We show that, depending on the KL exponent of the potential function, the sequence generated by our method converges finitely, linearly, or sublinearly when the KL exponent is $0$, $(0, \frac{1}{2}]$, or $(\frac{1}{2}, 1)$, respectively.

> **Our method is the first one in federated learning that is able to have sequential convergence guarantees in nonconvex nonsmooth settings.**

Furthermore, we weaken the KL assumptions made on the potential function compared to previous work on sequential analysis for the centralized minimax problem in Chen et al. (2021). In their work, the potential function depends on the maximizer $y(x) := \arg\max f(x, y)$ and the maximum function $f(x) := \max_y f(x, y)$. The potential nonconvexity and nonsmoothness of the max function generally make its subgradient discontinuous, posing challenges in calculating its KL exponent. In contrast, our potential function does not rely on $y(x) := \arg\max f(x, y)$. We introduce a calculus rule (Proposition 3) to deduce the KL exponent of our potential function directly from the maximizer-free function. As a result, our analysis offers a weaker assumption for sequential convergence in federated learning methods for minimax optimization problems.

We apply our method to the AUC maximization problem in federated learning, particularly under conditions of data heterogeneity. Our experiments demonstrate that the proposed method outperforms existing federated minimax approaches in both efficiency and performance.

## 1.2 RELATED WORK

**Federated learning for minimization problem** Classical federated learning methods for minimization problem include FedAvg (McMahan et al., 2017), LocalSGD (Stich, 2019), FedDualAvg, (Yuan et al., 2021a), FedSplit (Pathak & Wainwright, 2020) and SCAFFOLD (Karimireddy et al., 2020). In order to address the heterogeneity problem in FL, federated splitting methods are proposed, see Yuan et al. (2021a); Li et al. (2020); Reddi et al. (2021); Pathak & Wainwright (2020); Tran-Dinh et al. (2021) for examples. When the objective is minimizing a strongly convex objective function, Stich (2019) shows the convergence rate of LocalSGD is $O(1/nTb)$, where $n$ is the number of clients, $b$ is the batch size and $T$ is the communication round. On the other hand, Pathak & Wainwright (2020) shows the sequence generated by their proposed method converges linearly when the objective function is strongly convex. Our method is closely related to the FedDR method for the minimization problem in Tran-Dinh et al. (2021). However, our work differs from Tran-Dinh et al. (2021) in three perspectives: 1. We work on minimax problems. The existence of the maximization problem raises new challenges in theoretical analysis. To address this challenge, we propose new potential functions related to the variables in the maximization problem and are key to all our analysis. 2. We provide comprehensive sequential convergence analysis. Our result is also new when our method degenerates to solve the minimization problems in federated learning. 3. We conducted further investigation on the KL assumption used for analyzing the minimax problems. The existing studies on the KL property for minimax problems are quite few. Li & So (2022); Zheng et al. (2023) investigate a global KL property. Li & So (2022) show that when the objective function is nonconvex in $x$ and nonconcave in $y$, if the objective function is a KL function with respect to $y$ with an exponent in $[0, \frac{1}{2}]$, their method can achieve optimal iteration complexity. In Zheng et al. (2023), the authors propose a unified single-loop algorithm for solving centralized nonconvex-nonconcave, nonconvex-concave, and convex-nonconcave minimax problems. Under a one-sided KL assumption,

they show that the proposed method achieves a complexity of $O(\epsilon^{-4})$ in all cases and can improve upon previously existing complexity results in the same scenarios under specific KL exponents. On the other hand, Chen et al. (2021) also analyzes the sequential convergence of methods for the centralized minimax problem. Compared with Chen et al. (2021), we weakened the KL assumptions made on the potential function. In their work, the potential function relies on the maximizer $y(x) := \arg\max f(x, y)$ and the maximum function $f(x) := \max_y f(x, y)$. The exact form of $y(x)$ is not known, which makes verifying the KL exponent difficult. In our work, the potential function does not rely on $y(x) := \arg\max_y f(x, y)$, and we provide Proposition 3 to deduce the KL exponent of the maximizer-dependent potential function from that of the maximizer-free function. Therefore, our analysis provides a weaker assumption for the sequential convergence analysis of the method for the minimax optimization problem.

**Federated methods for minimax** Li et al. (2023); Deng et al. (2020); Peng et al. (2020) are among the early works that proposed federated minimax methods for adversarial training problems. Sharma et al. (2022) investigated local stochastic gradient descent ascent in nonconvex-concave and nonconvex-nonconcave settings. Their analysis assumed an equal number of SGDA-like local updates with full client participation, whereas our method allows for different local updates and partial client participation. Sharma et al. (2023) proposed a federated minimax optimization framework that includes local SGDA as a special case. They analyzed the convergence of the proposed algorithm under a global heterogeneity assumption that addresses inter-client data and system heterogeneity. Wu et al. (2023) analyzed the nonconvex-strongly-concave case and showed that their proposed method has a gradient complexity of $O(\kappa^2 n^{-1} \epsilon^{-3})$. Tarzanagh et al. (2022) proposed FEDNEST to address the general bilevel federated learning problem and discuss the minimax problem as a special case.

In contrast to the previous work on federated learning minimax methods, we do not assume heterogeneity bound assumption while achieving a smaller sample complexity. More importantly, we have convergence guarantees for the updates of the model parameters in nonconvex settings. This makes our method novel not only among federated minimax methods but also among federated minimization methods. We summarize the comparison in Table 1.

## 2 PRELIMINARIES

We denote $\mathbb{R}^n$ as the $n$-dimensional Euclidean space with inner product $\langle \cdot, \cdot \rangle$ and Euclidean norm $\| \cdot \|$. We denote the unit ball in $\mathbb{R}^n$ as $\mathcal{B}(0, 1)$. We denote the set of positive real value as $\mathbb{R}_{++}$. Given a point $x \in \mathbb{R}^n$ and a set $A$, we denote the distance from $x$ to $A$ as $d(x, A)$. An extended-real-valued function $f : \mathbb{R}^n \to [-\infty, \infty]$ is said to be proper if $\text{dom } f := \{x \in \mathbb{R}^n : f(x) < \infty\}$ is not empty and $f$ never equals $-\infty$. We say a proper function $f$ is closed if it is lower semicontinuous. Following Definition 8.3 of Rockafellar & Wets (1998), the regular subdifferential of a proper function $f : \mathbb{R}^n \to [-\infty, \infty]$ at $x \in \text{dom } f$ is defined as: $\hat{\partial} f(x) := \left\{ \xi \in \mathbb{R}^n : \liminf_{z \to x, \, z \neq x} \frac{f(z) - f(x) - \langle \xi, z-x \rangle}{\|z - x\|} \geq 0 \right\}$. The (limiting) subdifferential of $f$ at $x \in \text{dom } f$ is defined as $\partial f(x) := \left\{ \xi \in \mathbb{R}^n : \exists x^k \xrightarrow{f} x, \xi^k \to \xi \text{ with } \xi^k \in \hat{\partial} f(x^k), \forall k \right\}$, where $x^k \xrightarrow{f} x$ means both $x^k \to x$ and $f(x^k) \to f(x)$. For $x \notin \text{dom } f$, we define $\hat{\partial} f(x) = \partial f(x) = \emptyset$. We denote $\text{dom } \partial f := \{x : \partial f(x) \neq \emptyset\}$. When $f$ is convex, the limiting subdifferential reduces to the classical subdifferential in convex analysis.

For a proper function $f : \mathbb{R}^n \to [-\infty, \infty]$, we denote the proximal operator of $f$ as $\text{Prox}_{\beta f}(x) := \text{Arg} \min_{z \in \mathbb{R}^n} \left\{ f(z) + \frac{1}{2\beta} \|z - x\|^2 \right\}$.

Next, we make a general assumption on equation 1.

**Assumption 1.** *For equation 1, we assume the followings hold:*

*(i) Each $f_i$ is strongly concave in $y$ with modulus $\mu > 0$.*

*(ii) Each $f_i$ is differentiable and $\nabla f_i$ is Lipschitz continuous with modulus $L_f$.*

For the maximum of a strongly concave function, we have the following property, see Lin et al. (2020); Huang et al. (2021); Chen et al. (2021) for examples.

---

**Algorithm 1** Fast Federated Minimax DR (FFMDR) method for equation 1

---

1: Input: $x_i^0, z_i^0, y_i^0, \Upsilon_{i,0}$. Set $w_i^0 = z_i^0$. Set $\epsilon_{i,w} > 0$, $\beta \in (0, \frac{1}{L})$. Let $t = 0$.
2: Sample clients $\mathcal{S}^t \subseteq \{1, \dots, n\}$ according to Assumption 2. For each client $i \in \mathcal{S}^t$:
   Let
$$x_i^{t+1} = x_i^t + z^t - w_i^t \tag{2}$$
   Find an approximate solution $(w_i^{t+1}, y_i^{t+1})$ to $\min_{w_i} \max_{y_i} r_{i,t+1}(w_i, y_i)$ such that equation 10 is satisfied, where $r_{i,t+1}$ is defined in equation 7.
   Let $\tilde{z}_i^{t+1} = 2w_i^{t+1} - x_i^{t+1}$.
3: For the server: Let
$$z^{t+1} = \text{Prox}_{\frac{\beta}{n}g} \left( \frac{1}{n} \sum_{i=1}^n \tilde{z}_i^{t+1} \right) \tag{3}$$

4: If a termination criterion is not met, let $t = t + 1$ and go to Step 2.

---

**Proposition 1.** *Consider equation 1. Suppose Assumption 1 holds. Then for any $x$, there exists unique $y(x)$ such that $F_i(x) = f_i(x, y(x))$. In addition, $F_i$ is continuously differentiable and $\nabla F_i(x) = \nabla_x f_i(x, y(x))$ is Lipschitz continuous with modulus $L := L_f(1 + \kappa)$, where $\kappa := \frac{L_f}{\mu}$.*

We say $x$ is a stationary point of *equation* 1 if it satisfies $0 \in \nabla \sum_{i=1}^n \frac{1}{n} f_i(x) + \partial g(x)$. Thanks to Exercise 8.8 and Theorem 10.1 of Rockafellar & Wets (1998), we know that if $x$ is a local minimizer of equation 1, it is a stationary point.

Now we give the definition of the KL property.

**Definition 1 (Kurdyka-Łojasiewicz property and exponent).** *A proper closed function $f : \mathbb{R}^n \to (-\infty, \infty]$ is said to satisfy the Kurdyka-Łojasiewicz (KL) property at an $\hat{x} \in \text{dom}\,\partial f$ if there are $a \in (0, \infty]$, a neighborhood $V$ of $\hat{x}$ and a continuous concave function $\varphi : [0, a) \to [0, \infty)$ with $\varphi(0) = 0$ such that*

    *(i) $\varphi$ is continuously differentiable on $(0, a)$ with $\varphi' > 0$ on $(0, a)$;*

    *(ii) for any $x \in V$ with $f(\hat{x}) < f(x) < f(\hat{x}) + a$, it holds that $\varphi'(f(x) - f(\hat{x}))\text{dist}(0, \partial f(x)) \geq 1$.*

*If $f$ satisfies the KL property at $\hat{x} \in \text{dom}\,\partial f$ and $\varphi$ can be chosen as $\varphi(\nu) = a_0 \nu^{1-\alpha}$ for some $a_0 > 0$ and $\alpha \in [0, 1)$, then we say that $f$ satisfies the KL property at $\hat{x}$ with exponent $\alpha$. A proper closed function $f$ satisfying the KL property at every point in $\text{dom}\,\partial f$ is called a KL function, and a proper closed function $f$ satisfying the KL property with exponent $\alpha \in [0, 1)$ at every point in $\text{dom}\,\partial f$ is called a KL function with exponent $\alpha$.*

Many functions are KL functions. It is known that proper closed semi-algebraic functions (i.e., functions whose graphs are unions and intersections of polynomial functions) satisfy the KL property, see Attouch et al. (2010); Li & Pong (2018); Attouch et al. (2013); Bolte et al. (2017). Semi-algebraic functions include widely used losses such as quadratic loss, L2 loss, Huber loss, hinge loss, and 0-1 loss. KL property is a general property in convergence analysis when the considered function is not smoothness.

## 3 FAST FEDERATED MINIMAX DR METHOD

The proposed Fast Federated Minimax DR (FFMDR) method is presented in Algorithm 1. The idea is based on the Douglas-Rachford splitting method (Lions & Mercier) for the following reformation of equation 1:

$$\min_X \underbrace{\frac{1}{n} \sum_{i=1}^n F_i(x_i)}_{F(X)} + \underbrace{g(x_1) + \delta_{\mathcal{C}}(x_1, \dots, x_n)}_{\tilde{g}(X)}, \tag{4}$$

where $F_i(x_i) := \max_{y_i \in \mathbb{R}^d} f_i(x_i, y_i)$, $X = (x_1, \ldots, x_n)$ and $\mathcal{C} = \{X : x_1 = x_2 = \cdots = x_n\}$. The Classic DR method (Lions & Mercier) to equation 4 is as follows: pick any $X^0$, let $Z^0 = X^0$ and $W^0 = \text{prox}_{\beta F}(X^0)$. Then for $t = 0, \ldots, T$, update:

$$
\begin{aligned}
X^{t+1} &= X^t + Z^t - W^t, \\
W^{t+1} &= \text{Prox}_{\beta F}(X^{t+1}), \\
Z^{t+1} &= \text{Prox}_{\beta \tilde{g}}(2W^{t+1} - X^{t+1}).
\end{aligned}
\tag{5}
$$

Noting that $F_i$ in equation 1 is a maximization function and $F$ is separable, the update of $W^t$ in equation 5 is equivalent to

$$
W^{t+1} = \min_{W} \max_{Y} \sum_i f_i(w_i, y_i) + \frac{1}{2\beta} \|w_i - x_i^{t+1}\|^2,
\tag{6}
$$

where $W = (w_1, \ldots, w_n)$ and $Y = (y_1, \ldots, y_n)$. The above problem is a minimax problem and cannot be solve exactly in the federated setting. This requires us to consider an efficient method that can find an good inexact solution to equation 6. We notice that equation 6 is a smooth strongly convex strongly concave (SC-SC) minimax problem. Since we let $\beta < \frac{1}{L}$, Proposition 1 guarantees the existence of the unique solution to the minimax subproblem.

Denote

$$
r_{i,t+1}(w_i, y_i) := f_i(w_i, y_i) + \frac{1}{2\beta} \|w_i - x_i^{t+1}\|^2.
\tag{7}
$$

Then equation 6 is equivalent to

$$
\min_{w_i} \max_{y_i} r_{i,t+1}(w_i, y_i),
\tag{8}
$$

for $i = 1, \ldots, n$. Then, we only need an inner solver to solve a SC-SC smooth minimax problem. Many methods such as those in Benjamin et al. (2022); Fallah et al. (2020); Lin et al. (2020); Kovalev & Gasnikov (2022); Palaniappan & Bach (2016) can be applied as an inner solver for our subproblem. On the other hand, to have better convergence gurantees, we need an efficient termination criterion to terminate the inner solver. In the following lemma, we show how the SAGA in Palaniappan & Bach (2016) can be terminated in constant iterations when satisfying a termination criterion that depends on the current updates.

**Proposition 2.** *Suppose $r : \mathbb{R}^l \times \mathbb{R}^d \to \mathbb{R}$ is a $\mu_w$-strongly convex $\mu_y$ strongly convex smooth function. Suppose $\nabla r$ is Lipschitz continuous with modulus $l$. Apply SAGA in Palaniappan & Bach (2016) to solve $\min_w \max_y r(w, y)$. Let $(w^k, y^k)$ be the $k_{\text{th}}$ iteration of SAGA. Let $(\bar{w}, \bar{y})$ satisfies $\nabla r(\bar{x}, \bar{y}) \neq 0$. Let $\epsilon_w > 0$. Then there exists $k = O(\max\{\frac{l}{m}, \log(\kappa)\})$ such that*

$$
\mathbb{E} \left\| (w^{k+1}, y^{k+1}) - (w_\star, y_\star) \right\|^2 \leq \epsilon_w \mathbb{E} \| (\bar{w}, \bar{y}) - (w^{k+1}, y^{k+1}) \|^2,
\tag{9}
$$

*where $(x^*, y^*)$ is the unique solution.*

In inspired by equation 9, we propose to terminate the solver used in client $i$ for solving equation 8 when[1]

$$
\mathbb{E}_t \left\| (w_i^{k+1}, y_i^{k+1}) - (w_{i,\star}^{t+1}, y_{i,\star}^{t+1}) \right\|^2 \leq \epsilon_{i,w} \mathbb{E}_t \Upsilon_{i,t+1},
\tag{10}
$$

where $(w_{i,\star}^{t+1}, y_{i,\star}^{t+1})$ is the exact solution to equation 8 and

$$
\Upsilon_{i,t+1} := \| (w_i^t, y_i^t) - (w_i^{t+1}, y_i^{t+1}) \|^2.
\tag{11}
$$

On the other hand, using the first-order optimality condition of the problem in the update of $z^t$ in equation 5, $Z^{t+1}$ in equation 5 is equivalent to $(\underbrace{z^{t+1}, \ldots, z^{t+1}}_{n's})$ with $z^{t+1} = \text{Prox}_{\frac{\beta}{n} g}(\frac{1}{n} \sum_i (2w_i^{t+1} - x_i^{t+1}))$, see Appendix of A.1 in Tran-Dinh et al. (2021) for more details.

Finally, considering the cliendt drift, we make the following assumption.

**Assumption 2.** *At each round, the client $i$ has the probability $p_i \in (0, 1]$ to attend the training.*

Based on this fact, Assumption 2 and Proposition 2, we obtain Algorithm 1.

---

[1]We denote $E_t \xi$ as the expectation of the outputs $\xi$ of local stochastic solver conditioned on $\{x_1^t, \ldots, x_n^t\}, \{y_1^t, \ldots, y_n^t\}, \{z^t\}, \{w_1^t, \ldots, w_n^t\}$.

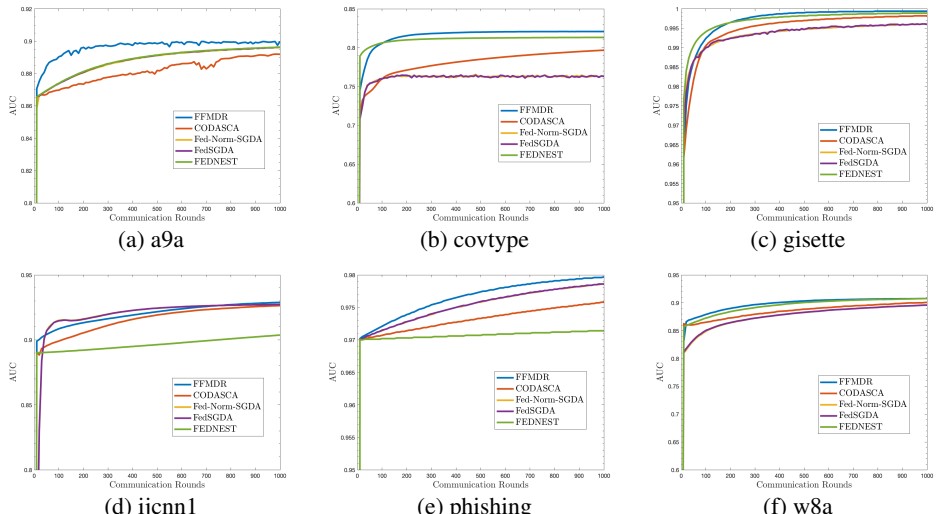

Figure 1: AUC values w.r.t. communication rounds on test dataset: a9a, covtype, gisette, ijcnn1, phishing and w8a.

# 4 CONVERGENCE ANALYSIS

## 4.1 SAMPLE COMPLEXITY OF ALGORITHM 1

In this section, we analyze Algorithm 1 in a general stochastic case. We first present a descent-type lemma of a new potential function.

**Theorem 1.** *Consider equation 1. Suppose Assumptions 1 and 2 hold. Assume $\frac{1}{\beta} > L$, where $L$ is defined as in Proposition 1. Let $\{(x_1^t, \ldots, x_n^t)\}, \{(y_1^t \ldots, y_n^t)\}, \{(w_1^t, \ldots, w_n^t)\}, \{z^t\}$ be generated by Algorithm 1. Let $L$ be the one in Proposition 1. Given a $\delta > 0$, define*

$$H(X, W, Z, Y, W', Y') := F(W) + \tilde{g}(Z) + \frac{1}{2\beta}\left(\|X - W\|^2 - \|X - Z\|^2\right) + \frac{1}{\beta}\|W - Z\|^2$$

$$+ \frac{\delta}{\beta}\|W - W'\|^2 + \frac{1}{12L^2}\sum_i p_i\|(y_i, w_i) - (y_i', w_i')\|^2. \tag{11}$$

*where $F$ and $\tilde{g}$ is defined in equation 4. Denote $X^t = (x_1^t, \ldots, x_n^t)$, $Y^t = (y_1^t, \ldots, y_n^t)$, $W^t = (w_1^t, \ldots, w_n^t)$, $Z^t = (z^t, \ldots, z^t)$. and $H_t := \mathbb{E}H(X^t, W^t, Z^t, Y^t, W^{t-1}, Y^{t-1})$. Let $\delta_\beta \in (0, \frac{1}{2})$. Let $\beta \in (0, \frac{1}{L})$ be such that $(1 + \beta L)^2 - \frac{3}{2} + \frac{5}{2}\beta L < -\delta_\beta$. Let $\delta' \in [0, \delta_\beta)$. Let $\iota > 0$ and $\tau \in (0, 1)$ be small enough such that $\frac{1 - L\beta}{2}\tau^2 + (1 + \beta L)^2(2\iota + \iota^2) + (\beta L - 1)^2\iota < \delta'$. Denote $\delta := \delta_\beta - \delta'$. Suppose that $\epsilon_w$ is small enough such that $\left(\Gamma\frac{2}{(\frac{1}{\beta} - L)^2} + \frac{1}{\tau^2}\frac{1}{2(\frac{1}{\beta} - L)}\right)6CL^2\epsilon_w \le \frac{\delta - \delta_\epsilon}{\beta}$, for some $\delta_\epsilon > 0$, where $\Gamma := \frac{(1 + \iota)^2}{\beta\iota} + \frac{2}{\beta}\left(\frac{1}{\iota} + \beta L - 1\right)$ and $C := 2\left(\frac{(L_f + \frac{1}{\beta})^2}{\mu^2} + 1\right)\left(L_f + \frac{1}{\beta}\right)^2$.*

*Then, for $t \ge 1$,*

$$H_{t+1} \le H_t - \frac{\delta_\epsilon}{\beta}\|W^t - W^{t-1}\|^2. \tag{12}$$

**Remark 1.** *By letting $\delta_\beta = 1/4$, $\delta' = 1/8$, $\tau = 1/\sqrt{8}$, $\iota = 1/64$, $\delta_\epsilon = 1/16$, $\beta < \frac{-9 + \sqrt{82}}{L}$ and $\epsilon_w \le \frac{392}{96}\frac{(1 - \beta L)^2}{\beta^3}C^{-1}L^{-2}$, we have the conclusion in Theorem 1 with $H_{t+1} \le H_t - \frac{1}{16\beta}\|W^t - W^{t-1}\|^2$.*

Now we calculate the complexity of Algorithm 1.

**Theorem 2.** *Let assumptions in Theorem 1 hold. Let $\{(x_1^t, \ldots, x_n^t)\}, \{(y_1^t \ldots, y_n^t)\}, \{(w_1^t, \ldots, w_n^t)\}, \{z^t\}$ be generated by Algorithm 1. We further suppose $\epsilon_w$ and $\beta$ are small enough such that*

Table 2: Maximum AUC values obtained by each algorithm after 1000 communication rounds.

| Algorithm | a9a | covtype | gisette | ijcnn1 | phishing | w8a |
|---|---|---|---|---|---|---|
| CODASCA (Yuan et al., 2021b) | 0.8920 | 0.7967 | 0.9982 | 0.9264 | 0.9758 | 0.9007 |
| Fed-Norm-SGDA (Sharma et al., 2023) | 0.8961 | 0.7645 | 0.9961 | 0.9273 | 0.9786 | 0.8959 |
| FedSGDA (Wu et al., 2023) | 0.8963 | 0.7645 | 0.9962 | 0.9272 | 0.9786 | 0.8958 |
| FEDNEST (Tarzanagh et al., 2022) | 0.8963 | 0.8132 | 0.9989 | 0.9037 | 0.9714 | 0.9075 |
| FFMDR (This Work) | **0.8998** | **0.8208** | **0.9994** | **0.9288** | **0.9797** | **0.9076** |

$\frac{1}{2(\frac{1}{\beta}-L)}C\epsilon_w + 6L^2\sum_i p_i \le \frac{\delta}{\beta}$, where $C$ is defined in Theorem 1. Then it holds that

$$\frac{1}{T+1}\sum_{t=1}^{T+1}\mathbb{E}d^2(0, \nabla\sum_{i=1}^n F_i(z^t) + \partial g(z^t)) \le \frac{n}{\min_i p_i}\frac{1}{T+1}\left(D_1\bar{H}_0 + D_2\Upsilon_0 + D_3\|Y^0 - y(W^0)\|^2\right),$$

where $\bar{H}_0 := F(W^0) + \tilde{g}(Z^0) + \frac{1}{2\beta}\|X^0 - W^0\|^2 - \frac{1}{2\beta}\|X^0 - Z^0\|^2$, $D_1 := \frac{15L^2\beta}{\delta_\epsilon}$, $D_2 := 6\max\{1, L\}\epsilon_w + \frac{15L^2\beta}{\delta_\epsilon}C_u$, $D_3 := 3C_2 + \frac{15L^2\beta}{\delta_\epsilon}\frac{3}{2(\frac{1}{\beta}-L)}C\epsilon_w$, $C_u := 2\Gamma(\epsilon_w + 1) + \frac{\frac{1}{\beta}-L}{2}(\frac{1}{\tau^2} - 1)\epsilon_w + 6\max\{1, L\}\epsilon_w$ and $(X^0, Y^0, W^0, Z^0)$ are defined as in Theorem 1.

**Remark 2.** *This theorem indicates that the communication complexity of Algorithm 1 is $O(\kappa^2\epsilon^{-2})$. When the inner solver is chosen as SAGA, Theorem 2 together with Proposition 2 shows that the sample complexity of Algorithm 1 is $O(\kappa^2\log(\kappa)n^{-1}\epsilon^2)$.*

### 4.2 SEQUENTIAL CONVERGENCE OF ALGORITHM 1

In this section, we are devoted to analyze the convergence properties of the sequence generated by Algorithm 1 with equation 10. We make the following assumption.

**Assumption 3.** *Suppose for all $t$, equation 10 is deterministic and all clients attend the training at each round.*

**Theorem 3.** *Consider equation 1. Let $\{(X^t, W^t, Z^t, Y^t)\}$ as in Theorem 1. Suppose Assumption 3 holds. Suppose $F$ and $g$ are bounded from below and $g$ is level-bounded. Suppose in addition that $H$ is a KL function with exponent $\alpha \in [0, 1)$. Then $\{(X^t, W^t, Z^t, Y^t)\}$ is convergent. In addition, denoting $(X^*, W^*, Z^*, Y^*) := \lim_t(X^t, W^t, Z^t, Y^t)$, it holds that*

*(i) If $\alpha = 0$, then $\{(X^t, W^t, Z^t)\}$ converges finitely.*

*(ii) If $\alpha \in (0, \frac{1}{2}]$, then there exist $b > 0$, $t_1 \in \mathbb{N}$ and $\rho_1 \in (0, 1)$ such that $\max\{\|W^t - W^*\|, \|X^t - X^*\|, \|Z^t - Z^*\|, \|Y^t - Y^*\|\} \le b\rho_1^t$ for $t \ge t_1$.*

*(iii) If $\alpha \in (\frac{1}{2}, 1)$, then there exist $t_2 \in \mathbb{N}$ and $c > 0$ such that $\max\{\|W^t - W^*\|, \|X^t - Y^*\|, \|Z^t - Z^*\|, \|Y^t - Y^*\|\} \le ct^{-\frac{1}{4\alpha-2}}$ for $t \ge t_2$.*

Finally, we elaborate on how to verify the KL assumption in Theorem 3. Note that the KL assumption is on $H$ in equation 11. Since the $F$ in $H$ is a max function, $H$ can be viewed as a max function, i.e.,

$$H(X, W, Z, Y, W', Y') := \max_{Y''} U(X, W, Z, Y, W', Y', Y''),$$

where $Y'' := (y_1'', \ldots, y_n'')$ and

$$U(X, W, Z, Y, W', Y', W') := \frac{1}{n}\sum_{i=1}^n f_i(w_i, y_i'') + \tilde{g}(Z) + \frac{1}{2\beta}\left(\|X - W\|^2 - \|X - Z\|^2\right)$$

$$+ \frac{1}{\beta}\|W - Z\|^2 + \frac{\delta}{\beta}\|W - W'\|^2 + \frac{1}{12L^2}\sum_i p_i\|(y_i, w_i) - (y_i', w_i')\|^2.$$

Therefore, it is hard to directly verify the KL property of $H$. However, it is easier to verify the KL property of $U$. For example, when $U$ is a proper closed semi-algebraic function that has a closed

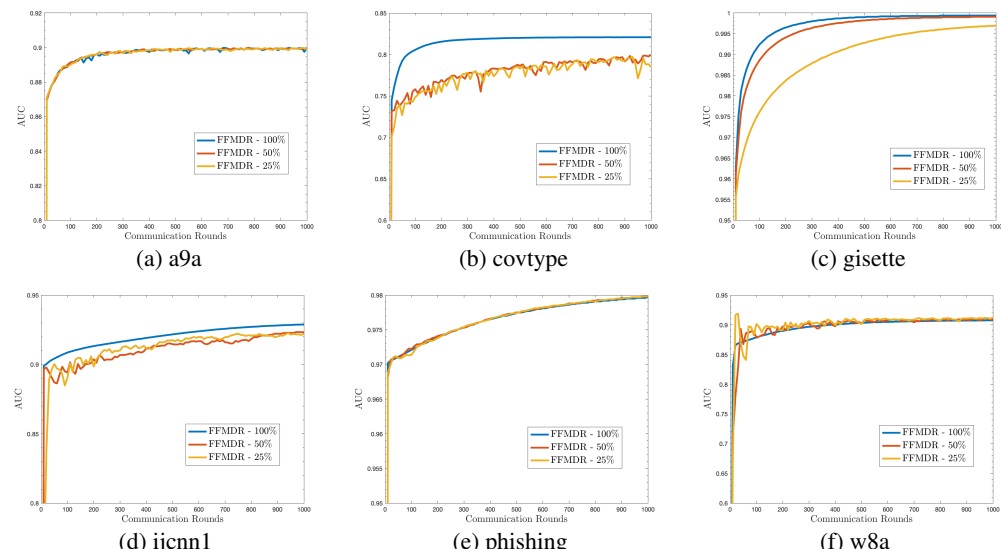

Figure 2: AUC values w.r.t. communication rounds on test dataset: a9a, covtype, gisette, ijcnn1, phishing and w8a.

domain and is continuous on their domains, $U$ is a KL function (Attouch et al., 2010). Given this fact, it is natural to ask whether we can deduce the KL property of a max function like $H$ from the KL property of the objective in the maximization like $U$. The following property provides a positive answer.

**Proposition 3.** *Let $f(x,y) : \mathbb{R}^m \times \mathbb{R}^n \to (-\infty, \infty)$ be a smooth function strongly concave in $y$ and $g : \mathbb{R}^m \to (-\infty, \infty)$ is a continuous function. Let $F(x,y) := f(x,y) + g(x)$. Suppose for any $y$, $F(\cdot, y)$ has the KL property at $x$ with exponent $\alpha \in [0,1)$ with constants $\epsilon(y)$, $c(y)$ and $a(y)$. Suppose $\epsilon(y)$, $c(y)$ and $a(y)$ are continuous in $y$. Let $G(x) = \max_y F(x,y)$. Let $x \in \mathrm{dom}\, \partial G$. Then $G$ has KL property at $x$ with exponent $\alpha$.*

**Remark 3.** *If we further use Theorem 3.3 in Li & Pong (2018), the KL exponent of $U$ can be deduced from that of $f(x,y) + g(x)$. A similar rule is investigated in Yu et al. (2022) where the authors address the infimum projection of a function, i.e., $h(x) := \inf_y f(x,y)$, while we address the max function $h(x) := \max_y f(x,y)$. The maximization is more challenging for preserving the KL exponent compared to the infimum projection. Here is a counterexample mentioned in Jiang & Li (2019). Suppose $H_{\inf}(x) = \min\{h_1(x) := x_1^2, h_2(x) := (x_1+1)^2 + x_2^2 - 1\}$. According to Theorem 3.1 in [2], the KL exponent of $H_{\inf}$ is $1/2$. However, if we consider the maximization $H_{\max} : \mathbb{R}^2 \to \mathbb{R}$ with $H_{\max}(x) = \max\{h_1(x) := x_1^2, h_2(x) := (x_1+1)^2 + x_2^2 - 1\}$, the following work shows that the KL exponent is $3/4$ when $h_1 = h_2$, even though the KL exponents of both $h_1$ and $h_2$ are $1/2$. Thus, the maximization requires more assumptions to preserve the KL exponent. In the minimax problem we consider, the objective function is strongly concave. In this case, we show that the KL exponent of the maximization function is preserved.*

**Remark 4.** *We provide an example where the assumptions in Proposition 3 is satisfied. For simplicity, we consider the following robust classification problem (Sinha et al., 2017):*

$$\min_\theta \max_\delta F(\theta, \delta) := \underbrace{\log(1 + \exp(-y\theta(x+\delta)))}_{\ell(\theta, \delta)} - c|\delta|^2 + \lambda|\theta|, \tag{13}$$

*where $(x,y) \in \mathbb{R} \times \{-1, 1\}$ is a data point, $\theta \in \mathbb{R}$ is the weight, $\delta$ is a perturbation and $c, \lambda > 0$ are scalers. Now fix any $\delta$. For any $\bar{\theta}$, there exists $\epsilon(\delta)$ continuous w.r.t. $\delta$ such that $F(\cdot, \delta)$ satisfies the KL property at $\bar{\theta}$ with exponent $\frac{1}{2}$ and constants $\epsilon(\delta)$, $c = 1$ and $a = 1$. More details can be found in the supplementary material.*

## 5 EXPERIMENTS

**Learning task** In this section, we apply our method to maximizing the Area under the ROC curve (AUC) problem (Natole et al., 2018) in the federated learning settings. This problem is formed as the following minimax problem:

$$\min_{\mathbf{w}\in\mathbb{R}^l, a\in\mathbb{R}, b\in\mathbb{R}} \max_{\alpha\in\mathbb{R}} \frac{1}{n}\sum_{i=1}^n \sum_{\eta\in\mathcal{D}_i} [f_i(\mathbf{w}, a, b, \alpha; \eta)] + g(\mathbf{w}), \tag{14}$$

where, $\eta = (x, y)$ is a datapoint, $n$ is the number of clients, $f_i(\mathbf{w}, a, b, \alpha; \eta) = p(1-p) + (1-p)(\mathbf{w}^T x - a)^2 \mathbb{I}_{[y=1]} + p(\mathbf{w}^T x - b)^2 \mathbb{I}_{[y=-1]} + 2(1+\alpha)\mathbf{w}^T x(p\mathbb{I}_{[y=-1]} - (1-p)\mathbb{I}_{[y=1]}) - p(1-p)\alpha^2$, $\mathbb{I}_A(x) = 1$ when $x \in A$ for any set $A$ and $\mathbb{I}_A(x) = 0$ otherwise. Here $p$ is the probability of $Pr(y = 1)$. The goal of AUC maximization tasks is to pursue a high AUC score for binary classification, which is defined by $Pr(\mathbf{w}^T x > \mathbf{w}^T x' | y = 1, y' = -1)$. This $F$ is an equivalent formulation and it is strongly concave in $\alpha$. The $g(\mathbf{w})$ in equation 14 is a convex regularization. In our experiments, we consider $g(\mathbf{w}) = \lambda \|\mathbf{w}\|_1$ where $\lambda = 0.001$ is fixed during the experiment. In our experiment, the total number of clients is set to 20.

**Dataset** We perform our experiments on six real-world dataset for binary classification: a9a, covtype, gisette, ijcnn1, phishing and w8a, all of which can be downloaded from the LIBSVM repository (Chang & Lin, 2011). The training data is distributed to all clients heterogeneously where each client only owns the data from one class.

**Compared methods** We compare our stochastic method with CODASCA in Yuan et al. (2021b), Fed-Norm-SGDA in Sharma et al. (2023) and FedSGDA in Wu et al. (2023). All these baselines are applicable to the AUC maximization problem in stochastic manner with a non-smooth regularization. CODASCA is an algorithm to solve federated AUC maximization problem for heterogeneous data. Other compared methods are general minimax algorithms which have been introduced in previous sections. In our experiments, the local solver of FFMDR is chosen as SGDA.

**Parameters** For FFMDR, we select the best value of $\frac{1}{2\beta}$ from $\{1, 0.1, 0.01, 0.001\}$, $\epsilon_w$ from $\{0.95, 0.75, 0.5, 0.25, 0.05\}$. For all methods, the stepsize is selected from $\{0.1, 0.01, 0.001, 0.0001, 0.00001\}$ so that it achieves the best experimental result. The batchsize is fixed to be 40. The local epoch is fixed to be 5.

**Results** In Figure 1, we plot the AUC values of each algorithm with respect to the number of communication rounds. In Table 2, we report detailed AUC scores obtained by each algorithm after 1000 communication rounds. From these experimental results we can see our FFMDR algorithm achieves the best AUC scores on all of the six datasets. Also, our method converges faster than the compared methods in most cases. These experimental results verify the performance of our proposed method to solve federated minimax problems with data heterogeneity.

Additionally, we also test our FFMDR method in the case where only a fraction of clients can participate in the training process in each communication round. The result is shown in Figure 2, where the percentage of clients attending the training in each round is $100\%/50\%/25\%$. Figure 2 indicates that in most cases, our FFMDR method with partial attendance of the clients also works as well as FFMDR with full attendance of clients.

## 6 CONCLUSION

In this paper, we proposed a new federated minimax method for nonconvex, strongly concave minimax problems. We demonstrated that our method has smaller sample complexity compared to existing federated minimax methods. More importantly, we showed the proposed method has global finite-step/linear/sublinear convergence guarantees for the updates of model parameters under KL assumption on novel potential function. We further made the KL exponent of the potential function easier to check by relating the maximizer-dependent potential function from that of the maximizer-free function. Empirically, our method is applied to the AUC maximization problem and consistently outperforms existing federated minimax methods in scenarios with high data heterogeneity.

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

## A  PROOF OF PROPOSITION 2

The minimax subproblem in Algorithm 1 for each selected client can be generalize to the following problem: Consider the general minimax problem

$$
\min_{w,y} r(w,y) := \frac{1}{l} \sum_{j=1}^{l} r_j(w,y;\xi_j)
$$

$$
= \sum_{j=1}^{l} \underbrace{\frac{1}{l} r_i(w,y;\xi_j) - \frac{1}{l}\frac{\lambda}{2}\|w\|^2 + \frac{1}{l}\frac{\gamma}{2}\|y\|^2}_{R_j(w,y;\xi_j)} + \underbrace{\frac{1}{l}\frac{\lambda}{2}\|w\|^2 - \frac{1}{l}\frac{\gamma}{2}\|y\|^2}_{s(w,y)},
\tag{15}
$$

where $\{\xi_1,\ldots,\xi_l\}$ is the dataset, $r$ is $\lambda$-strongly convex and $\gamma$-strongly concave. We consider the Algorithm 2 (SAGA) in (Palaniappan & Bach, 2016). For completeness, we let present Algorithm 2 for equation 15. The next proposition restate Proposition 2 and shows that equation 10 can be satisfied after finite iterates of Algorithm 2.

---

**Algorithm 2** SAGA for equation 15

---

1: Input: $(W, Y) \in \mathbb{R}^l \times \mathbb{R}^d$, $\varsigma > 0$. Mini-batch size $m$. $L > 0$ and $\bar{L} > 0$. Let $\sigma := \left(\max\{\frac{l}{m} - 1, L^2 + 3\frac{\bar{L}}{m}\}\right)^{-1}$

2: Compute $g^j = \nabla R_j(w,y;\xi_j)$ for $j = 1,\ldots,l$ and $G = \nabla \sum_{j=1}^{l} R_j(w,y;\xi_j)$

3: Let $k = 0$.

4: Uniformly sample a mini-batch $\{j_1,\ldots,j_m\} \subseteq \{1,\ldots,l\}$. Compute $h_i = \nabla R_{j_i}(w,y;\xi_{j_i})$ for $i \in \{1,\ldots,m\}$.
   Let

$$
(w,y) = \mathrm{Prox}_{\frac{1}{\sigma}s}\left((w,y) - \sigma \begin{bmatrix} \frac{1}{\lambda} & 0 \\ 0 & \frac{1}{\gamma} \end{bmatrix} \left(G + \frac{1}{m}\sum_{j=j_1}^{j_m}\left(lh_j - lg^{j_i}\right)\right)\right)
$$

5: Replace $G$ with $G - \frac{1}{m}\sum_{i=1}^{m}\left(g^{j_i} - h_j\right)$ and let $g^{j_i} = h_j$ for $i \in \{1,\ldots,m\}$

6: If a termination criterion is satisfied, terminate and output $(w,y)$. Else, let $k = k + 1$ and go to Step 3.

---

**Proposition 4.** *Apply Algorithm 2 to equation 15. Let $(\bar{w},\bar{y})$ satisfies $\nabla r(\bar{x},\bar{y}) \neq 0$. Let $\epsilon_w > 0$. Then, there exists $k = O(\max\{\frac{l}{m}, \log(\kappa)\})$ such that*

$$
\mathbb{E}\left\|(w,y) - (w_\star, y_\star)\right\|^2 \leq \epsilon_w \mathbb{E}\|(\bar{w},\bar{y}) - (w,y)\|^2.
$$

*Proof.* Since $r$ is strongly convex stronly concave, $\min_w \max_y r(w,y)$ has the unique solution $(x_\star, y_\star)$. Using Theorem 2 in Palaniappan & Bach (2016), there exist $\lambda = (\max\{\frac{3l}{2m}, 1 + \frac{L^2}{\min\{\lambda,\gamma\}^2} + \frac{3\bar{L}^2}{m\min\{\lambda,\gamma\}^2}\})^{-1} \in (0,1)$ such that

$$
\mathbb{E}\left\|(w^{k+1},y^{k+1}) - (w_\star, y_\star)\right\|^2 \leq (1-\lambda)^k \left\|(w^0,y^0) - (w_\star, y(w_\star))\right\|^2. \tag{16}
$$

Since $\nabla r(\bar{w},\bar{y}) \neq 0$, we know that $(\bar{w},\bar{y})$ is not the solution to $\min_y r(w, w(y))$. Thus, $\|(\bar{w},\bar{y}) - (w_\star, y_\star)\|^2 > 0$.

Since $a^2 \geq \frac{1}{2}(a+b)^2 - b^2$ for any vectors $a$ and $b$, it holds that

$$
\mathbb{E}\|(w^{k+1},y^{k+1}) - (\bar{x},\bar{y})\|^2 \geq \frac{1}{2}\|(w_\star, y_\star) - (\bar{x},\bar{y})\|^2 - \mathbb{E}\|(w^{k+1},y^{k+1}) - (x_\star, y_\star)\|^2
$$

$$
\geq \frac{1}{2}\|(w_\star, y_\star) - (\bar{x},\bar{y})\|^2 - (1-\lambda)^k\left\|(w^0,y^0) - (w_\star, y_\star)\right\|^2, \tag{17}
$$

where the second inequality uses equation 16.

Let $k \geq \log_{1-\lambda} \frac{\frac{1}{4}\|(w_\star, y_\star) - (\bar{x}, \bar{y})\|^2}{\|(w^0, y^0) - w_\star, y(w_\star))\|^2} = O(\max\{\frac{l}{m}, \log(\kappa)\})$ such that

$$2(1-\lambda)^k \left\|(w^0, y^0) - w_\star, y(w_\star))\right\|^2 \leq \frac{1}{2}\|(w_\star, y_\star) - (\bar{x}, \bar{y})\|^2.$$

Then equation 17 can be further passed to

$$\begin{aligned}
\mathbb{E}\|(w^{k+1}, y^{k+1}) - (\bar{x}, \bar{y})\|^2 &\geq (1-\lambda)^k \left\|(W^t, Y^t) - w_\star^{t+1}, y(w_\star^{t+1}))\right\|^2 \\
&\geq \mathbb{E}\left\|(w^{k+1}, y^{k+1}) - (w_\star, y_\star)\right\|^2.
\end{aligned} \tag{18}$$

Combining this with equation 18, and equation 16, we have that

$$\mathbb{E}\left\|(w^{k+1}, y^{k+1}) - (w_\star, y_\star)\right\|^2 \leq \mathbb{E}\|(w^{k+1}, y^{k+1}) - (\bar{x}, \bar{y})\|^2. \tag{19}$$

$\square$

# B   DETAILS FOR RESULTS IN SECTION 4.1

We first present the following useful fact.

**Fact 1.** *Let $f : \mathbb{R}^n \to \mathbb{R}$ be a strongly convex function with modulus $\mu$. Suppose in addition that $f$ is smooth and has Lipschitz continuous gradient with modulus $L$. Then there exists unique minimizers $x^*$ that minimize $f$ and it holds that*

$$\|\nabla f(x)\|^2 \geq 2\mu \left(f(x) - f(x^*)\right) \geq \mu^2 \|x - x^*\|^2. \tag{20}$$

We next present a proposition on $\Upsilon_{i,t+1}$.

**Proposition 5.** *Suppose Assumptions 1 and 2 hold. Assume $\frac{1}{\beta} > L$, where $L$ is the one defined as in Proposition 1. Suppose $12 \max\{1, L\}\epsilon_w \leq \frac{1}{4}$. Assume that $\Upsilon_{i,0} > \|y_i^0 - y_i(w_{i,\star}^0)\|^2 + \|w_{i,\star}^0 - w_i^0\|^2$, where $w_{i,\star}^0 := \min_{w_i} f(w_i, y_i(w_i)) + \frac{1}{2\beta}\|w_i - x_i^0\|^2$. Then,*

*(i) For $t \geq 0$,*

$$\sum_i p_i \mathbb{E}\Upsilon_{i,t+1} \leq \frac{1}{2}\left(\sum_i p_i \mathbb{E}\Upsilon_{i,t} - \sum_i p_i \mathbb{E}\Upsilon_{i,t+1}\right) + 6L^2 \sum_i p_i \mathbb{E}\|w_i^t - w_i^{t+1}\|^2. \tag{21}$$

*(ii) When we choose the deterministic case. It holds that*

$$\sum_i p_i \mathbb{E}\|\nabla r_{i,t+1}(w_i^{t+1}, y_i(w_i^{t+1}))\|^2 \leq C\epsilon_w \sum_i p_i \mathbb{E}\Upsilon_{i,t+1}, \tag{22}$$

*where $C := 2\left(\frac{(L_f + \frac{1}{\beta})^2}{\mu^2} + 1\right)\left(L_f + \frac{1}{\beta}\right)^2$.*

*Proof.* For (i), note that for $t \geq 0$, it holds that

$$\begin{aligned}
&\|(w_i^t, y_i^t) - (w_i^{t+1}, y_i^{t+1})\|^2 \\
&\leq 3\|(w_i^t, y_i^t) - (w_i^t, y_i(w_i^t))\|^2 + 3\|(w_i^t, y_i(w_i^t)) - (w_i^{t+1}, y_i(w_i^{t+1}))\|^2 \\
&\quad + 3\|(w_i^{t+1}, y_i(w_i^{t+1})) - (w_i^{t+1}, y_i^{t+1})\|^2 \\
&= 3\|y_i^t - y_i(w_i^t)\|^2 + 3\|(w_i^t, y_i(w_i^t)) - (w_i^{t+1}, y_i(w_i^{t+1}))\|^2 + 3\|y_i(w_i^{t+1}) - y_i^{t+1}\|^2 \\
&\leq 3\|y_i^t - y_i(w_i^t)\|^2 + 3L^2\|w_i^t - w_i^{t+1}\|^2 + 3\|y_i(w_i^{t+1}) - y_i^{t+1}\|^2
\end{aligned} \tag{23}$$

where the second inequality uses Proposition 1. In addition, under the assumption that $\Upsilon_{i,0} \geq \|y_i^0 - y_i(w_{i,\star}^0)\|^2 + \|w_{i,\star}^0 - w_i^0\|^2$, for $t \geq 0$, it holds that for $i \in \mathcal{S}^{t-1}$,

$$\begin{aligned}
\mathbb{E}_{t-1}\|y_i^t - y_i(w_i^t)\|^2 &\leq 2\mathbb{E}_{t-1}\|Y_i^t - y(w_{i,\star}^t)\|^2 + 2\mathbb{E}_{t-1}\|y(w_{i,\star}^t) - y_i(w_i^t)\|^2 \\
&\leq 2\max\{1, L\}\mathbb{E}_{t-1}\left(\|Y_i^t - y_i(w_{i,\star}^t)\|^2 + \|w_{i,\star}^t - w_i^t\|^2\right) \\
&\leq 2\max\{1, L\}\epsilon_w \mathbb{E}_{t-1}\Upsilon_{i,t},
\end{aligned} \tag{24}$$

where the second inequality is thanks to equation 10. Taking expectation with respect to $\mathcal{S}^{t-1}$, the above inequality becomes

$$\sum_i p_i \mathbb{E}_{t-1} \|y_i^t - y_i(w_i^t)\|^2 = \mathbb{E}_{\mathcal{S}^{t-1}} \mathbb{E}_{t-1} \|y_i^t - y_i(w_i^t)\|^2$$

$$\leq 2\max\{1, L\}\epsilon_w \mathbb{E}_{\mathcal{S}^{t-1}} \mathbb{E}_{t-1} \Upsilon_{i,t} = 2\max\{1, L\}\epsilon_w \sum_i p_i \mathbb{E}_{t-1} \Upsilon_{i,t}, \tag{25}$$

Taking expectation with respect to $\mathcal{Y}^{t-1} = \{\mathcal{S}^0, \ldots, \mathcal{S}^{t-2}, (x^1, Y^1, W^1), \ldots, (x^{t-1}, Y^{t-1}, W^{t-1})\}$, we have

$$\sum_i p_i \mathbb{E} \|y_i^t - y_i(w_i^t)\|^2 \leq 2\max\{1, L\}\epsilon_w \sum_i p_i \mathbb{E} \Upsilon_{i,t}, \tag{26}$$

Similarly, for $t \geq 0$, it holds that

$$\sum_i p_i \mathbb{E}\|y_i^{t+1} - y_i(w_i^{t+1})\|^2 \leq 2\sum_i p_i \mathbb{E}\|y_i^{t+1} - y_i(w_{i,\star}^{t+1})\|^2 + 2\sum_i p_i \mathbb{E}\|y_i(w_{i,\star}^{t+1}) - y_i(w_i^{t+1})\|^2$$

$$\leq 2\max\{1, L\} \sum_i p_i \mathbb{E}\left(\|y_i^{t+1} - y_i(w_{i,\star}^{t+1})\|^2 + \|w_\star^{t+1} - w_i^{t+1}\|^2\right)$$

$$\leq 2\max\{1, L\}\epsilon_w \sum_i p_i \mathbb{E} \Upsilon_{i,t+1}. \tag{27}$$

Combining equation 23, equation 24 and equation 27, it holds that

$$\sum_i p_i \mathbb{E} \Upsilon_{i,t+1}$$

$$\leq 3\left(2\max\{1, L\}\epsilon_w \sum_i p_i \mathbb{E} \Upsilon_{i,t}\right) + 6\max\{1, L\}\epsilon_w \sum_i p_i \mathbb{E} \Upsilon_{i,t+1} + 3L^2 \sum_i p_i \mathbb{E}_t \|w_i^t - w_i^{t+1}\|^2.$$

Since $\epsilon_w$ is small enough such that $6\max\{1, L\}\epsilon_w \leq 6\max\{1, L\}\epsilon_w \leq \frac{1}{5}$, rearranging the above inequality and recalling the definition of $\Upsilon_{i,t+1}$, we have that

$$\sum_i p_i \mathbb{E} \Upsilon_{i,t+1} \leq \frac{1}{2} \sum_i p_i \left(\mathbb{E} \Upsilon_{i,t} - \mathbb{E} \Upsilon_{i,t+1}\right) + 6L^2 \sum_i p_i \mathbb{E}_t \|w_i^t - w_i^{t+1}\|^2.$$

For (ii), note that for $i \in \mathcal{S}^t$,

$$\|\nabla r_{i,t+1}(w_i^{t+1}, y_i(w_i^{t+1}))\|^2$$

$$\leq 2\|\nabla r_{i,t+1}(w_i^{t+1}, y_i(w_i^{t+1})) - \nabla r_{i,t+1}(w_i^{t+1}, y_i^{t+1})\|^2 + 2\|\nabla r_{i,t+1}(w_i^{t+1}, y_i^{t+1})\|^2 \tag{28}$$

$$\leq 2(L_f + \frac{1}{\beta})^2 \|y_i(w_i^{t+1}) - y_i^{t+1}\|^2 + 2\|\nabla r_{i,t+1}(w_i^{t+1}, y_i^{t+1})\|^2,$$

where the second inequality is because $r_{i,t+1}$ is Lipschitz continuous with modulus $L_f + \frac{1}{\beta}$. In addition, since $\nabla r_{i,t+1}(w_{i,\star^{t+1}}, y_{i,\star^{t+1}})$ is the solution of $\min_{w_i} \max_{y_i} r_{i,t+1}(y_i, w_i)$, it holds that for $i \in \mathcal{S}^t$,

$$\|\nabla r_{i,t+1}(w_i^{k+1}, y_i^{k+1})\|^2 = \|\nabla r_{i,t+1}(w_i^{k+1}, y_i^{k+1}) - \nabla r_{i,t+1}(w_{i,\star^{t+1}}, y_i(w_{i,\star^{t+1}}))\|^2$$

$$\leq \left(L_f + \frac{1}{\beta}\right)^2 \left\|(w_i^{k+1}, y_i^{k+1}) - (w_{i,\star^{t+1}}, y_i(w_{i,\star^{t+1}}))\right\|^2, \tag{29}$$

where the second inequality is because $r$ is Lipschitz continuous with modulus $L_f + \frac{1}{\beta}$. Combining equation 28 and equation 29, we have that for $i \in \mathcal{S}^t$,

$$
\|\nabla r_{i,t+1}(w_i^{t+1}, y_i(w_i^{t+1}))\|^2
$$

$$
\leq 2(L_f + \frac{1}{\beta})^2 \|y_i(w_i^{t+1}) - y_i^{t+1}\|^2 + 2\left(L_f + \frac{1}{\beta}\right)^2 \left\|(w_i^{k+1}, y_i^{k+1}) - (w_{i,\star^{t+1}}, y_{i,\star^{t+1}})\right\|^2
$$

$$
\leq 2\frac{(L_f + \frac{1}{\beta})^2}{\mu^2} \|\nabla_w f_i(w_i^{t+1}, y_i^{t+1})\|^2 + 2\left(L_f + \frac{1}{\beta}\right)^2 \left\|(w_i^{k+1}, y_i^{k+1}) - (w_{i,\star^{t+1}}, y_{i,\star^{t+1}})\right\|^2
$$

$$
\leq 2\left(\frac{(L_f + \frac{1}{\beta})^2}{\mu^2} + 1\right)\left(L_f + \frac{1}{\beta}\right)^2 \left\|(w_i^{k+1}, y_i^{k+1}) - (w_{i,\star^{t+1}}, y_{i,\star^{t+1}})\right\|^2
$$

where the second inequality is because $y_i(w_i^{t+1})$ is the minimizer of $\min_w -r_{i,t+1}(y, w)$ and the fact that $-r_{i,t+1}(y, w)$ is strongly convex with modulus $\mu$ and Proposition 1, the last inequality uses equation 29. Combining the above inequality with equation 10, taking the expectation on $\mathcal{S}^t$ and taking the expectation on $\mathcal{Y}^t$, we reach the conclusion (ii).

$\square$

Before prove Theorem 1, we need the following lemma.

**Lemma 1.** *Let*

$$
e_i^{t+1} := w_i^{t+1} - w_{i,\star}^{t+1}. \tag{30}
$$

*Suppose $\beta < L$, where $L$ defined in Proposition 1. Assume $w_i^0 = \mathrm{Prox}_{\beta f_i}(x_i^0, y_i(x_i^0))$. Then exists $\eta^{t+1} \in \partial \tilde{g}(Z^{t+1})$ such that the following relations hold:*

*(i) for all $i$,*

$$
0 \in \nabla f_i(\cdot, y_i(\cdot))(w_{i,\star}^{t+1}) + \frac{1}{\beta}(w_{i,\star}^{t+1} - x_i^{t+1}) \tag{31}
$$

*and*

$$
\tilde{z}_i^{t+1} = 2w_i^{t+1} - x_i^{t+1}. \tag{32}
$$

*For $i \in \mathcal{S}^t$,*

$$
-\frac{1}{\beta}(w_{i,\star}^{t+1} - x_i^{t+1}) = \nabla f_i(\cdot, y_i(\cdot))(w_{i,\star}^{t+1})
$$

$$
\Leftrightarrow -\frac{1}{\beta}(w_i^{t+1} - e_i^{t+1} - x_i^{t+1}) = \nabla f_i(\cdot, y_i(\cdot))(w_{i,\star}^{t+1}) \tag{33}
$$

*(ii)*

$$
\eta^{t+1} = \frac{1}{\beta}(2W^{t+1} - X^{t+1}) - Z^{t+1}. \tag{34}
$$

*Proof.* We prove (i) by induction. For $t = 0$, we have by assumption that $w_i^0 = \mathrm{Prox}_{\beta f_i}(x_i^0, y_i(x_i^0))$. Then $x_i^0 = w_i^0 + \nabla f_i(\cdot, y_i(\cdot))(w_{i,\star}^0)$, and $\tilde{z}_i^0 = 2w_i^0 - x_i^0$. Now suppose equation 33 and equation 32 holds at iteration $t$. For iteration $t+1$, when $i \in \mathcal{S}^t$, equation 33 follows from the firs-order optimality condition of the subproblem in equation 10. When $i \notin \mathcal{S}^t$, since $x_i^{t+1} = x_i^t$, by induction, we have that

$$
\nabla f_i(\cdot, y_i(\cdot))(w_{i,\star}^{t+1}) + \frac{1}{\beta}(w_{i,\star}^{t+1} - x_i^{t+1}) = \nabla f_i(\cdot, y_i(\cdot))(w_{i,\star}^t) + \frac{1}{\beta}(w_{i,\star}^t - x_i^t) = 0.
$$

In addition, for $i \notin \mathcal{S}^t$, we have $\tilde{z}_i^{t+1} = \tilde{z}_i^t = 2w_i^t - x_i^t = 2w_i^{t+1} - x_i^{t+1}$.

equation 34 follows from (i), Excercise 8.8 of Rockafellar & Wets (1998) and the firs-order optimality condition of the subproblem in equation 3. $\square$

Next, we show the detailed version of Theorem 1 and its proof.

**Theorem 4.** *Consider equation 1. Suppose the conditions in Proposition 5 hold. Apply Algorithm 1 to equation 1. Let $\{(x_i^{t+1}, w_i^{t+1}, y_i^t, z^{t+1})\}$ be defined as in Algorithm 1. Define $X^t = (x_1^t, \ldots, x_n^t)$, $Y^t = (y_1^t, \ldots, y_n^t)$, $W^t = (w_1^t, \ldots, w_n^t)$ and $Z^t = (z^t, \ldots, z^t)$. Let $\delta_\beta \in (0, \frac{1}{2})$. Let $\beta \in (0, \frac{1}{L})$ be such that*

$$(1 + \beta L)^2 - \frac{3}{2} + \frac{5}{2}\beta L < -\delta_\beta. \tag{35}$$

*Let $\delta' \in [0, \delta_\beta)$. Let $\iota > 0$ and $\tau \in (0, 1)$ be small enough such that*

$$\frac{1 - L\beta}{2}\tau^2 + (1 + \beta L)^2(2\iota + \iota^2) + (\beta L - 1)^2\iota < \delta'. \tag{36}$$

*Denote $\delta := \delta_\beta - \delta'$. Suppose that $\epsilon_w$ is small enough such that*

$$\left( \Gamma \frac{2}{(\frac{1}{\beta} - L)^2} + \frac{1}{\tau^2} \frac{1}{2(\frac{1}{\beta} - L)} \right) 6CL^2\epsilon_w \leq \frac{\delta - \delta_\epsilon}{\beta},$$

*for some $\delta_\epsilon > 0$, where $\Gamma := \frac{(1+\iota)^2}{\beta\iota} + \frac{2}{\beta}\left(\frac{1}{\iota} + \beta L - 1\right)$ and $C$ is defined as in Proposition 5. Then the following statements hold:*

*(i) Let $e_i^{t+1}$ be defined as in equation 30. It holds that*

$$\sum_i p_i \mathbb{E}\|e_i^{t+1}\|^2 \leq \frac{1}{(\frac{1}{\beta} - L)^2}\left( C\epsilon_w \sum_i p_i \mathbb{E}\Upsilon_{i,t+1} \right), \tag{37}$$

*where $C$ is defined in Proposition 5.*

*(ii) It holds that,*

$$\sum_i p_i \mathbb{E}\|x_i^{t+1} - x_i^t\|^2$$

$$\leq (1 + \beta L)^2 \left( 1 + \iota + (1 + \beta L)^2 \left(1 + \frac{1}{\iota}\right) \frac{2}{(\frac{1}{\beta} - L)^2} C\epsilon_w \right) \sum_i p_i \mathbb{E}\Upsilon_{i,t+1} \tag{38}$$

$$+ (1 + \beta L)^2 \left(1 + \frac{1}{\iota}\right) \left( \frac{2}{(\frac{1}{\beta} - L)^2} C\epsilon_w \sum_i p_i \mathbb{E}\Upsilon_{i,t} \right).$$

*(iii) Define*

$$H(X, W, Z, Y, W', Y')$$
$$:= F(W) + \tilde{g}(Z) + \frac{1}{2\beta}\left( \|X - W\|^2 - \|X - Z\|^2 \right) + \frac{1}{\beta}\|W - Z\|^2 \tag{39}$$
$$+ \frac{\delta}{\beta}\|W - W'\|^2 + \frac{1}{12L^2}\sum_i p_i \|(y_i, w_i) - (y_i', w_i')\|^2.$$

*where $\tilde{g}$ is defined in equation 4. It holds that for $t \geq 1$,*

$$\mathbb{E}H(X^{t+1}, W^{t+1}, Z^{t+1}, Y^{t+1}, W^t, Y^t)$$
$$\leq \mathbb{E}H(X^t, W^t, Z^t, Y^t, W^{t-1}, Y^{t-1}) - \frac{\delta_\epsilon}{\beta}\sum_i p_i \mathbb{E}\|w_i^t - w_i^{t-1}\|^2 \tag{40}$$
$$- \frac{1}{2\beta}\sum_i p_i \mathbb{E}\|z_i^{t+1} - z_i^t\|^2.$$

*Proof.* For (i), note that $r_{i,t+1}(w_i, y_i(w_i))$ is strongly convex with modulus $\frac{1}{\beta} - L$, using the definition of $e_i^{t+1}$, it holds that

$$\sum_i p_i \mathbb{E} \|e_i^{t+1}\|^2 = \mathbb{E}_{\mathcal{S}^t} \sum_{i \in \mathcal{S}^t} \mathbb{E}_{\mathcal{Y}^t} \mathbb{E}_t \|e_i^{t+1}\|^2 = \mathbb{E}_{\mathcal{S}^t} \sum_{i \in \mathcal{S}^t} \mathbb{E}_{\mathcal{Y}^t} \mathbb{E}_t \|w_i^{t+1} - w_{i,\star}^{t+1}\|^2$$

$$\leq \frac{1}{(\frac{1}{\beta} - L)^2} \mathbb{E}_{\mathcal{S}^t} \sum_{i \in \mathcal{S}^t} \mathbb{E}_{\mathcal{Y}^t} \mathbb{E}_t \|\nabla r_{i,t+1}(\cdot, y_i(\cdot))(w_i^{t+1})\|^2$$

$$= \frac{1}{(\frac{1}{\beta} - L)^2} \mathbb{E}_{\mathcal{S}^t} \sum_{i \in \mathcal{S}^t} \mathbb{E}_{\mathcal{Y}^t} \mathbb{E}_t \|\nabla_y r_{i,t+1}(w_i^{t+1}, y_i(w_i^{t+1}))\|^2$$

$$= \frac{1}{(\frac{1}{\beta} - L)^2} \sum_i p_i \mathbb{E}_{\mathcal{Y}^t} \mathbb{E}_t \|\nabla_y r_{i,t+1}(w_i^{t+1}, y_i(w_i^{t+1}))\|^2$$

$$\leq \frac{1}{(\frac{1}{\beta} - L)^2} \left( C\epsilon_w \sum_i p_i \mathbb{E} \Upsilon_{i,t+1} \right),$$

where the first inequality uses equation 20, the second equality uses the last inequality uses equation 22. Taking expectation on $\mathcal{Y}^t$, we obtain equation 37.

For (ii), using equation 33, we have that

$$\sum_i p_i \mathbb{E} \|x_i^{t+1} - x_i^t\|^2 = \mathbb{E}_{\mathcal{S}^t} \sum_{i \in \mathcal{S}^t} \mathbb{E}_{\mathcal{Y}^t} \mathbb{E}_t \|x_i^{t+1} - x_i^t\|^2$$

$$\leq (1 + \beta L)^2 \mathbb{E}_{\mathcal{S}^t} \sum_{i \in \mathcal{S}^t} \mathbb{E}_{\mathcal{Y}^t} \mathbb{E}_t \|w_{i,\star}^{t+1} - w_{i,\star}^t\|^2$$

$$\leq (1 + \beta L)^2 \left( (1 + \iota) \mathbb{E}_{\mathcal{S}^t} \sum_{i \in \mathcal{S}^t} \mathbb{E}_{\mathcal{Y}^t} \mathbb{E}_t \|w_i^{t+1} - w_i^t\|^2 + \left( 1 + \frac{1}{\iota} \right) \mathbb{E}_{\mathcal{S}^t} \sum_{i \in \mathcal{S}^t} \mathbb{E}_{\mathcal{Y}^t} \mathbb{E}_t \| - e_i^{t+1} - e_i^t\|^2 \right)$$

$$= (1 + \beta L)^2 \left( (1 + \iota) \sum_i p_i \|w_i^{t+1} - w_i^t\|^2 + \left( 1 + \frac{1}{\iota} \right) \sum_i p_i \mathbb{E} \| - e_i^{t+1} - e_i^t\|^2 \right), \tag{41}$$

where the second inequality uses the Young's inequality. Noting that thanks to equation 10, we have that

$$\sum_i p_i \mathbb{E} \| - e_i^{t+1} - e_i^t\|^2 = \mathbb{E}_{\mathcal{S}^t} \sum_{i \in \mathcal{S}^t} \mathbb{E}_{\mathcal{Y}^t} \mathbb{E}_t \| - e_i^{t+1} - e_i^t\|^2$$

$$\leq 2 \mathbb{E}_{\mathcal{S}^t} \sum_{i \in \mathcal{S}^t} \mathbb{E}_{\mathcal{Y}^t} \mathbb{E}_t \|e_i^{t+1}\|^2 + 2 \mathbb{E} \|e_i^t\|^2$$

$$\leq \frac{2}{(\frac{1}{\beta} - L)^2} C\epsilon_w \mathbb{E}_{\mathcal{S}^t} \sum_{i \in \mathcal{S}^t} \mathbb{E}_{\mathcal{Y}^t} \left( \mathbb{E}_t \Upsilon_{i,t} + \mathbb{E}_t \Upsilon_{i,t+1} \right) \tag{42}$$

$$= \frac{2}{(\frac{1}{\beta} - L)^2} C\epsilon_w \sum_i p_i \left( \mathbb{E} \Upsilon_{i,t} + \mathbb{E} \Upsilon_{i,t+1} \right),$$

where the last inequality is because of equation 37.

Combining this with equation 41 we have that

$$\sum_i p_i \mathbb{E}\|x_i^{t+1} - x_i^t\|^2$$

$$\leq (1+\beta L)^2 (1+\iota) \sum_i p_i \mathbb{E}\|w_i^{t+1} - w_i^t\|^2$$

$$+ (1+\beta L)^2 \left(1+\frac{1}{\iota}\right) \left(\frac{2}{(\frac{1}{\beta}-L)^2} C\epsilon_w \sum_i p_i \left(\mathbb{E}\Upsilon_{i,t} + \mathbb{E}\Upsilon_{i,t+1}\right)\right)$$

$$\leq (1+\beta L)^2 (1+\iota) \sum_i p_i \mathbb{E}\Upsilon_{i,t+1}$$

$$+ (1+\beta L)^2 \left(1+\frac{1}{\iota}\right) \left(\frac{2}{(\frac{1}{\beta}-L)^2} C\epsilon_w \sum_i p_i \left(\mathbb{E}\Upsilon_{i,t} + \mathbb{E}\Upsilon_{i,t+1}\right)\right).$$

Now we prove (iii). Denote

$$\bar{H}(X, W, Z) := F(W) + \tilde{g}(Z) + \frac{1}{2\beta}\left(\|X-W\|^2 - \|X-Z\|^2\right). \tag{43}$$

Note that

$$\bar{H}(X^{t+1}, W^t, Z^t) - \bar{H}(X^t, W^t, Z^t)$$

$$= \frac{1}{2\beta}\left(\|X^{t+1}-W^t\|^2 - \|X^{t+1}-Z^t\|^2\right) - \frac{1}{2\beta}\left(\|X^t-W^t\|^2 - \|X^t-Z^t\|^2\right)$$

$$= -\frac{1}{\beta}\left\langle X^{t+1}-X^t, W^t-Z^t\right\rangle$$

$$\stackrel{(a)}{=} \frac{1}{\beta}\|X^{t+1}-X^t\|^2 = \frac{1}{\beta}\sum_{i\in\mathcal{S}^t}\|x_i^{t+1}-x_i^t\|^2.$$

where (a) uses equation 2 and the last in equality is because $X^{t+1} = X^t$ for $i \notin \mathcal{S}^t$.

Taking expectation on $\mathcal{S}^t$ and then on $\mathcal{Y}^t$, the above inequality becomes

$$\mathbb{E}\bar{H}(X^{t+1}, W^t, Z^t) - \mathbb{E}\bar{H}(X^t, W^t, Z^t) = \frac{1}{\beta}\sum_i p_i \mathbb{E}\|x_i^{t+1}-x_i^t\|^2. \tag{44}$$

Note that $w_{i,\star}^{t+1}$ in Step 3 of Algorithm 1 is the minimizer of $\min_y r_{i,t+1}(w_i, y_i(w_i))$, where $r_{i,t}$ is defined in Algorithm 1. Since $\beta < \frac{1}{L}$, the objective $\tilde{F}(W)$ is strongly convex with modulus $\frac{1}{\beta} - L$. Thus, using equation 20, we have that for $i \in \mathcal{S}^t$,

$$\mathbb{E}_t r_{i,t+1}(w_i^{t+1}, y_i(w_i^{t+1}))$$

$$\leq \mathbb{E}_t r_{i,t+1}(w_{i,\star}^{t+1}, y_i(w_{i,\star}^{t+1})) + \frac{1}{2(\frac{1}{\beta}-L)}\mathbb{E}_t\|\nabla_y r(w_i^{t+1}, y_i(w_i^{t+1}))\|^2$$

$$\leq \mathbb{E} r_{i,t+1}(w_{i,\star}^{t+1}, y_i(w_{i,\star}^{t+1})) + \frac{1}{2(\frac{1}{\beta}-L)}\left(C\epsilon\mathbb{E}_t\Upsilon_{i,t+1}\right),$$

where the last inequality is due to equation 10, the second equality uses the last inequality uses equation 22. Using the above inequality, we have that

$$
\mathbb{E}_t \bar{H}(X^{t+1}, W^{t+1}, Z^t) - \mathbb{E}_t \bar{H}(X^{t+1}, W^t, Z^t)
$$

$$
= \sum_{i=1}^{n} \mathbb{E}_t r_{i,t+1}(w_i^{t+1}, y_i(w_i^{t+1})) - F(W^t) - \frac{1}{2\beta} \mathbb{E}_t \|X^{t+1} - W^t\|^2
$$

$$
\leq \sum_{i \in \mathcal{S}^t} \mathbb{E}_t r_{i,t+1}(w_{i,\star}^{t+1}, y_i(w_{i,\star}^{t+1})) + \sum_{i \in \mathcal{S}^t} \frac{1}{2(\frac{1}{\beta} - L)} C \epsilon_w \Upsilon_{i,t+1} - \mathbb{E} F(W^t)
$$

$$
- \frac{1}{2\beta} \mathbb{E}_t \|X^{t+1} - W^t\|^2
\tag{45}
$$

$$
\leq \sum_{i \in \mathcal{S}^t} \mathbb{E} r_{i,t+1}(w_i^t, y_i(w_i^t)) - \frac{\frac{1}{\beta} - L}{2} \|w_i^t - w_{i,\star}^{t+1}\|^2 - \mathbb{E} F(W^t) - \frac{1}{2\beta} \mathbb{E}_t \|X^{t+1} - W^t\|^2
$$

$$
+ \sum_{i \in \mathcal{S}^t} \frac{1}{2(\frac{1}{\beta} - L)} C \epsilon_w \mathbb{E}_t \Upsilon_{i,t+1}
$$

$$
= - \sum_{i \in \mathcal{S}^t} \frac{\frac{1}{\beta} - L}{2} \mathbb{E}_t \|w_{i,\star}^{t+1} - w_i^t\|^2 + \sum_{i \in \mathcal{S}^t} \frac{1}{2(\frac{1}{\beta} - L)} C \epsilon_w \mathbb{E}_t \Upsilon_{i,t+1}.
$$

Note that

$$
\|w_{i,\star}^{t+1} - w_i^t\|^2 = \|w_{i,\star}^{t+1} - w_i^{t+1}\|^2 + 2 \langle w_{i,\star}^{t+1} - w_i^{t+1}, w_i^{t+1} - w_i^t \rangle + \|w_i^{t+1} - w_i^t\|^2
$$

$$
\geq \|w_{i,\star}^{t+1} - w_i^{t+1}\|^2 - \left( \frac{1}{\tau^2} \|w_{i,\star}^{t+1} - w_i^{t+1}\|^2 + \tau^2 \|w_i^{t+1} - w_i^t\|^2 \right) + \|w_i^{t+1} - w_i^t\|^2
$$

$$
= (1 - \frac{1}{\tau^2}) \|w_{i,\star}^{t+1} - w_i^{t+1}\|^2 + (1 - \tau^2) \|w_i^{t+1} - w_i^t\|^2,
$$

where $\tau \in (0, 1)$ by assumption. Using this, equation 45 can be further passed to

$$
\mathbb{E}_t \bar{H}(X^{t+1}, W^{t+1}, Z^t) - \mathbb{E}_t \bar{H}(X^{t+1}, W^t, Z^t)
$$

$$
\leq - \sum_{i \in \mathcal{S}^t} \frac{\frac{1}{\beta} - L}{2} (1 - \tau^2) \mathbb{E}_t \|w_i^{t+1} - w_i^t\|^2 + \sum_{i \in \mathcal{S}^t} \frac{\frac{1}{\beta} - L}{2} (\frac{1}{\tau^2} - 1) \mathbb{E}_t \|w_{i,\star}^{t+1} - w_i^{t+1}\|^2
$$

$$
+ \sum_{i \in \mathcal{S}^t} \frac{1}{2(\frac{1}{\beta} - L)} (C \epsilon_w \mathbb{E}_t \Upsilon_{i,t+1})
$$

Taking expectation on $\mathcal{S}^t$ and then on $\mathcal{Y}^t$, the above inequality becomes

$$
\mathbb{E} \bar{H}(X^{t+1}, W^{t+1}, Z^t) - \mathbb{E} \bar{H}(X^{t+1}, W^t, Z^t)
$$

$$
\leq - \sum_i p_i \frac{\frac{1}{\beta} - L}{2} (1 - \tau^2) \mathbb{E} \|w_i^{t+1} - w_i^t\|^2 + \frac{\frac{1}{\beta} - L}{2} (\frac{1}{\tau^2} - 1) \sum_i p_i \mathbb{E} \|w_{i,\star}^{t+1} - w_i^{t+1}\|^2
\tag{46}
$$

$$
+ \sum_i p_i \frac{1}{2(\frac{1}{\beta} - L)} (C \epsilon_w \mathbb{E} \Upsilon_{i,t+1})
$$

On the other hand, note that

$$
\bar{H}(X, W, Z)
$$

$$
= F(W) + \tilde{g}(Z) + \frac{1}{2\beta} \left( \|X - W\|^2 - (\|X - W\|^2 - 2 \langle X - W, Z - W \rangle + \|W - Z\|^2) \right)
$$

$$
= F(W) + \tilde{g}(Z)
$$

$$
+ \frac{1}{2\beta} \left( \|X - W\|^2 - (\|X - W\|^2 - \|X - Z - 2W\|^2 + \|X - W\|^2 + \|Z - W\|^2 + \|W - Z\|^2) \right)
$$

$$
= F(W) + \tilde{g}(Z) + \frac{1}{2\beta} \|X - Z - 2y\|^2 + \frac{1}{2\beta} \left( -\|X - W\|^2 - 2\|W - Z\|^2 \right)
\tag{47}
$$

In addition, note that $Z^{t+1}$ is the minimizer of $\min \tilde{g}(Z) + \frac{1}{2\beta}\|2W^{t+1} - X^{t+1} - Z\|^2$, whose objective is strongly convex with modulus $\frac{1}{\beta}$. Using this fact together with equation 47, we have that

$$
\begin{aligned}
&\bar{H}(X^{t+1}, W^{t+1}, Z^{t+1}) - \bar{H}(X^{t+1}, W^{t+1}, Z^t) \\
&= \left( g(Z^{t+1}) + \frac{1}{2\beta}\|X^{t+1} - Z^{t+1} - 2W^{t+1}\|^2 - \frac{1}{\beta}\|W^{t+1} - Z^{t+1}\|^2 \right) \\
&\quad - \tilde{g}(Z^t) - \frac{1}{2\beta}\|X^{t+1} - Z^t - 2W^{t+1}\|^2 + \frac{1}{\beta}\|W^{t+1} - Z^t\|^2 \\
&\leq \left( g(Z^t) + \frac{1}{2\beta}\|X^{t+1} - Z^t - 2W^{t+1}\|^2 - \frac{1}{2\beta}\|Z^{t+1} - Z^t\|^2 - \frac{1}{\beta}\|W^{t+1} - Z^{t+1}\|^2 \right) \\
&\quad - \tilde{g}(Z^t) - \frac{1}{2\beta}\|X^{t+1} - Z^t - 2W^{t+1}\|^2 + \frac{1}{\beta}\|W^{t+1} - Z^t\|^2 \\
&= -\frac{1}{2\beta}\|Z^{t+1} - Z^t\|^2 - \frac{1}{\beta}\|W^{t+1} - Z^{t+1}\|^2 + \frac{1}{\beta}\|W^{t+1} - Z^t\|^2
\end{aligned}
\tag{48}
$$

where the last equality uses equation 2.

Now, we bound the last term in the above inequality. Note that

$$
\begin{aligned}
\|W^{t+1} - Z^t\|^2 &= \|W^{t+1} - W^t + W^t - Z^t\|^2 \\
&= \sum_{i \in \mathcal{S}^t} \|w_i^{t+1} - w_i^t - x_i^{t+1} + x_i^t\|^2 + \sum_{i \notin \mathcal{S}^t} \|w_i^t - z_i^t\|^2 \\
&= \sum_{i \in \mathcal{S}^t} \|w_i^{t+1} - w_i^t\|^2 - 2\langle w_i^{t+1} - w_i^t, x_i^{t+1} - x_i^t \rangle + \|x_i^t - x_i^{t+1}\|^2 + \sum_{i \notin \mathcal{S}^t} \|w_i^t - z_i^t\|^2.
\end{aligned}
\tag{49}
$$

On the other hand, Using Exercise 8.8 of Rockafellar & Wets (1998), it holds that $\partial(F(\cdot) + \frac{L}{2}\|\cdot\|^2)(W) = \nabla F(W) + LW$. Since $F(\cdot) + \frac{L}{2}\|\cdot\|^2$ is convex, we have that $F(\cdot) + \frac{L}{2}\|\cdot\|^2$ is monotone. This together with equation 33 implies that for $i \in \mathcal{S}^t$,

$$
\begin{aligned}
0 &\leq \left\langle -\frac{1}{\beta}(w_{i,\star}^{t+1} - x_i^{t+1}) + Lw_{i,\star}^{t+1} - \left( -\frac{1}{\beta}(w_{i,\star}^t - x_i^t) + Lw_{i,\star}^t \right), w_{i,\star}^{t+1} - w_{i,\star}^t \right\rangle \\
&= \langle \xi_{i,\star}^{t+1} + Lw_{i,\star}^{t+1} - \xi_{i,\star}^t - Lw_{i,\star}^t, w_{i,\star}^{t+1} - w_{i,\star}^t \rangle \\
&= \left\langle -\frac{1}{\beta}(w_i^{t+1} - e_i^{t+1} - x_i^{t+1}) + L\left( w_i^{t+1} - e_i^{t+1} \right) + \frac{1}{\beta}(w_i^t - e_i^t - x_i^t) - L\left( w_i^t - e_i^t \right), w_i^{t+1} - w_i^t \right\rangle \\
&\quad + \left\langle -\frac{1}{\beta}(w_i^{t+1} - e_i^{t+1} - x_i^{t+1}) + L\left( w_i^{t+1} - e_i^{t+1} \right) + \frac{1}{\beta}(w_i^t - e_i^t - x_i^t) - L\left( w_i^t - e_i^t \right), -e_i^{t+1} + e_i^t \right\rangle.
\end{aligned}
$$

Multiply both sides of the above inequality by $2\beta$ and rearranging terms, we have that

$$
\begin{aligned}
&-\langle x_i^{t+1} - x_i^t, w_i^{t+1} - w_i^t \rangle \leq \langle x_i^{t+1} - x_i^t, -e_i^{t+1} + e_i^t \rangle + (\beta L - 1)\|w_i^{t+1} - w_i^t\|^2 \\
&\quad + 2\langle (\beta L - 1)\left( w_i^{t+1} - w_i^t \right), -e_i^{t+1} + e_i^t \rangle + (\beta L - 1)\|e_i^{t+1} - e_i^t\|^2 \\
&\overset{(a)}{\leq} \frac{\iota}{2}\|x_i^{t+1} - x_i^t\|^2 + \frac{1}{2\iota}\| - e_i^{t+1} + e_i^t\|^2 + (\beta L - 1)\|w_i^{t+1} - w_i^t\|^2 \\
&\quad + |\beta L - 1|^2 \frac{\iota}{2}\|w_i^{t+1} - w_i^t\|^2 + \frac{1}{2\iota}\| - e_i^{t+1} + e_i^t\|^2 + (\beta L - 1)\|e_i^{t+1} - e_i^t\|^2 \\
&= \frac{\iota}{2}\|x_i^{t+1} - x_i^t\|^2 + (\beta L - 1 + \frac{|\beta L - 1|^2\iota}{2})\|w_i^{t+1} - w_i^t\|^2 + \left( \frac{1}{\iota} + \beta L - 1 \right)\| - e_i^{t+1} + e_i^t\|^2
\end{aligned}
\tag{50}
$$

where $\iota > 0$ and (a) uses Young's inequality for products.

Combining this with equation 49 we obtain that

$$\|W^{t+1} - Z^t\|^2 \le \sum_{i \notin \mathcal{S}^t} \|w_i^t - z_i^t\|^2 + \sum_{i \in \mathcal{S}^t} \|w_i^{t+1} - w_i^t\|^2 + \|x_i^t - x_i^{t+1}\|^2$$

$$+ \sum_{i \in \mathcal{S}^t} \iota\|x_i^{t+1} - x_i^t\|^2 + (2\beta L - 2 + |\beta L - 1|^2 \iota)\|w_i^{t+1} - w_i^t\|^2$$

$$+ 2\left(\frac{1}{\iota} + \beta L - 1\right)\| - e_i^{t+1} + e_i^t\|^2$$

$$= \sum_{i \notin \mathcal{S}^t} \|w_i^t - z_i^t\|^2 + \sum_{i \in \mathcal{S}^t} (1 + \iota)\|x_i^{t+1} - x_i^t\|^2 + (2\beta L - 1 + |\beta L - 1|^2 \iota)\|w_i^{t+1} - w_i^t\|^2$$

$$+ \sum_{i \in \mathcal{S}^t} 2\left(\frac{1}{\iota} + \beta L - 1\right)\| - e_i^{t+1} + e_i^t\|^2.$$

This together with equation 48 we have that

$$\bar{H}(X^{t+1}, W^{t+1}, Z^{t+1}) - \bar{H}(X^{t+1}, W^{t+1}, Z^t)$$

$$\le -\frac{1}{2\beta}\|Z^{t+1} - Z^t\|^2 - \frac{1}{\beta}\|Z^{t+1} - W^{t+1}\|^2 + \frac{1}{\beta}\sum_{i \notin \mathcal{S}^t} \|w_i^t - z_i^t\|^2$$

$$+ \sum_{i \in \mathcal{S}^t} \frac{1+\iota}{\beta}\|x_i^{t+1} - x_i^t\|^2 + \frac{1}{\beta}(2\beta L - 1 + |\beta L - 1|^2 \iota)\|w_i^{t+1} - w_i^t\|^2$$

$$+ \frac{2}{\beta}\left(\frac{1}{\iota} + \beta L - 1\right)\| - e_i^{t+1} + e_i^t\|^2$$

$$\le -\frac{1}{2\beta}\|Z^{t+1} - Z^t\|^2 - \frac{1}{\beta}\|W^{t+1} - Z^{t+1}\|^2 + \frac{1}{\beta}\sum_{i \notin \mathcal{S}^t} \|w_i^t - z_i^t\|^2$$

$$+ \sum_{i \in \mathcal{S}^t} \frac{1+\iota}{\beta}\|x_i^{t+1} - x_i^t\|^2 + \frac{1}{\beta}(2\beta L - 1 + |\beta L - 1|^2 \iota)\|w_i^{t+1} - w_i^t\|^2$$

$$+ \frac{2}{\beta}\left(\frac{1}{\iota} + \beta L - 1\right)\| - e_i^{t+1} + e_i^t\|^2.$$

Taking expectation on $\mathcal{S}^t$ and then on $\mathcal{Y}^t$, the above inequality becomes

$$\mathbb{E}\bar{H}(X^{t+1}, W^{t+1}, Z^{t+1}) - \mathbb{E}\bar{H}(X^{t+1}, W^{t+1}, Z^t)$$

$$\le -\frac{1}{2\beta}\mathbb{E}\|Z^{t+1} - Z^t\|^2 - \frac{1}{\beta}\mathbb{E}\|W^{t+1} - Z^{t+1}\|^2 + \frac{1}{\beta}\sum_i (1 - p_i)\|w_i^t - z_i^t\|^2$$

$$+ \sum_i p_i \frac{1+\iota}{\beta}\mathbb{E}\|x_i^{t+1} - x_i^t\|^2 + \frac{1}{\beta}(2\beta L - 1 + |\beta L - 1|^2 \iota)\mathbb{E}\|w_i^{t+1} - w_i^t\|^2 \tag{51}$$

$$+ \frac{2}{\beta}\left(\frac{1}{\iota} + \beta L - 1\right)\mathbb{E}\| - e_i^{t+1} + e_i^t\|^2.$$

Now summing equation 44, equation 46 and equation 51, we obtain that

$$
\begin{aligned}
\mathbb{E}\bar{H}&(X^{t+1}, W^{t+1}, Z^{t+1}) - \mathbb{E}\bar{H}(X^t, W^t, Z^t) \\
&\leq \frac{1}{\beta}\sum_i p_i \mathbb{E}\|x_i^{t+1} - x_i^t\|^2 - \frac{1}{2\beta}\mathbb{E}\|Z^{t+1} - Z^t\|^2 - \frac{1}{\beta}\|W^{t+1} - Z^{t+1}\|^2 \\
&\quad + \frac{1}{\beta}\sum_i (1 - p_i)\|w_i^t - z_i^t\|^2 + \sum_i -\frac{\frac{1}{\beta} - L}{2}(1 - \tau^2)p_i \mathbb{E}\|w_i^{t+1} - w_i^t\|^2 \\
&\quad + \frac{\frac{1}{\beta} - L}{2}(\frac{1}{\tau^2} - 1)p_i \mathbb{E}\|w_{i,\star}^{t+1} - w_i^{t+1}\|^2 + \frac{1}{2(\frac{1}{\beta} - L)}(C\epsilon_w p_i \mathbb{E}\Upsilon_{i,t+1}) \\
&\quad + \sum_i \frac{1 + \iota}{\beta}p_i \mathbb{E}\|x_i^{t+1} - x_i^t\|^2 + \frac{1}{\beta}(2\beta L - 1 + |\beta L - 1|^2\iota)p_i \mathbb{E}\|w_i^{t+1} - w_i^t\|^2 \\
&\quad + \frac{2}{\beta}\left(\frac{1}{\iota} + \beta L - 1\right)p_i \mathbb{E}\| - e_i^{t+1} + e_i^t\|^2 \\
&= \frac{1}{\beta}\mathbb{E}\|W^t - Z^t\|^2 - \frac{1}{\beta}\|W^{t+1} - Z^{t+1}\|^2 - \frac{1}{2\beta}\mathbb{E}\|Z^{t+1} - Z^t\|^2 \\
&\quad + \sum_i \frac{\frac{1}{\beta} - L}{2}(\frac{1}{\tau^2} - 1)p_i \mathbb{E}\|w_{i,\star}^{t+1} - w_i^{t+1}\|^2 + \frac{1}{2(\frac{1}{\beta} - L)}(C\epsilon_w p_i \mathbb{E}\Upsilon_{i,t+1}) \\
&\quad + \frac{2}{\beta}\left(\frac{1}{\iota} + \beta L - 1\right)p_i \mathbb{E}\| - e_i^{t+1} + e_i^t\|^2 + \sum_i \frac{1 + \iota}{\beta}p_i \mathbb{E}\|x_i^{t+1} - x_i^t\|^2 \\
&\quad + \left(\frac{1}{\beta}(2\beta L - 1 + |\beta L - 1|^2\iota) - \frac{\frac{1}{\beta} - L}{2}(1 - \tau^2)\right)p_i \mathbb{E}\|w_i^{t+1} - w_i^t\|^2.
\end{aligned}
\tag{52}
$$

On the other hand, equation 41 together with equation 52 yields

$$
\begin{aligned}
\mathbb{E}\bar{H}&(X^{t+1}, W^{t+1}, Z^{t+1}) - \bar{H}(X^t, W^t, Z^t) \\
&\leq \frac{1}{\beta}\mathbb{E}\|W^t - Z^t\|^2 - \frac{1}{2\beta}\mathbb{E}\|Z^{t+1} - Z^t\|^2 - \frac{1}{\beta}\mathbb{E}\|W^{t+1} - Z^{t+1}\|^2 \\
&\quad + \sum_i \frac{1 + \iota}{\beta}\left((1 + \beta L)^2\left((1 + \iota)\mathbb{E}\|W^{t+1} - W^t\|^2 + \left(1 + \frac{1}{\iota}\right)p_i \mathbb{E}\| - e_i^{t+1} - e_i^t\|^2\right)\right) \\
&\quad + \sum_i \frac{\frac{1}{\beta} - L}{2}(\frac{1}{\tau^2} - 1)p_i \mathbb{E}\|w_{i,\star}^{t+1} - w_i^{t+1}\|^2 + \frac{1}{2(\frac{1}{\beta} - L)}(C\epsilon_w \mathbb{E}\Upsilon_{i,t+1}) \\
&\quad + \frac{2}{\beta}\left(\frac{1}{\iota} + \beta L - 1\right)p_i \mathbb{E}\| - e_i^{t+1} + e_i^t\|^2 \\
&\quad + \sum_i \left(\frac{1}{\beta}(2\beta L - 1 + |\beta L - 1|^2\iota) - \frac{\frac{1}{\beta} - L}{2}(1 - \tau^2)\right)p_i \mathbb{E}\|w_i^{t+1} - w_i^t\|^2
\end{aligned}
$$

Rearranging the above term we have

$$\mathbb{E}\bar{H}(X^{t+1}, W^{t+1}, Z^{t+1}) - \bar{H}(X^t, W^t, Z^t)$$

$$\leq \frac{1}{\beta}\mathbb{E}\|W^t - Z^t\|^2 - \frac{1}{2\beta}\mathbb{E}\|Z^{t+1} - Z^t\|^2 - \frac{1}{\beta}\mathbb{E}\|W^{t+1} - Z^{t+1}\|^2$$

$$+ \sum_i \left( \frac{(1+\iota)^2}{\beta\iota} + \frac{2}{\beta}\left(\frac{1}{\iota} + \beta L - 1\right)\right) p_i\mathbb{E}\| - e_i^{t+1} - e_i^t\|^2$$

$$+ \sum_i \frac{\frac{1}{\beta} - L}{2}(\frac{1}{\tau^2} - 1)p_i\mathbb{E}\|w_{i,\star}^{t+1} - w_i^{t+1}\|^2 + \frac{1}{2(\frac{1}{\beta} - L)}\left(C\epsilon_w \mathbb{E}\Upsilon_{i,t+1}\right)$$

$$+ \sum_i \frac{1}{\beta}\left( \underbrace{(2\beta L - 1 + |\beta L - 1|^2\iota) - \frac{1 - L\beta}{2}(1 - \tau^2) + (1+\iota)^2(1+\beta L)^2}_{\Theta}\right) p_i\mathbb{E}\|w_i^{t+1} - w_i^t\|^2$$

$$\tag{53}$$

Now, rearranging the formula of $\Theta$, we have that

$$\Theta = (1+\beta L)^2 - \frac{3}{2} + \frac{5}{2}\beta L + \frac{1 - L\beta}{2}\tau^2 + (1+\beta L)^2(2\iota + \iota^2) + (\beta L - 1)^2\iota$$

$$\leq -\delta_\beta + \frac{1 - L\beta}{2}\tau^2 + (1+\beta L)^2(2\iota + \iota^2) + (\beta L - 1)^2\iota \leq -\delta_\beta + \delta' = -\delta,$$

where the second inequality uses equation 57, the last inequality uses equation 58, and the last equality uses the definition of $\delta$.

Then equation 53 can be further passed to

$$\mathbb{E}\bar{H}(X^{t+1}, W^{t+1}, Z^{t+1}) - \mathbb{E}\bar{H}(X^t, W^t, Z^t)$$

$$\leq \frac{1}{\beta}\mathbb{E}\|W^t - Z^t\|^2 - \frac{1}{2\beta}\mathbb{E}\|Z^{t+1} - Z^t\|^2 - \frac{1}{\beta}\mathbb{E}\|W^{t+1} - Z^{t+1}\|^2$$

$$+ \sum_i \left( \underbrace{\frac{(1+\iota)^2}{\beta\iota} + \frac{2}{\beta}\left(\frac{1}{\iota} + \beta L - 1\right)}_{\Gamma}\right) p_i\mathbb{E}\| - e_i^{t+1} - e_i^t\|^2$$

$$+ \sum_i \frac{\frac{1}{\beta} - L}{2}(\frac{1}{\tau^2} - 1)\mathbb{E}\|W_\star^{t+1} - W^{t+1}\|^2 + \frac{1}{2(\frac{1}{\beta} - L)}\left(C\epsilon_w p_i\mathbb{E}\Upsilon_{i,t+1}\right) - \frac{\delta}{\beta}\mathbb{E}\|W^{t+1} - W^t\|^2.$$

$$\tag{54}$$

Now, using equation 37 and equation 42, equation 54 can be further passed to

$$\mathbb{E}\bar{H}(X^{t+1}, W^{t+1}, Z^{t+1}) - \mathbb{E}\bar{H}(X^t, W^t, Z^t)$$

$$\leq \frac{1}{\beta}\mathbb{E}\|W^t - Z^t\|^2 - \frac{1}{2\beta}\mathbb{E}\|Z^{t+1} - Z^t\|^2 - \frac{1}{\beta}\mathbb{E}\|W^{t+1} - Z^{t+1}\|^2$$

$$+ \sum_i \Gamma\left( \frac{2}{(\frac{1}{\beta} - L)^2}C\epsilon_w p_i \left(\mathbb{E}\Upsilon_{i,t} + \mathbb{E}\Upsilon_{i,t+1}\right)\right)$$

$$+ \sum_i(\frac{1}{\tau^2} - 1)\frac{1}{2(\frac{1}{\beta} - L)}\left(C\epsilon_w p_i\mathbb{E}\Upsilon_{i,t+1}\right) + \frac{1}{2(\frac{1}{\beta} - L)}\left(C\epsilon_w\mathbb{E}\Upsilon_{i,t+1}\right) - \frac{\delta}{\beta}p_i\mathbb{E}\|w_i^{t+1} - w_i^t\|^2$$

$$= \frac{1}{\beta}\mathbb{E}\|W^t - Z^t\|^2 - \frac{1}{2\beta}\mathbb{E}\|Z^{t+1} - Z^t\|^2 - \frac{1}{\beta}\mathbb{E}\|W^{t+1} - Z^{t+1}\|^2$$

$$+ \sum_i \Gamma\left( \frac{2}{(\frac{1}{\beta} - L)^2}C\epsilon_w p_i \left(\mathbb{E}\Upsilon_{i,t} + \mathbb{E}\Upsilon_{i,t+1}\right)\right)$$

$$+ \sum_i \frac{1}{\tau^2}\frac{1}{2(\frac{1}{\beta} - L)}\left(C\epsilon_w p_i\mathbb{E}\Upsilon_{i,t+1}\right) - \frac{\delta}{\beta}p_i\mathbb{E}\|w_i^{t+1} - w_i^t\|^2$$

$$\tag{55}$$

Now, we bound the term with $\Upsilon_{i,t}$ in the above inequality. Using equation 21, the above inequality can be further passed to

$$\mathbb{E}\bar{H}(X^{t+1}, W^{t+1}, Z^{t+1}) - \mathbb{E}\bar{H}(X^t, W^t, Z^t)$$

$$\leq \frac{1}{\beta}\mathbb{E}\|W^t - Z^t\|^2 - \frac{1}{2\beta}\mathbb{E}\|Z^{t+1} - Z^t\|^2 - \frac{1}{\beta}\mathbb{E}\|W^{t+1} - Z^{t+1}\|^2 - \sum_i \frac{\delta}{\beta}p_i\mathbb{E}\|w_i^{t+1} - w_i^t\|^2$$

$$+ \sum_i \left(\Gamma\frac{2}{(\frac{1}{\beta} - L)^2} + \frac{1}{\tau^2}\frac{1}{2(\frac{1}{\beta} - L)}\right)\left(C\epsilon_w\left(\frac{1}{2}(p_i\mathbb{E}\Upsilon_{i,t} - p_i\mathbb{E}\Upsilon_{i,t+1}) + 6L^2 p_i\mathbb{E}\|w_i^{t-1} - w_i^t\|^2\right)\right)$$

$$= \frac{1}{\beta}\mathbb{E}\|W^t - Z^t\|^2 - \frac{1}{2\beta}\mathbb{E}\|Z^{t+1} - Z^t\|^2 - \frac{1}{\beta}\mathbb{E}\|W^{t+1} - Z^{t+1}\|^2 - \sum_i \frac{\delta}{\beta}p_i\mathbb{E}\|w_i^{t+1} - w_i^t\|^2$$

$$+ \sum_i \left(\Gamma\frac{2}{(\frac{1}{\beta} - L)^2} + \frac{1}{\tau^2}\frac{1}{2(\frac{1}{\beta} - L)}\right)C\epsilon_w 6L^2 p_i\mathbb{E}\|w_i^{t-1} - w_i^t\|^2$$

$$+ \sum_i \left(\Gamma\frac{2}{(\frac{1}{\beta} - L)^2} + \frac{1}{\tau^2}\frac{1}{2(\frac{1}{\beta} - L)}\right)\left(C\epsilon_w\left(\frac{1}{2}(p_i\mathbb{E}\Upsilon_{i,t-1} - p_i\mathbb{E}\Upsilon_{i,t})\right)\right)$$

$$\leq \frac{1}{\beta}\mathbb{E}\|W^t - Z^t\|^2 - \frac{1}{2\beta}\mathbb{E}\|Z^{t+1} - Z^t\|^2 - \frac{1}{\beta}\mathbb{E}\|W^{t+1} - Z^{t+1}\|^2 - \frac{\delta}{\beta}p_i\mathbb{E}\|w_i^{t+1} - w_i^t\|^2$$

$$+ \sum_i \frac{\delta}{\beta}p_i\mathbb{E}\|w_i^{t-1} - w_i^t\|^2 + \sum_i \frac{1}{12L^2}(p_i\mathbb{E}\Upsilon_{i,t} - p_i\mathbb{E}\Upsilon_{i,t+1}),$$

(56)

where the last inequality uses the assumption that $\epsilon_w$ is small enough such that $\left(\Gamma\frac{2}{(\frac{1}{\beta} - L)^2} + \frac{1}{\tau^2}\frac{1}{2(\frac{1}{\beta} - L)}\right)6CL^2\epsilon_w \leq \frac{\delta - \delta_\epsilon}{\beta}$.

Rearranging the above inequality and recalling the definition of $H$, we have that

$$\mathbb{E}H(X^{t+1}, W^{t+1}, Z^{t+1}, Y^{t+1}, W^t, Y^t)$$

$$\leq \mathbb{E}H(X^t, W^t, Z^t, Y^t, W^{t-1}, Y^{t-1}, w^{t-2}, y^{t-2}) - \sum_i \frac{\delta_\epsilon}{\beta}p_i\mathbb{E}\|w_i^t - w_i^{t-1}\|^2$$

$$- \sum_i \frac{1}{2\beta}p_i\mathbb{E}\|z_i^{t+1} - z_i^t\|^2.$$

Finally, we summarize and simplify the hyper parameter we use in this proof. In this proof, we first let $\delta_\beta \in (0, \frac{1}{2})$. Let $\beta \in (0, \frac{1}{L})$ be such that

$$(1 + \beta L)^2 - \frac{3}{2} + \frac{5}{2}\beta L < -\delta_\beta.$$ (57)

To satisfy this, we let $\delta_\beta = 1/4$ and $\beta < \frac{-9 + \sqrt{82}}{L}$.

Then we let $\delta' \in [0, \delta_\beta)$. Let $\iota > 0$ and $\tau \in (0, 1)$ be small enough such that

$$\frac{1 - L\beta}{2}\tau^2 + (1 + \beta L)^2(2\iota + \iota^2) + (\beta L - 1)^2\iota < \delta'.$$ (58)

To satisfy this, we let $\delta' = 1/8$, $\tau = 1/\sqrt{8}$, $\iota = 1/64$ and $\beta \leq \frac{3}{10L}$.

Finally, we denote $\delta := \delta_\beta - \delta'$. Suppose that $\epsilon_w$ is small enough such that

$$\left(\Gamma\frac{2}{(\frac{1}{\beta} - L)^2} + \frac{1}{\tau^2}\frac{1}{2(\frac{1}{\beta} - L)}\right)6CL^2\epsilon_w \leq \frac{\delta - \delta_\epsilon}{\beta},$$ (59)

for some $\delta_\epsilon > 0$, where $\Gamma := \frac{(1+\iota)^2}{\beta\iota} + \frac{2}{\beta}\left(\frac{1}{\iota} + \beta L - 1\right)$ and $C$ is defined as in Proposition 5. Note that since $\tau = \tau = 1/\sqrt{8}$ and $\iota = 1/64$ and $\beta L < 1$, then $\Gamma < \frac{(1+\iota)^2}{\beta\iota} + \frac{2}{\beta}\frac{1}{\iota}$ and thus

$$\Gamma\frac{2}{(\frac{1}{\beta} - L)^2} + \frac{1}{\tau^2}\frac{1}{2(\frac{1}{\beta} - L)} \leq 392\frac{\beta}{1 - \beta L}.$$ (60)

To satisfy equation 59, it suffices to let $\delta_\epsilon = 1/16$ and

$$\epsilon_w \leq \frac{392}{96} \frac{(1 - \beta L)^2}{\beta^3} C^{-1} L^{-2}.$$

In summary, by $\delta_\beta = 1/4$, $\delta' = 1/8$, $\tau = 1/\sqrt{8}$, $\iota = 1/64$, $\delta_\epsilon = 1/16$, $\beta < \frac{-9+\sqrt{82}}{L}$ and $\epsilon_w \leq \frac{392}{96} \frac{(1-\beta L)^2}{\beta^3} C^{-1} L^{-2}$, we have the conclusion. $\qquad\square$

Next, we present a corollary that will be used in the convergence analysis.

**Corollary 1.** *Let assumptions in Theorem 4 hold. Denote $H_t := \mathbb{E} H(X^t, W^t, Z^t, Y^t, W^{t-1}, Y^{t-1})$. Then it holds that*

$$d^2(0, \sum_{i=1}^n \nabla f(z^t) + \partial g(z^t)) \leq \frac{n}{\underline{p}} \sum_i C_2 \left(p_i \mathbb{E} \Upsilon_{i,t} + p_i \mathbb{E} \Upsilon_{i,t+1}\right) \tag{61}$$

*where $C_2 := \max\{\left(\frac{4}{\beta^2} + 4L^2\right)(1 + \beta L)^2 (1 + \iota), (1 + \beta L)^2 \left(1 + \frac{1}{\iota}\right) \frac{2}{(\frac{1}{\beta} - L)^2} C \epsilon_w\}$.*

*Proof.* Recalling the definition of $\mathcal{C}$ in equation 4, it holds that

$$N_\mathcal{C}(Z^t) = \left\{ (d_1, \dots, d_n) : \sum_{i=1}^n d_i = 0, d_i \in \mathbb{R}^l \right\}. \tag{62}$$

Using Corollary 10.9 and Proposition 10.5 in Rockafellar & Wets (1998), we have that

$$\partial \tilde{g}(Z^t) = \left\{(\xi^t, 0, \dots, 0) : \xi^t \in \partial g(z^t)\right\} + N_\mathcal{C}(Z^t). \tag{63}$$

combining equation 62 and equation 63, for any $(d_1, \dots, d_n) \in N_\mathcal{C}(Z^t)$ and $\xi^t \in \partial g(z^t)$,

$$\begin{aligned}
\left\| \frac{1}{n} \sum_{i=1}^n \nabla f_i(z^t) + \xi^t \right\|^2 &= \left\| \frac{1}{n} \sum_{i=1}^n \nabla f_i(z^t) + \xi^t + \sum_{i=1}^n d_i \right\|^2 \\
&= n \left\| \frac{1}{n} \nabla f_1(z^t) + \xi^t + \sum_{i=1}^n d_i \right\|^2 + n \sum_{i=2}^n \left\| \frac{1}{n} \nabla f_i(z^t) \right\|^2 = n \|\nabla F(Z^t) + \eta^t\|^2
\end{aligned} \tag{64}$$

where $\eta^t \in \partial \tilde{g}(Z^t)$.

On the other hand, using Lemma 1, we obtain that

$$\nabla f_i(z^t) = -\frac{1}{\beta}(w_{i,\star}^t - x_i^t) + \nabla f_i(z_i^t) - \nabla f_i(w_{i,\star}^t), \text{ for all } i.$$

This together with equation 64 and equation 34 implies that

$$\begin{aligned}
\frac{1}{n} \left\| \frac{1}{n} \sum_{i=1}^n \nabla f_i(z^t) + \xi^t \right\|^2 &\leq \mathbb{E}_t \|\nabla F(Z^t) + \eta^t\|^2 \\
&= \mathbb{E}_t \sum_{i=1}^n \| -\frac{1}{\beta}(w_i^t - e_i^t - x_i^t) + \nabla f_i(z^t) - \nabla f_i(w_{i,\star}^t) + \frac{1}{\beta}(2w_i^t - x_i^t - z_i^t)\|^2 \\
&= \mathbb{E}_t \sum_{i=1}^n \|\frac{1}{\beta} e_i^t + \left(\nabla f_i(z^t) - \nabla f_i(w_i^t)\right) + \left(\nabla f_i(w_i^t) - \nabla f_i(w_{i,\star}^t)\right) + \frac{1}{\beta}(w_i^t - z^t)\|^2 \\
&\leq \mathbb{E}_t \sum_{i=1}^n \left(\frac{4}{\beta^2} + 4L^2\right)\left(\|e_i^t\|^2 + \|z^t - w_i^t\|^2\right),
\end{aligned} \tag{65}$$

where the inequality uses the Lipschiz continuity of $F$ and Cauchy-Schwarz inequality.

On the other hand, since each client has the probability $p_i$ to be sampled, it holds that

$$\mathbb{E}_{\mathcal{S}^t} \sum_{i \in \mathcal{S}^t} \|w_i^t - z^t\|^2 = \sum_{i=1}^n p_i \|w_i^t - z^t\|^2 \geq \underline{p} \sum_{i=1}^n \|w_i^t - z^t\|^2, \tag{66}$$

where $\underline{p} = \min\{p_1, \ldots, p_n\}$. Similarly, we have

$$\mathbb{E}_{\mathcal{S}^t} \sum_{i \in \mathcal{S}^t} \|e_i^{t+1}\|^2 \geq \underline{p} \sum_{i=1}^n \|e_i^{t+1}\|^2.$$

Combining this with equation 65 and equation 66, we have

$$\frac{1}{n} \left\| \frac{1}{n} \sum_{i=1}^n \nabla f_i(z^t) + \xi^t \right\|^2 \leq \frac{1}{\underline{p}} \mathbb{E}_{\mathcal{S}^t} \sum_{i \in \mathcal{S}^t} \left( \frac{4}{\beta^2} + 4L^2 \right) \left( \mathbb{E}_t \|e_i^t\|^2 + \mathbb{E}_t \|Z^t - w_i^t\|^2 \right). \tag{67}$$

Using equation 37 and equation 38, the above inequality can be further passed to

$$\frac{1}{n} \left\| \frac{1}{n} \sum_{i=1}^n \nabla f_i(z^t) + \xi^t \right\|^2$$

$$\leq \frac{1}{\underline{p}} \mathbb{E}_{\mathcal{S}^t} \sum_{i \in \mathcal{S}^t} \left( \frac{4}{\beta^2} + 4L^2 \right) \left( \frac{1}{(\frac{1}{\beta} - L)^2} \left( C\epsilon_w \mathbb{E} \Upsilon_{i,t} \right) \right)$$

$$+ \frac{1}{\underline{p}} \mathbb{E}_{\mathcal{S}^t} \sum_{i \in \mathcal{S}^t} \left( \frac{4}{\beta^2} + 4L^2 \right) \left( (1 + \beta L)^2 (1 + \iota) \mathbb{E} \Upsilon_{i,t+1} + (1 + \beta L)^2 \left( 1 + \frac{1}{\iota} \right) \left( \frac{2}{(\frac{1}{\beta} - L)^2} C\epsilon_w \mathbb{E} \Upsilon_{i,t} \right) \right)$$

$$= \frac{1}{\underline{p}} \mathbb{E}_{\mathcal{S}^t} \sum_{i \in \mathcal{S}^t} \left( \frac{4}{\beta^2} + 4L^2 \right) \left( \frac{1}{(\frac{1}{\beta} - L)^2} \left( C\epsilon_w \mathbb{E} \Upsilon_{i,t} \right) \right)$$

$$+ \frac{1}{\underline{p}} \mathbb{E}_{\mathcal{S}^t} \sum_{i \in \mathcal{S}^t} \left( \frac{4}{\beta^2} + 4L^2 \right) \left( (1 + \beta L)^2 (1 + \iota) \mathbb{E} \Upsilon_{i,t+1} + (1 + \beta L)^2 \left( 1 + \frac{1}{\iota} \right) \left( \frac{2}{(\frac{1}{\beta} - L)^2} C\epsilon_w \mathbb{E} \Upsilon_{i,t} \right) \right)$$

$$\leq \frac{1}{\underline{p}} \mathbb{E}_{\mathcal{S}^t} \sum_{i \in \mathcal{S}^t} C_2 (\mathbb{E} \Upsilon_{i,t} + \mathbb{E} \Upsilon_{i,t+1}) = \frac{1}{\underline{p}} \sum_i C_2 (p_i \mathbb{E} \Upsilon_{i,t} + p_i \mathbb{E} \Upsilon_{i,t+1})$$

where $C_2$ is defined in the statement. Thus, (ii) holds. $\square$

Now, we give the detailed statement of Theorem 2 and its proofs.

**Theorem 5.** *Let assumptions in Theorem 1 hold. Let $\{(X^t, W^t, Z^t)\}$ be generated by Algorithm 1. We further suppose $\epsilon_w$ and $\beta$ are small enough such that $\frac{1}{2(\frac{1}{\beta} - L)} C\epsilon_w + 6L^2 \sum_i p_i \leq \frac{\delta}{\beta}$, where $C$ is defined as in Proposition 5. Then It holds that*

$$\frac{1}{T+1} \sum_{t=1}^{T+1} \mathbb{E} d^2(0, \nabla \sum_{i=1}^n f_i(z^t) + \partial g(z^t)) \leq \frac{n}{\underline{p}} \frac{1}{T+1} \left( D_1 \bar{H}_0 + D_2 \Upsilon_0 + D_3 \|Y^0 - Y(W^0)\|^2 \right),$$

*where $\bar{H}_0 := F(W^0) + \tilde{g}(Z^0) + \frac{1}{2\beta} \left( \|X^0 - W^0\|^2 - \|X^0 - Z^0\|^2 \right)$, $D_1 := \frac{15L^2\beta}{\delta_\epsilon}$, $D_2 := 6 \max\{1, L\}\epsilon_w + \frac{15L^2\beta}{\delta_\epsilon} C_u$, $D_3 := 3C_2 + \frac{15L^2\beta}{\delta_\epsilon} \frac{3}{2(\frac{1}{\beta} - L)} C\epsilon_w$, $D_4 := 13 + \frac{15L^2\beta}{\delta_\epsilon} C_1$, $D_5 := $ with $C_u := 2\Gamma(\epsilon_w + 1) + \frac{\frac{1}{\beta} - L}{2} (\frac{1}{\tau^2} - 1)\epsilon_w + 6 \max\{1, L\}\epsilon_w$.*

*Proof.* Using equation 61, it holds that

$$\sum_{t=1}^{T} \mathbb{E}\|\frac{1}{n}\sum_{i=1}^{n}\nabla f_i(z^t) + \xi^t\|^2 \le \frac{n}{\underline{p}}C_2 \sum_i C_2 \left(p_i\mathbb{E}\Upsilon_{i,t} + p_i\mathbb{E}\Upsilon_{i,t+1}\right)$$

$$\le \frac{n}{\underline{p}}C_2 \left(2\sum_{t=1}^{T+1}\sum_i p_i\mathbb{E}\Upsilon_{i,t}\right) \tag{68}$$

$$\le \frac{n}{\underline{p}}C_2 \left(\mathbb{E}\sum_{i=1}^{n}\Upsilon_{i,1} + 12L^2\sum_{t=1}^{T+1}\sum_i p_i\mathbb{E}\|w_i^{t-1} - w_i^t\|^2\right)$$

where the last inequality uses equation 21. We next bound $\mathbb{E}\Upsilon_1$.

$$\mathbb{E}\Upsilon_1 = \mathbb{E}\|(w^0, Y^0) - (W^1, Y^1)\|^2$$
$$\le 3\|(W^0, Y^0) - (W^0, Y(W^0))\|^2$$
$$+ 3\mathbb{E}\|(W^0, Y(W^0)) - (W^1, Y(W^1))\|^2 + 3\mathbb{E}\|(W^1, Y(W^1)) - (W^1, Y^1)\|^2 \tag{69}$$
$$= 3\|Y^0 - Y(W^0)\|^2 + 3\mathbb{E}\|(W^0, Y(W^0)) - (W^1, Y(W^1))\|^2 + 3\mathbb{E}\|Y(W^1) - Y^1\|^2$$
$$\le 3\|Y^0 - Y(W^0)\|^2 + 3L^2\mathbb{E}\|W^0 - W^1\|^2 + 3\mathbb{E}\|Y(W^1) - Y^1\|^2,$$

where the second inequality uses Proposition 1. Note that

$$\mathbb{E}\|Y^1 - Y(W^1)\|^2 \le 2\mathbb{E}\|Y^1 - Y(W_\star^1)\|^2 + 2\mathbb{E}\|Y(W_\star^1) - Y(W^1)\|^2$$
$$\le 2\max\{1, L\}\mathbb{E}\left(\|Y^1 - Y(W_\star^1)\|^2 + \|W_\star^1 - W^1\|^2\right) \tag{70}$$
$$\le 2\max\{1, L\}\epsilon_w\Upsilon_0,$$

where the second inequality is thanks to equation 10. Combining equation 69 with equation 70, we have that

$$\mathbb{E}\Upsilon_1 \le 3\|Y^0 - Y(W^0)\|^2 + 3L^2\mathbb{E}\|W^0 - W^1\|^2 + 3\left(2\max\{1, L\}\epsilon_w\Upsilon_0\right) \tag{71}$$

Combining equation 71 with equation 68, it holds that

$$\sum_{t=1}^{T}\mathbb{E}\|\frac{1}{n}\sum_{i=1}^{n}\nabla f_i(z^t) + \xi^t\|^2 \le \frac{n}{\underline{p}}\left(12L^2\sum_{t=1}^{T+1}\sum_i p_i\mathbb{E}\|w_i^{t-1} - w_i^t\|^2\right)$$
$$+ \frac{n}{\underline{p}}C_2\left(3\|Y^0 - Y(W^0)\|^2 + 3L^2\mathbb{E}\|W^0 - W^1\|^2 + 3\left(2\max\{1, L\}\epsilon_w\Upsilon_0\right)\right)$$
$$\le \frac{n}{\underline{p}}\left(15L^2\sum_{t=1}^{T+1}\sum_i p_i\mathbb{E}\|w_i^{t-1} - w_i^t\|^2\right) + C_2\frac{n}{\underline{p}}\left(3\|Y^0 - Y(W^0)\|^2 + 3\left(2\max\{1, L\}\epsilon_w\Upsilon_0\right)\right). \tag{72}$$

On the other hand, rearranging equation 40, we have that

$$\sum_i p_i\mathbb{E}\|w_i^t - w_i^{t-1}\|^2$$
$$\le \frac{\beta}{\delta_\epsilon}\left(\mathbb{E}H(X^t, W^t, Z^t, Y^t, W^{t-1}, Y^{t-1}) - \mathbb{E}H(X^{t+1}, W^{t+1}, Z^{t+1}, Y^{t+1}, W^t, Y^t)\right)$$
$$- \sum_i \frac{\beta}{\delta_\epsilon}\frac{1}{2\beta}p_i\mathbb{E}\|z_i^{t+1} - z_i^t\|^2$$
$$\le \frac{\beta}{\delta_\epsilon}\left(\mathbb{E}H(X^t, W^t, Z^t, Y^t, W^{t-1}, Y^{t-1}) - \mathbb{E}H(X^{t+1}, W^{t+1}, Z^{t+1}, Y^{t+1}, W^t, Y^t)\right).$$

Summing the above inequality from $t = 1$ to $T + 1$, we deduce that

$$\sum_{t=1}^{T+1}\sum_i p_i\mathbb{E}\|w_i^t - w_i^{t-1}\|^2$$
$$\le \frac{\beta}{\delta_\epsilon}\left(\mathbb{E}H(X^1, W^1, Z^1, Y^1, W^0, Y^0) - \mathbb{E}H(X^{T+1}, W^{T+1}, Z^{T+1}, Y^{T+1}, W^T, Y^T)\right) \tag{73}$$
$$\le \frac{\beta}{\delta_\epsilon}\left(\mathbb{E}H(X^1, W^1, Z^1, Y^1, W^0, Y^0) - B\right),$$

where $B$ is the lower bound of $\mathbb{E}H(X^{T+1}, W^{T+1}, Z^{T+1}, Y^{t+1}, W^T, Y^T)$ guaranteed in Corollary 1.

Now we bound $\mathbb{E}H(X^1, W^1, Z^1, Y^1, W^0, Y^0)$. To this end, we first bound $\mathbb{E}\bar{H}(x^1, W^1, z^1)$, where $\bar{H}$ is defined in equation 43. Making use of equation 55, it holds that

$$\mathbb{E}\bar{H}(X^1, W^1, Z^1) - \mathbb{E}\bar{H}(X^0, W^0, Z^0) \leq \frac{1}{\beta}\mathbb{E}\|W^0 - z^0\|^2 - \frac{1}{2\beta}\mathbb{E}\|z^1 - z^0\|^2 - \frac{1}{\beta}\mathbb{E}\|W^1 - z^1\|^2$$

$$+ \left(\Gamma\frac{2}{(\frac{1}{\beta} - L)^1} + \frac{1}{\tau^1}\frac{1}{2(\frac{1}{\beta} - L)}\right)(C\epsilon_w(\mathbb{E}\Upsilon_1 + \Upsilon_0)) - \frac{\delta}{\beta}\mathbb{E}\|W^1 - W^0\|^2$$

$$= \frac{1}{\beta}\mathbb{E}\|W^0 - z^0\|^2 - \frac{1}{2\beta}\mathbb{E}\|z^1 - z^0\|^2 - \frac{1}{\beta}\mathbb{E}\|W^1 - z^1\|^2$$

$$+ \left(\Gamma\frac{2}{(\frac{1}{\beta} - L)^1} + \frac{1}{\tau^1}\frac{1}{2(\frac{1}{\beta} - L)}\right)(C\epsilon_w(\mathbb{E}\Upsilon_1 + \Upsilon_0)) - \frac{\delta}{\beta}\mathbb{E}\|W^1 - W^0\|^2, \tag{74}$$

where the last equality use equation 2 and the settings that $W^0 = z^0$ at Step 1 in Algorithm 1. Using equation 10, it holds that

$$\mathbb{E}\|e^1\|^2 \leq \epsilon_w\Upsilon_1 \tag{75}$$

and

$$\mathbb{E}\| - e^1 - e^0\|^2 \leq 2\mathbb{E}\|e^1\|^2 + 2\|e^0\|^2 \leq 2(\epsilon_w\Upsilon_1) + 2\Upsilon_0$$

$$\leq 2((\epsilon_w + 1)\Upsilon_0) + 6L^2\sum_i p_i\mathbb{E}\|w_i^0 - w_i^1\|^2, \tag{76}$$

where the last inequality uses equation 21. Combining equation 75 and equation 76 with equation 74, we have that

$$\mathbb{E}\bar{H}(X^1, W^1, Z^1) - \mathbb{E}\bar{H}(X^0, W^0, Z^0)$$

$$\leq -\frac{1}{2\beta}\mathbb{E}\|Z^1 - Z^0\|^2 - \frac{1}{\beta}\mathbb{E}\|w^1 - Z^1\|^2 + 2\Gamma((\epsilon_w + 1)\Upsilon_0)$$

$$+ \frac{\frac{1}{\beta} - L}{2}(\frac{1}{\tau^2} - 1)(\epsilon_w\Upsilon_0) + \frac{1}{2(\frac{1}{\beta} - L)}(C\epsilon_w\Upsilon_1) - \frac{\delta}{\beta}\mathbb{E}\|w^1 - W^0\|^2 \tag{77}$$

$$\leq 2\Gamma((\epsilon_w + 1)\Upsilon_0) + \frac{\frac{1}{\beta} - L}{2}(\frac{1}{\tau^2} - 1)(\epsilon_w\Upsilon_0)$$

$$+ \frac{1}{2(\frac{1}{\beta} - L)}C\epsilon_w\Upsilon_1 - \frac{\delta}{\beta}\mathbb{E}\|w^1 - W^0\|^2 + 6L^2\sum_i p_i\mathbb{E}\|w_i^0 - w_i^1\|^2.$$

Combining equation 71 with equation 77, we have that

$$\mathbb{E}\bar{H}(X^1, W^1, Z^1) - \mathbb{E}\bar{H}(X^0, W^0, Z^0) \leq 2\Gamma((\epsilon_w + 1)\Upsilon_0) + \frac{\frac{1}{\beta} - L}{2}(\frac{1}{\tau^2} - 1)(\epsilon_w\Upsilon_0)$$

$$+ \frac{1}{2(\frac{1}{\beta} - L)}C\epsilon_w\left(3\|Y^0 - Y(W^0)\|^2 + 3\left(2\max\{1, L\}\epsilon_w\Upsilon_0\right)\right)$$

$$+ \frac{1}{2(\frac{1}{\beta} - L)}C\epsilon_w 3L^2\mathbb{E}\|W^0 - W^1\|^2 - \frac{\delta}{\beta}\mathbb{E}\|w^1 - W^0\|^2 + 6L^2\sum_i p_i\mathbb{E}\|w_i^0 - w_i^1\|^2 \tag{78}$$

$$\leq 2\Gamma((\epsilon_w + 1)\Upsilon_0) + \frac{\frac{1}{\beta} - L}{2}(\frac{1}{\tau^2} - 1)(\epsilon_w\Upsilon_0)$$

$$+ \frac{1}{2(\frac{1}{\beta} - L)}C\epsilon_w\left(3\|Y^0 - Y(W^0)\|^2 + 3\left(2\max\{1, L\}\epsilon_w\Upsilon_0\right)\right),$$

where the last inequality uses the assumption that $\epsilon_w$ and $\beta$ are small enough such that $\frac{1}{2(\frac{1}{\beta} - L)}C\epsilon_w + 6L^2\sum_i p_i \leq \frac{\delta}{\beta}$.

Rearranging the above inequality, recalling the definition of $H$, we have that

$$\mathbb{E}H(X^1, W^1, Z^1, Y^1, W^0, Y^0)$$

$$\leq \mathbb{E}\bar{H}(X^0, W^0, Z^0) + 2\Gamma((\epsilon_w + 1)\Upsilon_0) + \frac{\frac{1}{\beta} - L}{2}(\frac{1}{\tau^2} - 1)(\epsilon_w \Upsilon_0)$$

$$+ \frac{1}{2(\frac{1}{\beta} - L)} C\epsilon_w \left(3\|Y^0 - Y(W^0)\|^2 + 3\left(2\max\{1, L\}\epsilon_w \Upsilon_0\right)\right)$$

$$= F(W^0) + g(z^0) + \frac{1}{2\beta}\left(\|x^0 - W^0\|^2 - \|x^0 - z^0\|^2\right) + 2\Gamma((\epsilon_w + 1)\Upsilon_0)$$

$$+ \frac{\frac{1}{\beta} - L}{2}(\frac{1}{\tau^2} - 1)(\epsilon_w \Upsilon_0) \tag{79}$$

$$+ \frac{1}{2(\frac{1}{\beta} - L)} C\epsilon_w \left(3\|Y^0 - Y(W^0)\|^2 + 3\left(2\max\{1, L\}\epsilon_w \Upsilon_0\right)\right)$$

$$= F(W^0) + g(z^0) + \frac{1}{2\beta}\left(\|x^0 - W^0\|^2 - \|x^0 - z^0\|^2\right) + C_u \Upsilon_0$$

$$+ \frac{3}{2(\frac{1}{\beta} - L)} C\epsilon_w \|Y^0 - Y(W^0)\|^2$$

where $C_u := 2\Gamma(\epsilon_w + 1) + \frac{\frac{1}{\beta} - L}{2}(\frac{1}{\tau^2} - 1)\epsilon_w + 6\max\{1, L\}\epsilon_w$, $C_v := 2\Gamma + \frac{\frac{1}{\beta} - L}{2}(\frac{1}{\tau^2} - 1) + \frac{1}{2(\frac{1}{\beta} - L)} + 3$.
Now, summing equation 73 and equation 79, we have that

$$\sum_{t=1}^{T+1} p_i \mathbb{E}\|w_i^t - w_i^{t-1}\|^2$$

$$\leq -\frac{\beta}{\delta_\epsilon} B + \frac{\beta}{\delta_\epsilon}$$

$$\cdot \left(F(W^0) + g(z^0) + \frac{1}{2\beta}\left(\|x^0 - W^0\|^2 - \|x^0 - z^0\|^2\right) + C_u \Upsilon_0 + \frac{3}{2(\frac{1}{\beta} - L)} C\epsilon_w \|Y^0 - Y(W^0)\|^2\right).$$

Recalling equation 72 and the definition of $\eta^t$, we have that

$$\sum_{t=1}^{T+1} \mathbb{E}d^2(0, \nabla \sum_{i=1}^{n} f_i(z^t) + \partial g(z^t)) \leq \sum_{t=1}^{T+1} \mathbb{E}\|\frac{1}{n}\sum_{i=1}^{n} \nabla f_i(z^t) + \xi^t\|^2$$

$$\leq \frac{n}{p} C_2 \left(3\|Y^0 - Y(W^0)\|^2 + 3\frac{1}{p}\left(2\max\{1, L\}\epsilon_w \Upsilon_0\right)\right) + \frac{n}{p}\frac{15L^2\beta}{\delta_\epsilon}$$

$$\cdot \left(F(W^0) + g(z^0) + \frac{1}{2\beta}\left(\|x^0 - W^0\|^2 - \|x^0 - z^0\|^2\right) + C_u \Upsilon_0 + \frac{3}{2(\frac{1}{\beta} - L)} C\epsilon_w \|Y^0 - Y(W^0)\|^2\right).$$

Finally, dividing both sides with $T + 1$, we reach the conclusion. $\square$

## C DETAILS FOR RESULTS IN SECTION 4.2

We start with the following properties of the generated sequences.

**Theorem 6.** *Let assumptions in Theorem 4 hold. Suppose Assumption 3 holds. Suppose $F$ and $g$ are bounded from below and $g$ is level-bounded. Then the following statements hold.*

*(i)* $\{H_t\}$ *is convergent.*

*(ii)* $\lim \|X^{t+1} - X^t\| = \lim \|W^{t+1} - W^t\| = \lim \|Z^{t+1} - Z^t\| = \lim \|Y^{t+1} - Y^t\| = 0.$

*Proof.* For (i), since $g$ is level bounded and noting that $\bar{H}(X^t, W^t, Z^t) \leq H(X^t, W^t, Z^t, Y^t, W^{t-1}, Y^{t-1})$, forllowing similar arguments in Theorem 4 of Li & Pong (2016), it is easy to show that $\{(X^t, W^t, Z^t)\}$ is bounded when $\epsilon_w$ is small enough. Then we have that Note that

$$H(X^{t+1}, W^{t+1}, Z^{t+1}, W^t, Y^t) \geq F(W^{t+1}) + g(Z^{t+1}) - \frac{1}{2\beta} - \|X^{t+1} - Z^{t+1}\|^2$$

$$\geq B_f + B_g - \frac{2}{\beta} B_s^2.$$

where $B_f$, $B_g$ and $B_s$ in the second inequality are the lower bounds of $f$ and $g$ and bounds of $\{X^{t+1}\}$ and $\{Z^{t+1}\}$. This together with equation 40 shows that $H_t$ is nonincreasing. Thus, $\{H_t\}$ is convergent.

For (ii), since all clients attend training in eahc round, we have $p_1 = \cdots = p_n = 1$. Summing equation 40 from $t = 2$ to $T$, we have that

$$H(X^{T+1}, W^{T+1}, Z^{T+1}, W^T, Y^T)$$

$$\leq H(X^2, W^2, Z^2, W^1, Y^1) - \frac{\delta_\epsilon}{\beta} \sum_{t=1}^{T} \|W^{t+1} - W^t\|^2 - \frac{1}{2\beta} \sum_{t=1}^{T} \|Z^{t+1} - Z^t\|^2.$$

Rearranging the above inequality we have that

$$\frac{\delta_\epsilon}{\beta} \sum_{t=1}^{T} \|W^{t+1} - W^t\|^2 + \frac{1}{2\beta} \sum_{t=1}^{T} \|Z^{t+1} - Z^t\|^2$$

$$\leq H(X^2, W^2, Z^2, W^1, Y^1) - H(X^{T+1}, W^{T+1}, Z^{T+1}, W^T, Y^T) \tag{80}$$

$$\leq H(X^2, W^2, Z^2, W^1, Y^1) - \lim_{T \to \infty} H(X^{T+1}, W^{T+1}, Z^{T+1}, W^T, Y^T) < \infty,$$

where the second inequality is because $\{H(X^{T+1}, W^{T+1}, Z^{T+1}, W^T, Y^T)\}$ is convergent and nonincreading in the deterministic case thanks to equation 40. Taking $T$ in the above inequality to infinity, we deduce that $\{\|W^{t+1} - W^t\|\}$ and $\{\|Z^{t+1} - Z^t\|\}$ are summable. This implies that $\lim_t \|W^{t+1} - W^t\| = \lim_t \|Z^{t+1} - Z^t\| = 0$. The $\lim_t \|X^{t+1} - X^t\| = 0$ follows from equation 38. Now, using the deterministic case of equation 21 and the definition of $\Upsilon_{t+1}$, we have that

$$\sum_{t=0}^{T} \|Y^{t+1} - Y^t\|^2 \leq \sum_{t=0}^{T} \Upsilon_{t+1} \leq \frac{1}{2} (\Upsilon_0 - \Upsilon_{T+1}) + 6L^2 \sum_{t=0}^{T} \|W^t - W^{t+1}\|^2$$

$$\leq \sum_{t=0}^{T} \Upsilon_{t+1} \leq \frac{1}{2} \Upsilon_0 + 6L^2 \sum_{t=0}^{T} \|W^t - W^{t+1}\|^2. \tag{81}$$

Since $\{\|W^{t+1} - W^t\|\}$ is summable, taking $T$ in the above inequality to infinity, we deduce that $\lim_t \|Y^{t+1} - Y^t\| = 0$. □

Next, we show how the accumulation points of $\{(X^t, W^t, Z^t, Y^t)\}$ behave.

**Theorem 7.** *Let assumptions in Theorem 6 hold. Suppose Assumption 3 holds. Then $\{Y^t\}$ is bounded. Let $(X^*, W^*, Z^*, Y^*)$ be any accumulation point of $\{(X^t, W^t, Z^t, Y^t)\}$. Then the following results hold.*

*(i)* $W^* = Z^*$ *and* $Z^*$ *is a stationary point of equation 1.*

*(ii)* $H(X, W, Z, W', Y')$ *is constant on the set of accumulation points of* $\{(X^{t+1}, W^{t+1}, Z^{t+1}, Y^t)\}$.

*Proof.* We first show $\{Y^t\}$ is bounded. In fact, thanks to the first relation in equation 33 and the boundedness of $\{X^t\}$ shown in Theorem 6, we deduce that $\{Y(W_\star^{t+1})\}$ is bounded. This together with the fact that $\|Y^{t+1}\| \leq \|Y^{t+1} - Y(W_\star^{t+1})\| + \|Y(W_\star^{t+1})\|$ implies that $\{Y^t\}$ is bounded.

For (i), since $(X^*, W^*, Z^*, Y^*)$ is an accumulation point of $\{(X^t, W^t, Z^t, Y^t)\}$, there exists $\{t_j\}_j$ with $\lim_j (X^{t_j}, W^{t_j}, Z^{t_j}, Y^{t_j}) = (X^*, W^*, Z^*, Y^*)$. Using the fact that $\lim_t \|X^{t+1} - X^t\| = 0$ and equation 2, we know that $W^* = Z^*$. Using Lemma 1, there exists $\eta^t \in \partial \tilde{g}(Z^t)$ such that equation 33 and equation 34 hold. Thus,

$$
\begin{aligned}
0 &= \left( \frac{1}{n} \nabla f_1(w_{1,\star}^t) + \frac{1}{\beta}(w_1^t - e_1^t - x_1^t), \ldots, \frac{1}{n} \nabla f_n(w_{n,\star}^t) + \frac{1}{\beta}(w_n^t - e_n^t - x_n^t) \right) + \eta^t \\
&\quad - \frac{1}{\beta}(2W^t - X^t - Z^t) \\
&= \nabla F(W_\star^t) + \eta^t - \frac{1}{\beta}(e_1^t, \ldots, e_n^t) - \frac{1}{\beta}(X^{t+1} - X^t).
\end{aligned}
\tag{82}
$$

where the second equality uses equation 2.

On the other hand, note that $z^{t_j}$ is the minimizer of equation 3, $Z^{t_j} = \text{Prox}_{\tilde{g}}(2W^{t_j} - X^{t_j})$ and thus

$$
g(Z^{t_j}) + \frac{1}{2\beta} \|2W^{t_j} - X^{t_j} - Z^{t_j}\|^2 \leq g(Z^*) + \frac{1}{2\beta} \|2W^{t_j} - X^{t_j} - Z^*\|^2. \tag{83}
$$

Letting $i$ in the above inequality goes to infinity and making use of (i), we have that

$$
\begin{aligned}
\lim_j g(Z^{t_j}) + \frac{1}{\beta} \|W^* - X^*\|^2 &= \lim_j g(Z^{t_j}) + \frac{1}{\beta} \|W^{t_j} - X^{t_j}\|^2 \\
&\leq \limsup_i \left( g(Z^{t_j}) + \frac{1}{2\beta} \|2W^{t_j} - X^{t_j} - Z^{t_j}\|^2 \right) \\
&\quad - \lim_j \left( \frac{1}{2\beta} \|2W^{t_j} - X^{t_j} - Z^{t_j}\|^2 + \frac{1}{2\beta} \|W^{t_j} - X^{t_j}\|^2 \right) \\
&\leq g(Z^*) + \frac{1}{2\beta} \|W^* - X^*\|^2,
\end{aligned}
\tag{84}
$$

where the first equality makes use of $W^* = Z^*$, which implies that $\limsup_i g(Z^{t_j}) \leq g(Z^*)$. Thus, we have that $\limsup_i g(Z^{t_j}) \leq g(Z^*)$. This together with the closedness of $g$ gives that $\lim_j g(Z^{t_j}) = g(Z^*)$.

Combining equation 37 and Theorem 6 (ii), we deduce that $\lim_t \|e_i^t\| = 0$ and $\lim_t W_\star^t = W^*$. With this fact and equation 84, letting $t$ in equation 82 be $t_i$ and letting $i$ goes to infinity, recalling (i) and the continuity of $\nabla F$, we obtain that

$$
0 = \lim_j \nabla F(W_\star^t) + \lim_j \eta^{t_j} \in \nabla F(W^*) + \partial g(Z^*) = \nabla F(Z^*) + \partial g(Z^*),
$$

where the last equality uses the fact that $W^* = Z^*$. This together with Exercise 8.8 of Rockafellar & Wets (1998) gives the conclusion.

For (ii), we first note that thank to Theorem 6 (ii), it holds that $\lim_j Y^{t_j-1} = \lim_j Y^{t_j} = Y^*$, $\lim_j W^{t_j-1} = \lim_j W^{t_j} = W^*$. Denote $\Upsilon_t = \sum_{i=1}^n \Upsilon_{i,t}$. Then $\lim_j \Upsilon_{t_j} = 0$. Using Theorem 6 (i), we know that there exists $H_*$ such that $\lim_t H(X^t, W^t, Z^t, Y^t, W^{t-1}, Y^{t-1}) = H_*$. On the other hand, note that

$$
\begin{aligned}
&\|X^t - W^t\|^2 - \left( \|X^t - W^t\|^2 - 2\langle X^t - W^t, Z^t - W^t \rangle + \|W^t - Z^t\|^2 \right) \\
&= \|X^t - W^t\|^2 \\
&\quad - \left( \|X^t - W^t\|^2 - \|X^t - Z^t - 2W^t\|^2 + \|X^t - W^t\|^2 + \|Z^t - W^t\|^2 + \|W^t - Z^t\|^2 \right) \\
&= \|X^t - Z^t - 2W^t\|^2 - \|X^t - W^t\|^2 - 2\|W^t - Z^t\|^2.
\end{aligned}
\tag{85}
$$

Then

$$H_* = \lim_t H(X^t, W^t, Z^t, Y^t, W^{t-1}, Y^{t-1})$$

$$= \lim_j H(X^{t_j}, W^{t_j}, Z^{t_j}, W^{t_j-1}, Y^{t_j-1}, W^{t_j-2}, Y^{t_j-2})$$

$$= \lim_j \bar{H}(X^{t_j}, W^{t_j}, Z^{t_j}) + \frac{\delta}{\beta}\|W^{t_j} - W^{t_j-1}\|^2 + \frac{1}{12L^2}\lim_j \Upsilon_{t_j-1}$$

$$\overset{(a)}{=} \lim_j F(W^{t_j}) + g(Z^{t_j}) + \frac{1}{2\beta}\|X^{t_j} - Z^{t_j} - 2W^{t_j}\|^2$$

$$+ \frac{1}{2\beta}\left(-\|X^{t_j} - W^{t_j}\|^2 - 2\|W^{t_j} - Z^{t_j}\|^2\right) + \frac{\delta}{\beta}\|W^{t_j} - W^{t_j-1}\|^2$$

$$\overset{(b)}{=} F(W^*) + g(Z^*) = F(W^*) + g(Z^*) + \frac{1}{2\beta}\left(\|X^* - W^*\|^2 - \|X^* - Z^*\|^2\right) + \frac{1}{\beta}\|W^* - Z^*\|^2$$

$$\overset{(c)}{=} H(X^*, W^*, Z^*, W^*, W^*, Y^*),$$

$$(86)$$

where (a) uses equation 85 and the fact that $\lim_j \Upsilon_{t_j-1} = 0$, (b) and (c) use the continuity of $F$ and the fact that $\lim g(Z^{t_j}) = g(Z^*)$, $\lim_j W^{t_j-1} = \lim_j W^{t_j} = W^*$ and the fact that $W^* = Z^*$. $\quad\square$

To analyze the convergence rate of the generated sequence, we need the following additional theorem.

**Theorem 8.** *Let assumptions in Theorem 6 hold. Suppose Assumption 3 holds. Then, there exists $\Gamma_1 > 0$, $\Gamma_2 > 0$ and $\Gamma_3$ such that*

$$d(0, \partial H(X^{t+1}, W^{t+1}, Z^{t+1}, Y^{t+1}, W^t, Y^t))$$
$$\leq \Gamma_1 \|W^{t+2} - W^{t+1}\| + \Gamma_2 \|W^{t+1} - W^t\| + \Gamma_3 \sqrt{\Upsilon_t}.$$
$$(87)$$

**Remark 5.** *Note that this bound holds whenever $W^{t+1}$ in equation 10 is solved using a deterministic or stochastic method.*

*Proof.* Using Proposition 10.5 of Rockafellar & Wets (1998) together with Exercise 8.8 of Rockafellar & Wets (1998), we have that

$$\partial H(X^{t+1}, W^{t+1}, Z^{t+1}, Y^{t+1}, W^t, Y^t)$$

$$= \begin{bmatrix} \frac{1}{\beta}(Z^{t+1} - W^{t+1}) \\ \nabla F(W^{t+1}) - \frac{1}{\beta}(X^{t+1} - W^{t+1}) + \frac{2\delta}{\beta}(W^{t+1} - W^t) + \frac{2}{\beta}(W^{t+1} - Z^{t+1}) + \frac{1}{6L^2}(W^{t+1} - W^t) \\ \partial\tilde{g}(Z^{t+1}) - \frac{1}{\beta}(X^{t+1} - Z^{t+1}) - \frac{2}{\beta}(W^{t+1} - Z^{t+1}) \\ -\frac{1}{6L^2}(W^{t+1} - W^t) \\ \frac{1}{6L^2}(Y^{t+1} - Y^t) \\ -\frac{1}{6L^2}(Y^{t+1} - Y^t) \end{bmatrix}$$

$$= \begin{bmatrix} -\frac{1}{\beta}(X^{t+2} - X^{t+1}) \\ [\mathcal{A}_1, \dots, \mathcal{A}_n] \\ \partial\tilde{g}(Z^{t+1}) - \frac{1}{\beta}(X^{t+1} - Z^{t+1}) - \frac{2}{\beta}(W^{t+1} - Z^{t+1}) \\ -\frac{1}{6L^2}(W^{t+1} - W^t) \\ \frac{1}{6L^2}(Y^{t+1} - Y^t) \\ -\frac{1}{6L^2}(Y^{t+1} - Y^t) \end{bmatrix}$$

$$(88)$$

where $\mathcal{A}_i := \nabla\frac{1}{n}f_i(w_i^{t+1}) - \frac{1}{\beta}(x_i^{t+1} - w_i^{t+1}) + \frac{2\delta}{\beta}(w_i^{t+1} - w_i^t) + \frac{2}{\beta}(w_i^{t+1} - z_i^{t+1}) + \frac{1}{6L^2}(w_i^{t+1} - w_i^t)$ and the second equation makes uses the equation 2. Now we bound the second and third coordinates of in the above matrix.

Using equation 33, it holds that

$$
\begin{aligned}
\mathcal{A}_i &= \nabla \frac{1}{n} f_i(w_i^{t+1}) - \frac{1}{\beta}(x_i^{t+1} - w_i^{t+1}) + \frac{2\delta}{\beta}(w_i^{t+1} - w_i^t) + \frac{2}{\beta}(w_i^{t+1} - z_i^{t+1}) \\
&\quad + \frac{1}{6L^2}(w_i^{t+1} - w_i^t) - \frac{1}{\beta}(w_i^{t+1} - e_i^{t+1} - x_i^{t+1}) - \nabla \frac{1}{n} f_i(w_{i,\star}^{t+1}) \\
&= \nabla \frac{1}{n} f_i(w_i^{t+1}) - \nabla \frac{1}{n} f_i(w_{i,\star}^{t+1}) + \frac{2\delta}{\beta}(w_i^{t+1} - w_i^t) + \frac{2}{\beta}(w_i^{t+1} - z_i^{t+1}) \\
&\quad + \frac{1}{\beta}e_i^{t+1} + \frac{1}{6L^2}(w_i^{t+1} - w_i^t) \\
&= \nabla \frac{1}{n} f_i(w_i^{t+1}) - \nabla \frac{1}{n} f_i(w_{i,\star}^{t+1}) + \frac{2\delta}{\beta}(w_i^{t+1} - w_i^t) + \frac{2}{\beta}(x_i^{t+2} - x_i^{t+1}) + \frac{1}{\beta}e_i^{t+1} \\
&\quad + \frac{1}{6L^2}(w_i^{t+1} - w_i^t),
\end{aligned}
\tag{89}
$$

where the second equality uses equation 2. Thus, using Cauchy-Schwarz inequality, we have that

$$
\begin{aligned}
&\|\mathcal{A}_i\| \\
&= 4\|\nabla \frac{1}{n} f_i(w_i^{t+1}) - \nabla \frac{1}{n} f_i(w_{i,\star}^{t+1})\|^2 + \frac{16\delta^2}{\beta^2}\|w_i^{t+1} - w_i^t\|^2 + \frac{16}{\beta^2}\|x_i^{t+2} - x_i^{t+1}\|^2 + \frac{4}{\beta^2}\|e_i^{t+1}\|^2 \\
&\quad + \frac{1}{6L^2}(w_i^{t+1} - w_i^t) \\
&\leq 4L^2\|w_i^{t+1} - w_{i,\star}^{t+1}\|^2 + \frac{16\delta^2}{\beta^2}\|w_i^{t+1} - w_i^t\|^2 + \frac{16}{\beta^2}\|x_i^{t+2} - x_i^{t+1}\|^2 + \frac{4}{\beta^2}\|e_i^{t+1}\|^2 \\
&\quad + \frac{1}{6L^2}(w_i^{t+1} - w_i^t) \\
&= \frac{16\delta^2}{\beta^2}\|w_i^{t+1} - w_i^t\|^2 + \frac{16}{\beta^2}\|x_i^{t+2} - x_i^{t+1}\|^2 + \left(4L^2 + \frac{4}{\beta^2}\right)\|e_i^{t+1}\|^2 + \frac{1}{6L^2}(w_i^{t+1} - w_i^t),
\end{aligned}
\tag{90}
$$

where the first inequality uses the Lipshcitz continuity of $\nabla F$.

For the third coordinate on the right hand side of equation 88, using equation 34, we have that

$$
\begin{aligned}
&d^2(0, \partial \tilde{g}(Z^{t+1}) + \frac{1}{\beta}(X^{t+1} - Z^{t+1}) - \frac{2}{\beta}(W^{t+1} - Z^{t+1})) \\
&\leq \|\frac{1}{\beta}(2W^{t+1} - X^{t+1} - Z^{t+1}) + \frac{1}{\beta}(X^{t+1} - Z^{t+1}) - \frac{2}{\beta}(W^{t+1} - Z^{t+1})\|^2 \\
&= 0.
\end{aligned}
\tag{91}
$$

Denoting $E^t = (e_1^t, \ldots, e_n^t)$ and combining this with equation 88 and equation 90 gives that

$$d^2(0, \partial H(X^{t+1}, W^{t+1}, Z^{t+1}, Y^{t+1}, W^t, Y^t) \leq \frac{1}{\beta^2}\|X^{t+2} - X^{t+1}\|^2 + \frac{16\delta^2}{\beta^2}\|W^{t+1} - W^t\|^2$$

$$+ \frac{16}{\beta^2}\|X^{t+2} - X^{t+1}\|^2 + \left(4L^2 + \frac{4}{\beta^2}\right)\|E^{t+1}\|^2 + \frac{8\delta^2}{\beta^2}\|W^{t+1} - W^t\|^2 + \frac{2}{3L^2}\Upsilon_t$$

$$= \left(\frac{1}{\beta^2} + \frac{16}{\beta^2}\right)\|X^{t+2} - X^{t+1}\|^2 + \left(\frac{16\delta^2}{\beta^2} + \frac{8\delta^2}{\beta^2}\right)\|W^{t+1} - W^t\|^2$$

$$+ \left(4L^2 + \frac{4}{\beta^2}\right)\|E^{t+1}\|^2 + \frac{2}{3L^2}\Upsilon_t$$

$$\overset{a}{\leq} \left(\frac{1}{\beta^2} + \frac{16}{\beta^2}\right)\|X^{t+2} - X^{t+1}\|^2 + \left(\frac{16\delta^2}{\beta^2} + \frac{8\delta^2}{\beta^2}\right)\|W^{t+1} - W^t\|^2$$

$$+ \left(4L^2\frac{4}{\beta^2}\right)\frac{1}{(\frac{1}{\beta} - L)^2}C\epsilon_w\mathbb{E}\Upsilon_{t+1} + \frac{2}{3L^2}\Upsilon_t$$

$$\leq \left(\frac{1}{\beta^2} + \frac{16}{\beta^2}\right)\|X^{t+2} - X^{t+1}\|^2 + \left(\frac{16\delta^2}{\beta^2} + \frac{8\delta^2}{\beta^2}\right)\|W^{t+1} - W^t\|^2$$

$$+ \left(4L^2\frac{4}{\beta^2}\right)\frac{1}{(\frac{1}{\beta} - L)^2}C\epsilon_w\mathbb{E}\left(\frac{1}{2}\Upsilon_t + 6L^2\|W^t - W^{t+1}\|^2\right) + \frac{2}{3L^2}\Upsilon_t,$$

$$(92)$$

where (a) uses equation 37 and the last inequality uses equation 38. Now we bound $\|X^{t+2} - X^{t+1}\|^2$. Recalling equation 38, it holds that

$$\|X^{t+2} - X^{t+1}\|^2 \leq (1 + \beta L)^2 (1 + \kappa)\Upsilon_{t+2} + (1 + \beta L)^2 \left(1 + \frac{1}{\kappa}\right)\frac{2}{(\frac{1}{\beta} - L)^2}C\epsilon_w\Upsilon_{t+1}$$

$$\leq (1 + \beta L)^2 (1 + \kappa)\Upsilon_{t+2} + (1 + \beta L)^2 \left(1 + \frac{1}{\kappa}\right)\frac{2}{(\frac{1}{\beta} - L)^2}C\epsilon_w\left(\frac{1}{2}\Upsilon_t + 6L^2\|W^t - W^{t+1}\|^2\right).$$

$$(93)$$

where the last inequality use equation 21. In addition, summing equation 21 from $t + 1$ to $t + 2$, we have that

$$\Upsilon_{t+2} \leq \frac{1}{2}(\Upsilon_t - \Upsilon_{t+2}) + 6L^2\|W^{t+1} - W^{t+2}\|^2 + 6L^2\|W^t - W^{t+1}\|^2$$

$$\leq \frac{1}{2}\mathbb{E}\Upsilon_t + 6L^2\|W^{t+1} - W^{t+2}\|^2 + 6L^2\|W^t - W^{t+1}\|\cdot$$

$$(94)$$

Combining equation 93, equation 94 and equation 92, we see that there exist $\Gamma_1'$, $\Gamma_2'$ and $\Gamma_3'$ such that

$$d^2(0, \partial H(X^{t+1}, W^{t+1}, Z^{t+1}, Y^{t+1}, W^t, Y^t)) \leq \Gamma_1'\|W^{t+2} - W^{t+1}\|^2 + \Gamma_2'\|W^{t+1} - W^t\|^2 + \Gamma_3'\Upsilon_t.$$

$$(95)$$

Combining this with the fact that $a^2 + b^2 + c^2 < (a + b + c)^2$ for any $a > 0$, $b > 0$ and $c > 0$, the conclusion holds with $\Gamma_1 = \sqrt{\Gamma_1'}$, $\Gamma_2 = \sqrt{\Gamma_2'}$ and $\Gamma_3 = \Gamma_3'$. $\qquad\square$

Next, we show the proofs of Theorem 3. For convenience, we restate Corollary 3 as follows.

**Theorem 9.** *Let assumptions in Theorem 6 hold. Suppose Assumption 3 holds. Suppose in addition that H is a KL function with exponent $\alpha \in [0, 1)$. Then $\{(X^t, W^t, Z^t, Y^t)\}$ is convergent. In addition, denoting $(X^*, W^*, Z^*, Y^*) := \lim_t(X^t, W^t, Z^t, Y^t)$, it holds that*

*(I) If $\alpha = 0$, then $\{(X^t, W^t, Z^t)\}$ converges finitely and $\{W^t\}$ converges linearly for large t.*

*(II) If $\alpha \in (0, \frac{1}{2}]$, then there exist $a > 0$ and $\rho \in (0, 1)$ such that $\max\{\|W^t - W^*\|, \|X^t - X^*\|, \|Z^t - Z^*\|, \|Y^t - Y^*\|\} \leq a\rho^t$ for large t.*

*(III) If $\alpha \in (\frac{1}{2}, 1]$, then there exist $b > 0$ such that $\max\{\|W^t - W^*\|, \|X^t - X^*\|, \|Z^t - Z^*\|, \|Y^t - Y^*\|\} \leq bt^{-\frac{1}{4\alpha - 2}}$ for large t.*

*Proof.* We first show the global convergence and convergence rates of $\{W^t\}$. In the deterministic case, we have from Theorem 6 (i) that $\{H(X^t, W^t, Z^t, Y^t, W^{t-1}, Y^{t-1})\}$ is convergent. Denote its limit as $H_*$. For simplicity of the proofs, in the rest of the proof, we denote $H_t := H(X^t, W^t, Z^t, W^{t-1}, Y^{t-1})$. First, suppose there exists $t_0$ such that $H_t = H_*$. Since $\{H_t\}$ is nonincreasing and recalling equation 40, we know that $H_t \equiv H_*$ and $\|W^t - W^{t-1}\| = \|Z^{t+1} - Z^t\| = 0$ for all $t \geq t_0$. This implies that $W^t = w^{t_0}$ and $Z^{t+1} = z^{t_0}$ for all $t \geq t_0$. This together with equation 38 and equation 3 induces that $X^t = x^{t_0}$.

Now, we show the convergence of $\{Y^t\}$. Recalling equation 21, it holds that

$$\Upsilon_{t+1} \leq \frac{1}{2}(\Upsilon_t - \Upsilon_{t+1}) + 6L^2\|W^{t+1} - W^t\|^2$$

$$\Leftrightarrow \frac{3}{2}\Upsilon_{t+1} \leq \frac{1}{2}\Upsilon_t + 6L^2\|W^{t+1} - W^t\|^2.$$

Taking square root on both side of the second inequality in the above relation, we have that

$$\sqrt{\frac{3}{2}\Upsilon_{t+1}} \leq \sqrt{\frac{1}{2}\Upsilon_t + 6L^2\|W^{t+1} - W^t\|^2} \leq \sqrt{\frac{1}{2}\Upsilon_t} + \sqrt{6L^2}\|W^{t+1} - W^t\| \tag{96}$$

where the second inequality uses the fact that $a^2 + b^2 \leq (a+b)^2$ for any positive $a$ and $b$. Rearranging the above inequality, we have that

$$\sqrt{\Upsilon_{t+1}} \leq \frac{1}{\sqrt{3} - 1}(\sqrt{\Upsilon_t} - \sqrt{\Upsilon_{t+1}}) + \sqrt{\frac{12L^2}{3 - \sqrt{3}}}\|W^{t+1} - W^t\|. \tag{97}$$

Summing the above inequality from $t = 1$ to $T$, we have that

$$\sum_{t=1}^{T}\|Y^t - Y^{t+1}\| \leq \sum_{t=1}^{T}\sqrt{\Upsilon_{t+1}} \leq \frac{1}{\sqrt{3} - 1}\sqrt{\Upsilon_1} + \sqrt{\frac{12L^2}{3 - \sqrt{3}}}\sum_{t=1}^{T}\|W^{t+1} - W^t\|$$

Since $\{W^t\}$ converges finitely, $\sum_{t=1}^{\infty}\|W^{t+1} - W^t\| < \infty$. Thus, taking $T$ in the above inequality to infinity, we have that $\sum_{t=1}^{\infty}\|Y^t - Y^{t+1}\| < \infty$, implying that $\{W^t\}$ is convergent.

Next, we suppose that $H_t > H_*$ for all $t$. Since $H$ is a KL function and is constant on $\Omega$ thanks to Theorem 7 (ii), using Lemma 6 of Bolte et al. (2014), there exists $\epsilon > 0$, $a > 0$ and $\phi \in \Psi_a$ such that

$$\phi'(H(X, W, Z, Y, W', Y') - H_*)d(0, \partial H(X, W, Z, Y, W', Y')) \geq 1$$

when $(X, W, Z, Y, W', Y')$ belongs to the set that

$$d((X, W, Z, Y, W', Y'), \Omega) \leq \epsilon$$

$$\text{and}$$

$$H_* < H(X, W, Z, Y, W', Y') < H_* + a.$$

Denote the above set as $\mathcal{B}$. Thanks to Theorem 6 (ii), we know that $\lim_t d((X^t, W^t, Z^t, Y^t, W^{t-1}, Y^{t-1}), \Omega) = 0$. This together with the fact that $\{H_t\}$ is nonincreasing and convergent guaranteed by equation 40 and Theorem 6 (ii), we deduce that there exists $t_1$ such that $(X^t, W^t, Z^t, Y^t, W^{t-1}, Y^{t-1}) \in \mathcal{B}$ for any $t \geq t_1$. Thus, for $t \geq t_1$, it holds that

$$\phi'(H(X^t, W^t, Z^t, Y^t, W^{t-1}, Y^{t-1}) - H_*)d(0, \partial H(X^t, W^t, Z^t, Y^t, W^{t-1}, Y^{t-1}) \geq 1. \tag{98}$$

Using the concavity of $\phi$, the above inequality further implies that

$$(\phi(H_t - H_*) - \phi(H_{t+1} - H_*)))\, d(0, \partial H(X^t, W^t, Z^t, Y^t, W^{t-1}, Y^{t-1}))$$

$$\geq \phi'(H_t - H_*)d(0, \partial H(X^t, W^t, Z^t, Y^t, W^{t-1}, Y^{t-1}))\,(H_t - H_{t+1})$$

$$\geq H_t - H_{t+1} \geq \frac{\delta_\epsilon}{\beta}\|W^t - W^{t-1}\|^2,$$

where the last inequality uses equation 40. Combining the above inequality with equation 87, we have that

$$(\phi(H_t - H_*) - \phi(H_{t+1} - H_*)))\left(\Gamma_1\|W^{t+1} - W^t\| + \Gamma_2\|W^t - W^{t-1}\| + \Gamma_3\Upsilon_{t-1}\right)$$

$$\geq \frac{\delta_\epsilon}{\beta}\|W^t - W^{t-1}\|^2.$$

Rearranging and taking square roots on both sides of the inequality, we have that

$$\|W^t - W^{t-1}\|$$

$$\leq \sqrt{\frac{\beta}{\delta_\epsilon}\left(\phi(H_t - H_*) - \phi(H_{t+1} - H_*)\right)\left(\Gamma_1\|W^{t+1} - W^t\| + \Gamma_2\|W^t - W^{t-1}\| + \Gamma_3\Upsilon_{t-1}\right)}. \tag{99}$$

Combining equation 97 with equation 99 and denoting $\Gamma_4 := \max\{\Gamma_1, \Gamma_2, \Gamma_3\frac{1}{\sqrt{3}-1}, \Gamma_3\sqrt{\frac{12L^2}{3-\sqrt{3}}}\}$, we have that

$$\|W^t - W^{t-1}\|$$

$$\leq \frac{\beta\Gamma_4}{\delta_\epsilon}\left(\phi(H_t - H_*) - \phi(H_{t+1} - H_*)\right) \tag{100}$$

$$+ \frac{1}{4}\left(\|W^{t+1} - W^t\| + \|W^t - W^{t-1}\| + \|W^{t-2} - W^{t-1}\| + (\Upsilon_{t-2} - \Upsilon_{t-1})\right)$$

where the second inequality is because $\sqrt{ab} \leq \frac{1}{2}(a+b)$ for any positive $a$ and $b$.

Rearranging the above inequality, it holds that

$$\frac{1}{4}\|W^t - W^{t-1}\| \leq \frac{\beta\Gamma_4}{\delta_\epsilon}\left(\phi(H_t - H_*) - \phi(H_{t+1} - H_*)\right)$$

$$+ \frac{1}{4}\left(\|W^{t+1} - W^t\| - \|W^t - W^{t-1}\|\right)$$

$$+ \left(\|W^{t-2} - W^{t-1}\| - \|W^t - W^{t-1}\|\right) + \frac{1}{4}(\Upsilon_{t-2} - \Upsilon_{t-1})$$

Pick any $t_2 > t_1 + 1$. Sum the above inequality from $t = t_2$ to $T$, it holds that

$$\frac{1}{4}\sum_{t=t_2+1}^{T}\|W^t - W^{t-1}\|$$

$$\leq \frac{\beta\Gamma_4}{\delta_\epsilon}\left(\phi(H_{t_2+1} - H_*) - \phi(H_{T+1} - H_*)\right)) + \frac{1}{4}\left(\|W^{T+1} - W^T\| - \|W^{t_2+1} - W^{t_1}\|\right)$$

$$+ \frac{1}{4}\left(\|W^{t_2-2} - W^{t_2-1}\| - \|W^T - W^{T-1}\|\right)$$

$$\leq \frac{\beta\Gamma_4}{\delta_\epsilon}\phi(H_{t_2+1} - H_*) + \frac{1}{4}\|W^{T+1} - W^T\| + \frac{1}{4}\left(\|W^{t_2-2} - W^{t_2-1}\|\right),$$

where the second inequality uses the fact that $\phi(w) \geq 0$. Since $\lim_t \|W^{T+1} - W^T\| = 0$ thanks to equation 6 (ii), passing $T$ in the above inequality to infinity shows that

$$\frac{1}{4}\sum_{t=t_2+1}^{T}\|W^t - W^{t-1}\| \leq \frac{\beta\Gamma_4}{\delta_\epsilon}\phi(H_{t_2+1} - H_*) + \frac{1}{4}\left(\|W^{t_2-2} - W^{t_2-1}\|\right) < \infty. \tag{101}$$

Therefore, $\{W^t\}$ is convergent.

Next, we show the convergence rate of $\{W^t\}$. From the assumption, we know that $\phi(w) = cy^{1-\alpha}$ for some $c > 0$. Then $\phi'(w) = c(1-\alpha)y^{-\alpha}$. Consider the case $\alpha = 0$. If $H_t > H_*$ for all $t$, using equation 98, we deduce that

$$d(0, \partial H(X^t, W^t, Z^t, Y^t, W^{t-1}, Y^{t-1})) \geq \frac{1}{c}, \text{ for } t \geq t_1.$$

However, thanks to equation 87 and Theorem 6 (ii), we have that $\lim_t d(0, \partial H(X^t, W^t, Z^t, Y^t, W^{t-1}, Y^{t-1})) = 0$, a contradiction. Thus, when $\alpha = 0$, there exists $t_{00}$ such that $H_t = H_*$ for $t > t_{00}$. Due to the arguments at the beginning of this proof, we know in this case, $\{W^t\}$ converges finitely.

Now we consider the case where $\alpha \in (0, 1)$. Still, if there is a $t$ such that $H_t = H_*$, $\{W^t\}$ converges finitely. Thus, we only need to consider the case where $H_t > H_*$ for all $t$. Define

$S_t := \sum_{j=t} \|W^{j+1} - W^j\|$ and $\bar{H}_t = H_t - H^*$. Thanks to equation 101, $S_t$ is well defined. Using equation 101, for $t > t_1$, it holds that

$$S_t \leq \frac{2\beta \max\{\Gamma_1, \Gamma_2\}}{\delta_\epsilon} \phi(H_{t_2+1} - H_*) \leq \frac{2\beta \max\{\Gamma_1, \Gamma_2\}}{\delta_\epsilon} \phi(H_{t+1} - H_*) + \frac{1}{2}(S_{t-2} - S_t). \tag{102}$$

With this inequality, following the proofs in Theorem 4.3 of Wen et al. (2018) (beginning from (4.18) of Wen et al. (2018)), we have that

(i) If $\alpha \in (0, \frac{1}{2}]$, then there exist $a > 0$ and $\rho \in (0, 1)$ such that

$$\|W^t - W^*\| \leq S_t \leq a\rho^t \text{ for large } t. \tag{103}$$

(ii) If $\alpha \in (\frac{1}{2}, 1)$, then there exist $b > 0$ such that

$$\|W^t - W^*\| \leq S_t \leq bt^{-\frac{1}{4\alpha-2}} \text{ for large } t. \tag{104}$$

To show the convergence of $(X^t, Z^t, Y^t)$, we first show that $\{\Upsilon_t\}$ is summable. Summing equation 97 from $t = t_2$ to $T$, we know that

$$\sum_{t=t_2}^{T} \sqrt{\Upsilon_{t+1}} \leq \frac{1}{\sqrt{3}-1}(\sqrt{\Upsilon_{t_2}} - \sqrt{\Upsilon_{T+1}}) + \sqrt{\frac{12L^2}{3-\sqrt{3}}} \sum_{t_2}^{T} \|W^{t+1} - W^t\|$$

$$\leq \frac{1}{\sqrt{3}-1}(\sqrt{\Upsilon_{t_2}} - \sqrt{\Upsilon_{T+1}}) + \sqrt{\frac{12L^2}{3-\sqrt{3}}} \left( \frac{4\beta\Gamma_4}{\delta_\epsilon}\phi(H_{t_2+1} - H_*) + \frac{1}{4}\left(\|W^{t_2-2} - W^{t_2-1}\|\right) \right). \tag{105}$$

Taking $T$ in the above inequality to infinity, we deduce that $\sum_{t=t_2}^{\infty} \sqrt{\Upsilon_{t+1}} < \infty$.

Since $\|Y^{t+1} - Y^t\| \leq \sqrt{\Upsilon_{t+1}}$ by definition of $\Upsilon_t$, we deduce that $\|Y^{t+1} - Y^t\|$ is also summable and thus $\{Y^t\}$ is convergent to some $Y^*$. Furthermore, the above inequality show that

$$\|Y^{t_2} - Y^*\| \leq \sum_{t=t_2}^{\infty} \|Y^{t+1} - Y^t\| \leq \sum_{t=t_2}^{\infty} \sqrt{\Upsilon_{t+1}}. \tag{106}$$

Next we show that $\{X^t\}$ is convergent. Taking square root of equation 38 on both sides, we have that

$$\|X^{t+1} - X^t\| \leq \sqrt{(1+\beta L)^2 (1+\kappa) \Upsilon_{t+1} + (1+\beta L)^2 \left(1 + \frac{1}{\kappa}\right) \frac{2}{(\frac{1}{\beta} - L)^2} C\epsilon_w \Upsilon_t}$$

$$\leq \sqrt{(1+\beta L)^2 (1+\kappa)} \sqrt{\Upsilon_{t+1}} + \sqrt{(1+\beta L)^2 \left(1 + \frac{1}{\kappa}\right) \frac{2}{(\frac{1}{\beta} - L)^2} C\epsilon_w} \sqrt{\Upsilon_t}.$$

Since $\{\Upsilon_t\}$ is summable, the above inequality show that $\{\|X^{t+1} - X^t\|\}$ is summable and thus $\{X^t\}$ is convergent to some $X^*$. In addition, the above inequality shows that

$$\|X^{t_2} - X^*\| \leq \sum_{t=t_2}^{\infty} \|X^{t+1} - X^t\| \leq O(\sum_{t=t_2}^{\infty} \sqrt{\Upsilon_{t+1}} + \sum_{t=t_2}^{\infty} \sqrt{\Upsilon_t}). \tag{107}$$

This implies $\{X^t\}$ is convergent. Using equation 2, we deduce that $\{Z^t\}$ is convergent.

We next show the convergence rate of $\sum_{t=t}^{\infty} \sqrt{\Upsilon_t}$. Dividing both sides of equation 96 by $\sqrt{\frac{3}{2}}$, we have that

$$\sqrt{\Upsilon_{t+1}} \leq \frac{1}{\sqrt{3}\Upsilon_t} + \sqrt{2L^2}\|W^{t+1} - W^t\|.$$

Thus, summing the above inequality from $t_2$ to $T$, it holds that

$$\sum_{t=t_2}^{\infty} \sqrt{\Upsilon_t} \leq \sum_{t=t_2}^{\infty} \sqrt{\Upsilon_{t+1}} \leq \frac{1}{\sqrt{3}} \sum_{t=t_2}^{\infty} \sqrt{\Upsilon_t} + \sqrt{2L^2} \sum_{t=t_2}^{\infty} \|W^{t+1} - W^t\|.$$

Rearranging the above inequality, for any $t_2 > t_1 + 1$, we have that

$$\sum_{t=t_2}^{\infty} \sqrt{\Upsilon_t} \leq \frac{1}{1 - \frac{1}{\sqrt{3}}} \sqrt{2L^2} \sum_{t=t_2}^{\infty} \|W^{t+1} - W^t\| = \frac{1}{1 - \frac{1}{\sqrt{3}}} \sqrt{2L^2} S_{t_2}. \tag{108}$$

Combining this with equation 106, equation 107, equation 103 and equation 104, we deduce that the convergence rate of $\{(X^t, Y^t)\}$ is at least the same as that of $\{W^t\}$. Finally, using equation 2, we deduce that $\{Z^t\}$ is convergent and its convergence rate is at least the same as that of $\{W^t\}$. $\quad\square$

### C.1 Proofs of Proposition 3.

*Proof.* Fix an $x \in \text{dom } \partial G$. Let $y(x) = \arg\max_y F(x, y)$. Consider $F(\cdot, y(x))$. Since $F$ is strongly concave in $y$, we know that $y(x)$ is continuous, see Proposition 1 in Chen et al. (2021). From the assumption in this proposition, there exist $\epsilon(y(x))$, $c(y(x))$ and $a(y(x))$ such that

$$\text{dist}^{\frac{1}{\alpha}}(0, \partial_x F(\cdot, y(x))(\tilde{x})) \geq c(y(x))(F(\tilde{x},, y(x)) - F(x, y(x))$$

whenever $\tilde{x} \in \text{dom } \partial_x F(\cdot, y(x))$, $\|\tilde{x} - x\| \leq \epsilon(y(x))$ and $F(x, y(x)) < F(x, y(\tilde{x})) < F(\tilde{x}, y(x)) < F(x, y(x)) + a(y(x))$. Thanks to the continuity of $F(\cdot, y)$ for any fixed $y$, we suppose without loss of generality that $\epsilon(y(x))$ be small enough such that when $\|\tilde{x} - x\| \leq \epsilon(y(x))$, we have that $F(x, y(x)) < F(x, y(x)) + a(y(x))$. Thus, there exist $\epsilon(y(x))$, $c(y(x))$ and $a(y(x))$ such that

$$\text{dist}^{\frac{1}{\alpha}}(0, \partial F(\cdot, y(x))(\tilde{x})) \geq c(y(x))(F(\tilde{x}, y(x)) - F(x, y(x))) \tag{109}$$

whenever $\tilde{x} \in \text{dom } \partial_x F(\cdot, y(x))$ and $\|\tilde{x} - x\| \leq \epsilon(y(x))$.

Recalling the continuity assumptions on $c(y)$ as well as $\epsilon(y)$ the continuity of $y(x)$, there exists $\delta > 0$ small enough such that there exists $\epsilon \in (0, \inf_{\|\bar{x}-x\| \leq \delta} \epsilon(y(\bar{x})))$ and $\inf_{\|\bar{x}-x\| \leq \delta} c(y(\bar{x})) > 0$.

Now let $z$ be any point satisfying $\|z - x\| \leq \min\{\epsilon, \delta\}$ and $G(z) \geq G(x)$. Then by the definition of $y(x)$, it holds that

$$F(z, y(z)) - F(x, y(z)) \geq F(z, y(z)) - -F(x, y(x)) \geq 0. \tag{110}$$

For this $z$ using equation 109, there also exist $\epsilon(y(z))$ and $c(y(z))$ such that

$$\text{dist}^{\frac{1}{\alpha}}(0, \partial_x F(\tilde{x}, y(z))) \geq c(y(z))(F(\tilde{x}, y(z)) - F(x, y(z))) \tag{111}$$

whenever $\tilde{x} \in \text{dom } \partial F(\cdot, y(z))$ and $\|\tilde{x} - x\| \leq \epsilon(y(z))$. By assumption of this proposition, and by the choice of $z$, we have that

$$\|z - x\| \leq \epsilon < \inf_{\|\bar{x}-x\| \leq \delta} \epsilon(y(\bar{x})) \leq \epsilon(y(z)),$$

where the last inequality is because $\|z - x\| \leq \delta$. Thus, using equation 111, we have

$$\begin{aligned}
\text{dist}^{\frac{1}{\alpha}}(0, \partial_x F(z, y(z))) &\geq c(y(z))(F(z, y(z)) - F(x, y(z))) \\
&\geq c(F(z, y(z)) - F(x, y(z))) = c(F(z, y(z)) - F(x, y(x))) \\
&+ c(F(x, y(x)) - F(x, y(z))) \geq c(F(z, y(z)) - F(x, y(x))) \\
&= c(G(z) - G(x)),
\end{aligned}$$

where $c := \inf_{\|\bar{x}-x\| \leq \delta} c(y(\bar{x}))$, the second inequality is because $\|z - x\| \leq \min\{\epsilon, \delta\}$ and equation 110, the last inequality uses the definition of $y(x)$.

Thus, when $\|z - x\| \leq \delta$ and $G(z) \geq G(x)$, it holds that

$$\text{dist}^{\frac{1}{\alpha}}(0, \partial_x F(z, y(z))) \geq c(G(z) - G(x)).$$

When $G(z) < G(x)$, the above inequality holds trivially. Therefore, we deduce that

$$\text{dist}^{\frac{1}{\alpha}}(0, \partial G(z)) = \text{dist}(0, \nabla_x F(z, y(z)) + \partial g(x)) = \text{dist}(0, \partial_x F(z, y(z))) \geq c(G(z) - G(x)),$$

where the equality is from Danskin's theorem and Exercise 8.8 in Rockafellar & Wets (1998). $\quad\square$

## C.2 PROOFS OF REMARK 4

*Proof.* Fix any $\bar{\theta}$. By the continuity of $F(\cdot, \delta)$, it suffices to show that there exists $\epsilon(\delta)$ such that

$$F(\theta, \delta) - F(\bar{\theta}, \delta) \leq \text{dist}^2(0, \partial_\theta F(\theta, \delta)), \text{ for } |\theta| \leq \epsilon(\delta),$$

and $\epsilon(\delta)$ is continuous in $\delta$. Without loss of generality, we let $(x, y) = (0, 1)$. Then $F(\theta, \delta) = \underbrace{\log(1 + \exp(-\theta\delta))}_{\ell(\theta, \delta)} - c|\delta|^2 + \lambda|\theta|$. Thus,

$$\partial_\theta F(\theta, \delta) = \frac{-\delta \exp(-\delta\theta)}{1 + \exp(-\delta\theta)} + \lambda \partial|\theta|.$$

and

$$\text{dist}(0, \partial_\theta F(\theta, \delta)) = \begin{cases} \lambda - \frac{\delta \exp(-\delta\theta)}{1+\exp(-\delta\theta)}, & \theta \geq 0 \\ \lambda + \frac{\delta \exp(-\delta\theta)}{1+\exp(-\delta\theta)}, & \theta < 0. \end{cases} \tag{112}$$

Thus, for any $\epsilon > 0$ and any $|\theta| \leq \epsilon$, it holds that

$$\text{dist}^2(0, \partial_\theta F(\theta, \delta)) = \|\nabla_\theta F(\theta, \delta)\|^2 = (\lambda - \frac{\delta \exp(-\delta\theta)}{1 + \exp(-\delta\theta)})^2 \geq \max \left\{ (\lambda - \frac{|\delta|}{2})^2, (\lambda - |\delta|)^2 \right\}. \tag{113}$$

Now we divided $\bar{\theta}$ into three cases: $\bar{\theta} = 0$, $\bar{\theta} > 0$ and $\bar{\theta} < 0$.

Case I: $\bar{\theta} = 0$. In this case,

$$F(\theta, \delta) - F(0, \delta) = \log(1 + \exp(-\theta\delta)) + \lambda|\theta| - \log 2.$$

Let $\epsilon > 0$. When $|\theta| < \epsilon$, we have that

$$F(\theta, \delta) - F(0, \delta) \leq \log(1 + \exp(\epsilon|\delta|)) + \lambda\epsilon - \log 2 \leq \log(2\exp(\epsilon|\delta|)) + \lambda\epsilon - \log 2 \leq \epsilon(|\delta| + \lambda). \tag{114}$$

and

$$\text{dist}^2(0, \partial_\theta F(\theta, \delta)) \geq \left( \lambda - \frac{|\delta| \exp(|\delta||\theta|)}{1 + \exp(|\delta||\theta|)} \right)^2. \tag{115}$$

Note that

- If $|\delta| = \lambda$, then $\left( \lambda - \frac{\lambda \exp(|\delta||\theta|)}{1+\exp(\lambda|\theta|)} \right)^2 = \left( \frac{\lambda}{1+\exp(\lambda|\theta|)} \right)^2 \geq \left( \frac{\lambda}{1+\exp(\lambda\epsilon)} \right)^2$. Let $\epsilon_1(\delta) = \frac{\left( \frac{\lambda}{1+\exp(\lambda\epsilon)} \right)^2}{|\delta|+\lambda}$, we have that $\epsilon(|\delta| + \lambda) \leq \left( \lambda - \frac{\lambda \exp(|\delta||\theta|)}{1+\exp(\lambda|\theta|)} \right)^2$.

- If $\delta = 0$, then $\left( \lambda - \frac{\lambda \exp(|\delta||\theta|)}{1+\exp(\lambda|\theta|)} \right)^2 = \lambda^2$. Let $\epsilon_2(\delta) = \frac{\lambda^2}{|\delta|+\lambda}$, we have that $\epsilon(|\delta| + \lambda) \leq \left( \lambda - \frac{\lambda \exp(|\delta||\theta|)}{1+\exp(\lambda|\theta|)} \right)^2$.

- If $\delta \neq 0$ and $\lambda < \frac{1}{2}|\delta|$, then $\log\left( \frac{\lambda}{|\delta|-\lambda} \right) > 0$. Also, $\lambda = \frac{|\delta| \exp(|\delta||\theta|)}{1+\exp(|\delta||\theta|)}$ if and only if $|\theta| = \epsilon_3(\delta)$ with $\epsilon_3(\delta) := \frac{1}{|\delta|} \log\left( \frac{\lambda}{|\delta|-\lambda} \right)$. Thus, when $|\theta| < \frac{1}{2}\epsilon_{3.5}(\delta)$,

$$\left( \lambda - \frac{|\delta| \exp(|\delta||\theta|)}{1 + \exp(|\delta||\theta|)} \right)^2 > \left( \lambda - \frac{|\delta| \exp(\frac{1}{2}\epsilon_{3.5}(\delta)|\delta|)}{1 + \exp(\frac{1}{2}\epsilon_{3.5}(\delta)|\delta|)} \right)^2 > 0.$$

Letting $\epsilon_3(\delta) = \frac{\left( \lambda - \frac{|\delta| \exp(\frac{1}{2}\epsilon_3(\delta)|\delta|)}{1+\exp(\frac{1}{2}\epsilon_3(\delta)|\delta|)} \right)^2}{|\delta|+\lambda}$, we have that $\epsilon(|\delta| + \lambda) \leq \left( \lambda - \frac{\lambda \exp(|\delta||\theta|)}{1+\exp(\lambda|\theta|)} \right)^2$.

- If $\delta \neq 0$ and $\lambda \geq \frac{1}{2}|\delta|$, then $\lambda - \frac{|\delta| \exp(|\delta||\theta|)}{1+\exp(|\delta||\theta|)} > 0$. Thus,

$$\left(\lambda - \frac{|\delta| \exp(|\delta||\theta|)}{1 + \exp(|\delta||\theta|)}\right)^2 \geq \max\left\{(\lambda - \frac{|\delta|}{2})^2, (\lambda - |\delta|)^2\right\}.$$

Let $\epsilon_4(\delta) = \frac{\max\left\{(\lambda-\frac{|\delta|}{2})^2, (\lambda-|\delta|)^2\right\}}{|\delta|+\lambda}$, we have that $\epsilon(|\delta| + \lambda) \leq \left(\lambda - \frac{\lambda \exp(|\delta||\theta|)}{1+\exp(\lambda|\theta|)}\right)^2$.

Therefore, let $\epsilon(\delta) := \min_{i=1,2,3,4} \epsilon_i(\delta)$, we know that $\epsilon(\delta)$ is continuous and

$$\epsilon(|\delta| + \lambda) \leq \left(\lambda - \frac{|\delta| \exp(|\delta||\theta|)}{1 + \exp(|\delta||\theta|)}\right)^2.$$

This together with equation 114 and equation 115 shows that

$$F(\theta, \delta) - F(0, \delta) \leq \text{dist}^2(0, \partial_\theta F(\theta, \delta)), \text{ for } |\theta| \leq \epsilon(\delta).$$

Thus, $F(\cdot, \delta)$ satisfies the KL property at 0 with exponent $\alpha$ and constants $\epsilon(\delta)$.

Case II: $\bar{\theta} > 0$. Let $\epsilon > 0$. For any $\theta \in [\bar{\theta} - \epsilon, \bar{\theta} + \epsilon]$, we have that

$$F(\theta, \delta) - F(\bar{\theta}, \delta) \leq \log(1 + \exp(\theta|\delta|)) + \lambda\theta - \log(1 + \exp(-\bar{\theta}\delta)) - \lambda\bar{\theta}$$
$$\leq \log(2\exp(\theta|\delta|)) + \lambda\theta \leq \theta(|\delta| + \lambda) + \log 2 \leq (\bar{\theta} + \epsilon)(|\delta| + \lambda) + \log 2.$$

Following similar argument after (14) in Case I, we can show that there exists $\epsilon(\delta)$ continuous w.r.t $\delta$ such that $F(\cdot, \delta)$ satisfies the KL property at $\bar{\theta}$ with exponent $\alpha$ and constants $\epsilon(\delta)$.

Case III: $\bar{\theta} < 0$. Let $\epsilon > 0$. For any $\theta \in [\bar{\theta} - \epsilon, \bar{\theta} + \epsilon]$, we have that

$$F(\theta, \delta) - F(\bar{\theta}, \delta) \leq \log(1 + \exp(|\theta||\delta|)) + \lambda|\theta| - \log(1 + \exp(-\bar{\theta}\delta)) - \lambda|\bar{\theta}|$$
$$\leq \log(2\exp(|\theta||\delta|)) + \lambda|\theta| \leq |\theta|(|\delta| + \lambda) + \log 2 \leq (|\bar{\theta}| + \epsilon)(|\delta| + \lambda) + \log 2.$$

Following similar argument after (14) in Case I, we can show that there exists $\epsilon(\delta)$ continuous w.r.t $\delta$ such that $F(\cdot, \delta)$ satisfies the KL property at $\bar{\theta}$ with exponent $\alpha$ and constants $\epsilon(\delta)$.

$\square$

