# OpenReview forum: "A Fast Federated Method for Minimax Problems with Sequential Convergence Guarantees"
_ICLR.cc/2025/Conference — ICLR 2025 Conference Withdrawn Submission_

### Official Review · Reviewer_HJWW · 2024-10-31

**Soundness:** 3
**Presentation:** 2
**Contribution:** 2
**Rating:** 3
**Confidence:** 4

**Summary:**

Based on Douglas-Rachford splitting, the paper proposed a new algorithm (FFMDR) for federated minimax problems, and analyzed its convergence guarantees in nonconvex-strongly-concave, and KL-strongly-concave cases, it is claimed that their method has smaller sample complexity compared to existing federated minimax methods. Numerical experiments are provided to validate the outperformance.

**Strengths:**

1. Incorporate KL condition into the study, which is important while less studied in the field.
2. Better complexity compared to existing works.

**Weaknesses:**

1. Overclaim. I think some claims should be clarified. Here each $f_i=\sum_j f(\cdot,\cdot,\xi_j)$ also comes with a finite-sum formation, while in Local SGDA, FedNEST, FedSGDA papers (c.f. Table 1), their $f_i=E_j f(\cdot,\cdot,\xi_j)$ generally comes with a expectation formation. This difference is important in my opinion, because your SAGA (Proposition 2) cannot be applied in their cases (due to no access to the full objective $f_i$). So I think the comparison in Table 1 is not fair, and the outperformance should be further clarified.
2. Assumption 3 requires the subroutine output to be deterministic, which basically requires each client to use deterministic algorithms to solve local problems, which is impractical regarding the large-scale data character.
3. I don't understand the "nonsmooth" in Line 124 regarding the contribution. As far as I can see, the main component $f_i$ in your objective is differentiable, the regularizer $g$ is (implicitly) assumed to be prox-friendly. You mentioned in Line 130 "... nonsmoothness of the max function...", but Proposition 1 shows that the max function is smooth, which contrasts the nonsmoothness claim. Can you clarify it?
4. The writing is a bit sloppy in my opinion. Some notations are not clearly defined, for example $m$ and $\kappa$ in Proposition 2. The failure of model parameter convergence in Table 1 for other algorithms is also a bit misleading, because they didn't consider KL condition, and we should not expect such convergence in their nonconvex setting.

With that, I vote for a rejection, and expect a revision for better clarification on the authors' contribution. Thank you.

**Questions:**

/

---

### Official Review · Reviewer_xa1X · 2024-11-04

**Soundness:** 3
**Presentation:** 3
**Contribution:** 3
**Rating:** 6
**Confidence:** 3

**Summary:**

This paper introduces FFMDR, a new federated learning method for nonconvex strongly concave minimax problems, with performance guarantees provided through both theoretical analysis and numerical results.

**Strengths:**

1. This paper proposes FFMDR, a novel federated learning method for nonconvex strongly concave minimax problems, which allows partial client participation and addresses the data heterogeneity issue.
2. In the theoretical analysis, the authors demonstrate that FFMDR improves sample complexity from $O(\epsilon^{-3})$ to $O(\epsilon^{-2})$.
3. The authors provide convergence guarantees for the sequence of model parameters generated by FFMDR, showing that the sequences converge finitely, linearly, or sublinearly, depending on the KL exponent.
4. The authors relax the KL assumptions on the potential function by deducing the KL exponent of the maximizer-dependent potential function from that of the maximizer-free function..
5. The performance of FFMDR is experimentally evaluated on AUC maximization tasks, demonstrating that FFMDR outperforms existing federated minimax approaches.

**Weaknesses:**

1. In Algorithm 1, how can we determine if equation 10 is satisfied, given that the exact solution to equation 8 is unknown?
2. In the analysis of sequential convergence, the authors assume that equation 10 is deterministic and all clients attend the training at each round. How does this analysis extend to more general settings, such as when equation 10 is stochastic and only partial client participation is allowed?
3. The experiments are conducted on a single task—maximizing the area under the ROC curve for binary classification—using relatively simple datasets. More complex datasets and tasks (e.g., adversarial training, fairness learning) are recommended to comprehensively evaluate FFMDR's performance.
4. Given that the experimental results across different algorithms and settings, particularly in Table 2, are extremely close, more repetitions of the experiments are needed to mitigate the influence of stochasticity.


Minor:
1. In line 39, citations of the real-world applications of FL should be added.
2. In line 102, "provides" should be corrected to "provide".
3. Abbreviations should be defined upon their first appearance in the paper, e.g., "DR" in line 263.
4. In Proposition 2, one instance of "strongly convex" should be corrected to "strongly concave".

**Questions:**

Please refer to the Weaknesses section.

---

### Official Review · Reviewer_ku56 · 2024-11-04

**Soundness:** 2
**Presentation:** 2
**Contribution:** 2
**Rating:** 3
**Confidence:** 4

**Summary:**

The authors apply the Douglas-Rachford splitting algorithm to solve minimax problems in federated learning. Two contributions were claimed: (1) By applying a variance reduced algorithm to solve one of the subproblems, the authors claim to improve the complexity bound to $\kappa^2 /(n\epsilon^2)$ (for obtaining an epsilon-stationary solution); (2) The authors proved variable convergence under the KL assumption, while obtaining some new results on verifying the KL property. Small-scale experiments were conducted on a AUC maximization problem, which showed slight improvement over some baselines.

**Strengths:**

- The improved complexity result sounds interesting, although it is not clear if the comparison is always done under the same assumptions  and stationarity definition.

- The variable convergence result might be technically new for federated minimax problems, and could be of a minor, pedagogical value.

- The side result on verifying the KL property, in my opinion, might be the best contribution, although I have not checked its details and it does not seem to apply to the formulations in this work.

**Weaknesses:**

- The presentation needs to improve: there are many technical and writing issues that need to be fixed (see below). The inconsistencies in terms of assumption, theory and experiment hurt the significance of the claimed contributions.

- Table 2 and Figure 1 showed that the experimental improvements are marginal, and they do not support well the main theoretical results (not to mention the mismatch between Theorem 1 and the adoption of SGDA in experiments). It is perhaps also not a good practice to artificially zoom in the y-axis in Figure 1 to create the illusions of a bigger separation.

- The assumption behind the variable convergence result (namely deterministic updates) is rarely satisfied in federated learning and the results based on it are not overly interesting or significant: they are more or less expected and follow the well-known recipe.

**Questions:**

- Line 270 and line 279 mentioned that F is separable (in y), but this is not the case for Eq. (1), the problem setup or Eq (14), the experiments. I do not think you can treat y separately for each client. Can the authors explain?

- Proposition 2 is poorly presented: what is $(x^*, y^*), (w_\star, y_\star)$ and $(\bar x, \bar y)$? How does $\epsilon_w$ affect things? How do you go from Eq (9) to Eq (10)? Where does the randomness come from (do we assume each client runs some unbiased stochastic algorithm)?

- Assumption 2 is not the same as "allowing client drift:" how do you deal with stragglers or "unstable network conditions" (line 108)?

- Theorem 1 is difficult to read as well, due to the many terms that were introduced but not explained. For example, $\delta$ was introduced early on but later it was defined again as $\delta_\beta - \delta'$. What is $\delta'$ anyway? What are the roles of $\iota, \epsilon_w, \delta_\epsilon$? I was not able to parse the theorem and verify the claimed O(\kappa^2/\epsilon^2) complexity in Remark 2. In fact, I do not even know how we measure stationarity using $\epsilon$ (is it $d(0, \partial)$?)

- Proposition 3 requires g to be defined everywhere but the function U in Line 426 involves $\tilde g$, which is defined on a hyperplane at best (due to the product-space trick). How does Proposition 3 apply to the main problem studied in this work?

- Line 483: perhaps it is worth mentioning that Eq (13) is not semi-algebraic so how do you verify this? (The early parts gave the impression that semi-algebraic functions suffice, but here you need a heavier machinery.)

- Line 513: why do you use SGDA as the local solver in your experiments? Wouldn't that imply all of the experiments have nothing to do with the theoretical results? This discrepancy diminishes most of the claimed contributions.

---

### Official Review · Reviewer_ubaH · 2024-11-04

**Soundness:** 3
**Presentation:** 2
**Contribution:** 3
**Rating:** 5
**Confidence:** 3

**Summary:**

This paper proposes a new FL method, FFMDR, designed for minimax problems, which addresses client drift and data heterogeneity issues. The authors provide theoretical convergence guarantees, showing an improvement in sample complexity. They applied FFMDR to the FL AUC maximization problem, demonstrating superior performance compared to existing methods.

**Strengths:**

1. The paper provides a clear and thorough description of the minimax problem within the FL context, making readers quickly understand its background. The detailed related work and preliminary sections support a better understanding of the problem, which helps to clarify the novelty of the proposed method.
2. The authors utilize multiple real-world datasets to demonstrate the effectiveness of FFMDR. The empirical evaluation highlights the FFMDR's robustness and applicability to real FL scenarios, further supporting its contribution.

**Weaknesses:**

1. The experimental results in Table 2 and Fig 1 show that the improvement of FFMDR appears limited. This raises concerns about the improvement's reliability. It would be better to conduct multiple runs of experiments to report the model's average performance.
2. The number of clients in this paper is significantly lower than that typically used in FL scenarios. I would expect to see the results of increasing the client count to better align with standard FL settings.
3. As seen in Fig. 2, FFMDR's performance drops considerably when the number of participating clients per round is reduced across at least three datasets. It would be helpful to include other baseline comparisons under similar conditions. I am curious whether other methods also show this sensitivity to client participation levels.
4. The major concern is the authors' claim that FFMDR reduces sample complexity in solving minimax problems within the FL. The method essentially reformulates the minimax problem in Eq. (6) into another minimax problem in Eq. (8), which doesn't appear to directly lower the complexity of solving a single minimax problem. A more detailed explanation addressing this aspect would be valuable.
5. I gave a cursory review of the appendix to verify the technical accuracy of the derivations, although not all steps were closely examined. Some proof raises concerns due to unclear explanations or possible inaccuracies. For example:
- In L. 853, it is unclear how the stated first inequality was derived.
- In L.1275, the inequality might need a correction to the equality.
- In Eq. (52), the first term in the inequality doesn't appear in the following equation, and the sixth term in L.1309 seems to be missing a summation symbol.

**Questions:**

Refer to the weaknesses section.

---

### Official Review · Reviewer_qdcb · 2024-11-06

**Soundness:** 3
**Presentation:** 2
**Contribution:** 2
**Rating:** 3
**Confidence:** 4

**Summary:**

- This paper proposed a federated learning method for a minimax formulation of federated learning
- This method is motivated by allowing client drift and addressing the data heterogeneity.
- This method is proved to have improved sample complexity of $O(\epsilon^{-3})$ and convergence guarantees for model parameters under KL exponent assumptions
- The proposed method is validated on AUC maximization tasks.

**Strengths:**

- The literature review in the introduction section is logical and organized
- The writing is generally easy to follow

**Weaknesses:**

- Major weakness
	- "Sequential convergence" seems to be a major benefit of the proposed method. Yet, I am not sure what it means from the paper. In the Introduction section, "sequential convergence" is used without definition. In the Preliminaries section, where such concepts like this should be discussed, it is not mentioned at all. In the main text, the word "sequential" appears only once in the title of Section 4.2. I would suggest that the authors provide a clear definition of "sequential convergence" early in the paper, ideally in the Introduction or Preliminaries section. This would help readers better understand this key concept and its importance to the method.
- Undefined terms and notations
	- Equation (1): l and r are not defined
	- L78: "proximal operator" is not defined. I would recommend that the authors provide a brief definition of "proximal operator" when it is first introduced, as this may not be familiar to all readers.
	- L95: "complexity of federated learning methods" -> "complexity of optimization algorithms" or "algorithmic complexity".
	- L97: "convergence of the model parameters themselves" -> "statistical convergence" (if I understand correctly)
	- L195: distance is denoted as $d$, where in the rest of the paper, it is referred to as $\text{dist}$ (L96, L247)
- Writing
	- L93: "Our work introduces methods that offer convergence guarantees without relying on these heterogeneity bounds". When introducing the proposed methods, you may want to (1) describe the method itself, and (2) clarify the motivations. Put it another way, instead of just saying, "we don't do A", say "we do B" and "why we choose B over A ". For example, you may say this: "Our work introduces methods that offer convergence guarantees under (some assumption), hence do not require these (stringent) heterogeneity bounds."
- Grammatical errors, typos, and minor issues
	- L52: it is necessary (to) develop ...
	- Inconsistent format: L237: *equation* 1, L285: equation 6. I think in a more accepted reference, "equation" should be "Equation" (capitalized and non-italic) and the number should parenthesized (e.g., by using the macro \eqref)

**Questions:**

- L74: What does "proper closed function" mean?
- L85: What does "client drift" mean? From the context, it sounds like stragglers in distributed/federated learning

---

### Note · Authors · 2024-12-11

I have read and agree with the venue's withdrawal policy on behalf of myself and my co-authors.